# IMPROVING MUTUAL INFORMATION ESTIMATION WITH ANNEALED AND ENERGY-BASED BOUNDS

**Rob Brekelmans**[*,1]            **Sicong Huang**[*,2,3]            **Marzyeh Ghassemi**[2,4]

**Greg Ver Steeg**[1]            **Roger Grosse**[2,3]            **Alireza Makhzani**[2,3]

[1] Information Sciences Institute, University of Southern California
[2] Vector Institute      [3] University of Toronto      [4] MIT EECS / IMES / CSAIL

## ABSTRACT

Mutual information (MI) is a fundamental quantity in information theory and machine learning. However, direct estimation of MI is intractable, even if the true joint probability density for the variables of interest is known, as it involves estimating a potentially high-dimensional log partition function. In this work, we present a unifying view of existing MI bounds from the perspective of importance sampling, and propose three novel bounds based on this approach. Since a tight MI bound without density information requires a sample size exponential in the true MI, we assume either a single marginal or the full joint density information is known. In settings where the full joint density is available, we propose Multi-Sample Annealed Importance Sampling (AIS) bounds on MI, which we demonstrate can tightly estimate large values of MI in our experiments. In settings where only a single marginal distribution is known, we propose *Generalized* IWAE (GIWAE) and MINE-AIS bounds. Our GIWAE bound unifies variational and contrastive bounds in a single framework that generalizes INFONCE, IWAE, and Barber-Agakov bounds. Our MINE-AIS method improves upon existing energy-based methods such as MINE-DV and MINE-F by directly optimizing a tighter lower bound on MI. MINE-AIS uses MCMC sampling to estimate gradients for training and Multi-Sample AIS for evaluating the bound. Our methods are particularly suitable for evaluating MI in deep generative models, since explicit forms of the marginal or joint densities are often available. We evaluate our bounds on estimating the MI of VAEs and GANs trained on the MNIST and CIFAR datasets, and showcase significant gains over existing bounds in these challenging settings with high ground truth MI.

## 1 INTRODUCTION

Mutual information (MI) is among the most general measures of dependence between two random variables. Among other applications in machine learning, MI has been used for both training (Alemi et al., 2016; 2018; Chen et al., 2016; Zhao et al., 2018) and evaluating (Alemi & Fischer, 2018; Huang et al., 2020) generative models. Furthermore, successes in neural network function approximation have encouraged a wave of variational or contrastive methods for MI estimation from samples only (Belghazi et al., 2018; van den Oord et al., 2018; Poole et al., 2019). However, McAllester & Stratos (2020) have shown strong theoretical limitations on any estimator based on direct sampling without an analytic form of at least one marginal distribution. In light of these limitations, we consider MI estimation in settings where a single marginal or the full joint distribution are known.

In this work, we view MI estimation from the perspective of importance sampling. Using a general approach for constructing extended state space bounds on MI, we combine insights from importance-weighted autoencoder (IWAE) (Burda et al., 2016; Sobolev & Vetrov, 2019) and annealed importance sampling (AIS) (Neal, 2001) to propose *Multi-Sample* AIS bounds in Sec. 3. We empirically show that this approach can tightly estimate large values of MI when the full joint distribution is known.

---

*Equal Contribution. brekelma@usc.edu; {huang, makhzani}@cs.toronto.edu. See arXiv for extended paper.

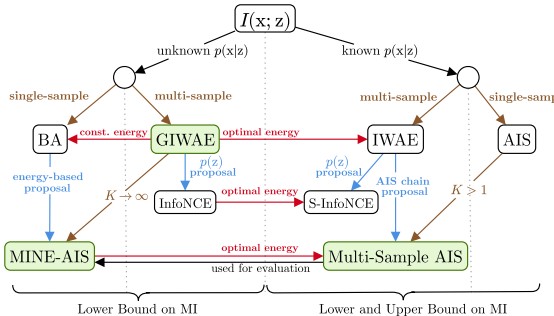

Figure 1: Schematic of MI bounds discussed in this paper. Green shading indicates our contributions, while columns and gold labels indicate single- or multi-sample bounds. Blue arrows indicate special cases using the indicated proposal distribution. Relationships based on learned critic functions are indicated by red arrows. We obtain only lower bounds on MI with unknown $p(\mathbf{x}|\mathbf{z})$, but both upper and lower bounds with known $p(\mathbf{x}|\mathbf{z})$. All bounds require a known marginal $p(\mathbf{z})$ for evaluation, apart from (Structured) INFO-NCE.

Our importance sampling perspective also suggests improved MI lower bounds that assume access to only joint samples for optimization, but require a single marginal distribution for evaluation. In Sec. 2.4, we propose *Generalized* IWAE (GIWAE), which generalizes both IWAE and INFONCE (Poole et al., 2019) and highlights how variational learning can complement multi-sample contrastive estimation to improve MI lower bounds. Finally, in Sec. 4 we propose MINE-AIS, which optimizes a tighter lower bound than MINE (Belghazi et al., 2018), called the *Implicit Barber-Agakov Lower bound* (IBAL). We demonstrate that the IBAL corresponds to the infinite-sample limit of the GIWAE lower bound, although our proposed energy-based training scheme involves only a single 'negative' contrastive sample obtained using Markov Chain Monte Carlo (MCMC). MINE-AIS then uses Multi-Sample AIS to evaluate the lower bound on MI, and shows notable improvement over existing variational methods in the challenging setting of MI estimation for deep generative models. We summarize the MI bounds discussed in this paper and the relationships between them in Fig. 1.

## 1.1 Problem Setting

The mutual information between two random variables $\mathbf{x}$ and $\mathbf{z}$ with joint distribution $p(\mathbf{x}, \mathbf{z})$ is

$$I(\mathbf{x}; \mathbf{z}) = \mathbb{E}_{p(\mathbf{x}, \mathbf{z})} \left[ \log \frac{p(\mathbf{x}, \mathbf{z})}{p(\mathbf{x})p(\mathbf{z})} \right] = H(\mathbf{x}) - H(\mathbf{x}|\mathbf{z}) = \mathbb{E}_{p(\mathbf{x}, \mathbf{z})}[\log p(\mathbf{x}|\mathbf{z})] - \mathbb{E}_{p(\mathbf{x})}[\log p(\mathbf{x})], \quad (1)$$

where $H(\mathbf{x}|\mathbf{z})$ denotes the conditional entropy $-\mathbb{E}_{p(\mathbf{x}, \mathbf{z})} \log p(\mathbf{x}|\mathbf{z})$. We primarily focus on bounds that assume either a single marginal distribution or the full joint distribution are available. A natural setting where the full joint distribution is available is estimating MI in deep generative models between the latent variables, with a known prior $\mathbf{z} \sim p(\mathbf{z})$, and data $\mathbf{x} \sim p(\mathbf{x})$ simulated from the model (Alemi & Fischer, 2018). [1] Settings where only a single marginal is known appear, for example, in simulation-based inference (Cranmer et al., 2020), where information about input parameters $\theta$ is known and a simulator can generate $\mathbf{x}$ for a given $\theta$, but the likelihood $p(\mathbf{x}|\theta)$ is intractable.

While sampling from the posterior $p(\mathbf{z}|\mathbf{x})$ for an arbitrary $\mathbf{x}$ is often intractable, we can obtain a single posterior sample for $\mathbf{x} \sim p(\mathbf{x})$ in cases where samples from the joint distribution $p(\mathbf{x})p(\mathbf{z}|\mathbf{x})$ are available. We will refer to bounds which involve only a single posterior sample as *practical*, and those involving multiple posterior samples as *impractical*.

When the conditional $p(\mathbf{x}|\mathbf{z})$ is tractable to sample and evaluate, simple Monte Carlo sampling provides an unbiased, low variance estimate of the conditional entropy term in Eq. (1). In this case, the difficulty of MI estimation reduces to estimating the log partition function $\log p(\mathbf{x})$, for which importance sampling (IS) based methods are among the most well studied and successful solutions.

## 2 Unifying Mutual Information Bounds via Importance Sampling

In this section, we present a unified view of mutual information estimation from the perspective of extended state space importance sampling. This general approach provides a probabilistic interpretation of many existing MI bounds and will suggest novel extensions in Sec. 3 and Sec. 4.

### 2.1 A General Approach for Extended State Space Importance Sampling Bounds

To estimate the log partition function, we construct a proposal $q_{\text{PROP}}(\mathbf{z}_{\text{ext}}|\mathbf{x})$ and target $p_{\text{TGT}}(\mathbf{x}, \mathbf{z}_{\text{ext}})$ distribution over an extended state space, such that the normalization constant of $p_{\text{TGT}}(\mathbf{x}, \mathbf{z}_{\text{ext}})$ is $\mathcal{Z}_{\text{TGT}} = \int p_{\text{TGT}}(\mathbf{x}, \mathbf{z}_{\text{ext}}) d\mathbf{z}_{\text{ext}} = p(\mathbf{x})$ and the normalization constant of $q_{\text{PROP}}(\mathbf{x}, \mathbf{z}_{\text{ext}})$ is $\mathcal{Z}_{\text{PROP}} = 1$. Taking expectations of the log importance weights $\log p_{\text{TGT}}(\mathbf{x}, \mathbf{z}_{\text{ext}})/q_{\text{PROP}}(\mathbf{z}_{\text{ext}}|\mathbf{x})$ under the proposal

---

[1] An alternative, "encoding" MI between the real data and the latent code is often of interest (see App. N), but cannot be directly estimated using our methods due to the unavailability of $p_d(\mathbf{x})$ or $q(\mathbf{z}) = \int p_d(\mathbf{x})q(\mathbf{z}|\mathbf{x})d\mathbf{x}$.

and target, respectively, we obtain lower and upper bounds on the log partition function

$$\underbrace{\mathbb{E}_{q_{\text{PROP}}(\mathbf{z}_{\text{ext}}|\mathbf{x})}\left[\log\frac{p_{\text{TGT}}(\mathbf{x},\mathbf{z}_{\text{ext}})}{q_{\text{PROP}}(\mathbf{z}_{\text{ext}}|\mathbf{x})}\right]}_{\text{ELBO}(\mathbf{x};q_{\text{PROP}},p_{\text{TGT}})} \leq \log p(\mathbf{x}) \leq \underbrace{\mathbb{E}_{p_{\text{TGT}}(\mathbf{z}_{\text{ext}}|\mathbf{x})}\left[\log\frac{p_{\text{TGT}}(\mathbf{x},\mathbf{z}_{\text{ext}})}{q_{\text{PROP}}(\mathbf{z}_{\text{ext}}|\mathbf{x})}\right]}_{\text{EUBO}(\mathbf{x};q_{\text{PROP}},p_{\text{TGT}})}. \tag{2}$$

These bounds correspond to extended state space versions of the ELBO and EUBO. In particular, the gap in the lower bound is the forward KL divergence $D_{\text{KL}}[q_{\text{PROP}}(\mathbf{z}_{\text{ext}}|\mathbf{x})\|p_{\text{TGT}}(\mathbf{z}_{\text{ext}}|\mathbf{x})]$ and the gap in the upper bound equal to the reverse KL divergence $D_{\text{KL}}[p_{\text{TGT}}(\mathbf{z}_{\text{ext}}|\mathbf{x})\|q_{\text{PROP}}(\mathbf{z}_{\text{ext}}|\mathbf{x})]$. See App. A.

## 2.2 Barber-Agakov Lower and Upper Bounds

As a first example, consider the standard $\text{ELBO}(q_\theta)$ and $\text{EUBO}(q_\theta)$ bounds, which are derived from simple importance sampling using a variational distribution $q_\theta(\mathbf{z}|\mathbf{x})$ and $\mathbf{z}_{\text{ext}} = \mathbf{z}$ in Eq. (2). Plugging these lower and upper bounds on $\log p(\mathbf{x})$ into Eq. (1), we obtain upper and lower bounds on MI as

$$I_{\text{BA}_L}(q_\theta) := \mathbb{E}_{p(\mathbf{x},\mathbf{z})}\left[\log\frac{q_\theta(\mathbf{z}|\mathbf{x})}{p(\mathbf{z})}\right] \leq I(\mathbf{x};\mathbf{z}) \leq \mathbb{E}_{p(\mathbf{x})q_\theta(\mathbf{z}|\mathbf{x})}\left[\log\frac{q_\theta(\mathbf{z}|\mathbf{x})}{p(\mathbf{x},\mathbf{z})}\right] - H(\mathbf{x}|\mathbf{z}) =: I_{\text{BA}_U}(q_\theta). \tag{3}$$

The left hand side of Eq. (3) is the well-known Barber-Agakov (BA) bound (Barber & Agakov, 2003), which has a gap of $\mathbb{E}_{p(\mathbf{x})}[D_{\text{KL}}[p(\mathbf{z}|\mathbf{x})\|q_\theta(\mathbf{z}|\mathbf{x})]]$. We refer to the right hand side as the BA upper bound $I_{\text{BA}_U}(q_\theta)$, with a gap of $\mathbb{E}_{p(\mathbf{x})}[D_{\text{KL}}[q_\theta(\mathbf{z}|\mathbf{x})\|p(\mathbf{z}|\mathbf{x})]]$. In contrast to $I_{\text{BA}_U}(q_\theta)$, note that $I_{\text{BA}_L}(q_\theta)$ does not require access to the conditional $p(\mathbf{x}|\mathbf{z})$ to evaluate the bound.

## 2.3 Importance Weighted Autoencoder

The IWAE lower and upper bounds on $\log p(\mathbf{x})$ (Burda et al., 2016; Sobolev & Vetrov, 2019) improve upon simple importance sampling by extending the state space using multiple samples $\mathbf{z}_{\text{ext}} = \mathbf{z}^{(1:K)}$. Consider a proposal $q_{\text{PROP}}^{\text{IWAE}}(\mathbf{z}^{(1:K)}|\mathbf{x})$ with $K$ independent samples from a given variational distribution $q_\theta(\mathbf{z}|\mathbf{x})$. The extended state space target $p_{\text{TGT}}^{\text{IWAE}}(\mathbf{z}^{(1:K)}|\mathbf{x})$ is a mixture distribution involving a single sample from the posterior $p(\mathbf{z}|\mathbf{x})$ or joint $p(\mathbf{x},\mathbf{z})$ and $K-1$ samples from $q_\theta(\mathbf{z}|\mathbf{x})$

$$q_{\text{PROP}}^{\text{IWAE}}(\mathbf{z}^{(1:K)}|\mathbf{x}) := \prod_{s=1}^{K} q_\theta(\mathbf{z}^{(s)}|\mathbf{x}) \qquad p_{\text{TGT}}^{\text{IWAE}}(\mathbf{x},\mathbf{z}^{(1:K)}) := \frac{1}{K}\sum_{s=1}^{K} p(\mathbf{x},\mathbf{z}^{(s)}) \prod_{k=1,k\neq s}^{K} q_\theta(\mathbf{z}^{(k)}|\mathbf{x}). \tag{4}$$

The log importance weight $\log\frac{p_{\text{TGT}}^{\text{IWAE}}(\mathbf{x},\mathbf{z}^{(1:K)})}{q_{\text{PROP}}^{\text{IWAE}}(\mathbf{z}^{(1:K)}|\mathbf{x})}$ reduces to the familiar ratio in the IWAE objective, while the normalization constant of $p_{\text{TGT}}^{\text{IWAE}}(\mathbf{x},\mathbf{z}^{(1:K)})$ is $p(\mathbf{x})$. As in Sec. 2.1, taking expectations under the proposal or target yields a lower or upper bound, respectively,

$$\underbrace{\mathbb{E}_{\prod_{k=1}^{K} q_\theta(\mathbf{z}^{(k)}|\mathbf{x})}\left[\log\frac{1}{K}\sum_{i=1}^{K}\frac{p(\mathbf{x},\mathbf{z}^{(k)})}{q_\theta(\mathbf{z}^{(k)}|\mathbf{x})}\right]}_{\text{ELBO}_{\text{IWAE}}(\mathbf{x};q_\theta,K)} \leq \log p(\mathbf{x}) \leq \underbrace{\mathbb{E}_{p(\mathbf{z}^{(1)}|\mathbf{x})\prod_{k=2}^{K} q_\theta(\mathbf{z}^{(k)}|\mathbf{x})}\left[\log\frac{1}{K}\sum_{i=1}^{K}\frac{p(\mathbf{x},\mathbf{z}^{(k)})}{q_\theta(\mathbf{z}^{(k)}|\mathbf{x})}\right]}_{\text{EUBO}_{\text{IWAE}}(\mathbf{x};q_\theta,K)}. \tag{5}$$

As for the standard ELBO and EUBO, the gap in the lower and upper bounds are $D_{\text{KL}}[q_{\text{PROP}}^{\text{IWAE}}\|p_{\text{TGT}}^{\text{IWAE}}]$ and $D_{\text{KL}}[p_{\text{TGT}}^{\text{IWAE}}\|q_{\text{PROP}}^{\text{IWAE}}]$, respectively. See App. B for detailed derivations. As in Sec. 1.1, with known $p(\mathbf{x}|\mathbf{z})$, the lower and upper bounds on $\log p(\mathbf{x})$, $\text{ELBO}_{\text{IWAE}}(\mathbf{x};q_\theta,K)$ and $\text{EUBO}_{\text{IWAE}}(\mathbf{x};q_\theta,K)$ translate to upper and lower bounds on MI, which we denote as $I_{\text{IWAE}_U}(q_\theta,K)$ and $I_{\text{IWAE}_L}(q_\theta,K)$.

While it is well-known that increasing $K$ leads to tighter IWAE bounds (Burda et al., 2016; Sobolev & Vetrov, 2019), in App. B.2, we characterize the improvement of $\text{ELBO}_{\text{IWAE}}(q_\theta,K)$ over $\text{ELBO}(q_\theta)$ as a KL divergence. For $\text{EUBO}_{\text{IWAE}}(q_\theta,K)$, we show that the KL divergence measuring its improvement over $\text{EUBO}(q_\theta)$ is limited to $\log K$, which implies $I_{\text{IWAE}_L}(q_\theta,K) \leq I_{\text{BA}_L}(q_\theta) + \log K$.

## 2.4 Generalized IWAE

In this section, we consider a family of *Generalized* IWAE (GIWAE) lower bounds, which improve upon $I_{\text{BA}_L}(q_\theta)$ using multiple samples and a contrastive critic function $T_\phi(\mathbf{x},\mathbf{z})$, but do not require access to $p(\mathbf{x}|\mathbf{z})$. While similar bounds appear in (Lawson et al., 2019; Sobolev, 2019), we provide a thorough discussion of special cases, and empirical analysis for MI estimation in Sec. 5.2. We also show that our IBAL bound in Sec. 4 corresponds to the infinite sample limit of GIWAE.

To derive a probabilistic interpretation for GIWAE, we begin by further extending the state space of the IWAE target distribution in Eq. (4), using a uniform index variable $p(s) = \frac{1}{K} \forall s$ that specifies which sample $\mathbf{z}^{(s)} \sim p(\mathbf{z}|\mathbf{x})$ is drawn from the posterior

$$p_{\text{TGT}}^{\text{GIWAE}}(\mathbf{z}^{(1:K)}, s|\mathbf{x}) := \frac{1}{K} \, p(\mathbf{z}^{(s)}|\mathbf{x}) \prod_{k=1, k \neq s}^{K} q(\mathbf{z}^{(k)}|\mathbf{x}). \tag{6}$$

Note that marginalization over $s$ leads to the IWAE target $p_{\text{TGT}}^{\text{IWAE}}(\mathbf{z}^{(1:K)}|\mathbf{x})$ in Eq. (4). For the GIWAE extended state space proposal, consider a categorical index variable $q_{\text{PROP}}^{\text{GIWAE}}(s|\mathbf{z}^{(1:K)}, \mathbf{x})$ drawn using self-normalized importance sampling (SNIS), with weights calculated by a learned critic function $T_\phi$

$$q_{\text{PROP}}^{\text{GIWAE}}(\mathbf{z}^{(1:K)}, s|\mathbf{x}) = \left( \prod_{k=1}^{K} q(\mathbf{z}^{(k)}|\mathbf{x}) \right) q_{\text{PROP}}^{\text{GIWAE}}(s|\mathbf{z}^{(1:K)}, \mathbf{x}), \quad \text{where} \quad q_{\text{PROP}}^{\text{GIWAE}}(s|\mathbf{z}^{(1:K)}, \mathbf{x}) = \frac{e^{T_\phi(\mathbf{x}, \mathbf{z}^{(s)})}}{\sum\limits_{k=1}^{K} e^{T_\phi(\mathbf{x}, \mathbf{z}^{(k)})}}.$$

We discuss the GIWAE probabilistic interpretation in App. C.1, and find that only the GIWAE upper bound on $\log p(\mathbf{x})$ provides practical benefit in App. C.2. The corresponding MI lower bound is

$$I_{\text{GIWAE}_L}(q_\theta, T_\phi, K) = \underbrace{\mathbb{E}_{p(\mathbf{x},\mathbf{z})} \left[ \log \frac{q_\theta(\mathbf{z}|\mathbf{x})}{p(\mathbf{z})} \right]}_{I_{\text{BA}_L}(q_\theta)} + \underbrace{\mathbb{E}_{p(\mathbf{x})p(\mathbf{z}^{(1)}|\mathbf{x}) \prod\limits_{k=2}^{K} q_\theta(\mathbf{z}^{(k)}|\mathbf{x})} \left[ \log \frac{e^{T_\phi(\mathbf{x}, \mathbf{z}^{(1)})}}{\frac{1}{K} \sum_{i=1}^{K} e^{T_\phi(\mathbf{x}, \mathbf{z}^{(k)})}} \right]}_{0 \leq \text{contrastive term} \leq \log K}. \tag{7}$$

We observe that the GIWAE lower bound decomposes into the sum of two terms, where the first is the BA variational lower bound for $q_\theta(\mathbf{z}|\mathbf{x})$ and the second is a contrastive term which distinguishes a single *positive* sample drawn from $p(\mathbf{z}|\mathbf{x})$ from $K - 1$ *negative* samples drawn from $q_\theta(\mathbf{z}|\mathbf{x})$.

**Relationship with BA**  With a constant $T_\phi(\mathbf{x}, \mathbf{z}) = \text{const}$, the second term in GIWAE vanishes and we have $I_{\text{GIWAE}_L}(q, T_\phi = \text{const}, K) = I_{\text{BA}_L}(q_\theta)$ for all $K$. For $K = 1$, $I_{\text{GIWAE}_L}(q, T_\phi, K = 1)$ also equals the BA lower bound for all $T_\phi$. Similarly to BA, GIWAE requires access to the analytical form of $p(\mathbf{z})$ to *evaluate* the bound on MI. However, both the BA and GIWAE lower bounds can be used to *optimize* MI even if no marginal is available. See App. N. for detailed discussion.

**Relationship with InfoNCE**  When the prior $p(\mathbf{z})$ is used in place of $q_\theta(\mathbf{z}|\mathbf{x})$, we can recognize the second term in Eq. (7) as the INFONCE contrastive lower bound (van den Oord et al., 2018; Poole et al., 2019), with $I_{\text{INFONCE}_L}(T_\phi, K) = I_{\text{GIWAE}_L}(p(\mathbf{z}), T_\phi, K)$. From this perspective, the GIWAE lower bound highlights how variational learning can complement contrastive bounds to improve MI estimation beyond the known $\log K$ limitations of INFONCE (van den Oord et al., 2018).

**Relationship with IWAE**  The following proposition characterizes the relationship between $I_{\text{GIWAE}_L}(q_\theta, T_\phi, K)$ in Eq. (7) and $I_{\text{IWAE}_L}(q_\theta, K)$ from Sec. 2.3. See App. C.5 for proofs.

**Proposition 2.1** (Improvement of IWAE over GIWAE). *For a given $q_\theta(\mathbf{z}|\mathbf{x})$ and any $T_\phi(\mathbf{x}, \mathbf{z})$,*

$$I_{\text{IWAE}_L}(q_\theta, K) = I_{\text{GIWAE}_L}(q_\theta, T_\phi, K) + \mathbb{E}_{p(\mathbf{x})p_{\text{TGT}}^{\text{GIWAE}}(\mathbf{z}^{(1:K)}|\mathbf{x})} \left[ D_{\text{KL}}[p_{\text{TGT}}^{\text{GIWAE}}(s|\mathbf{z}^{(1:K)}, \mathbf{x}) \| q_{\text{PROP}}^{\text{GIWAE}}(s|\mathbf{z}^{(1:K)}, \mathbf{x})] \right].$$

**Corollary 2.2.** *For a given $q_\theta(\mathbf{z}|\mathbf{x})$ and $K > 1$, the optimal critic function is the true log importance weight up to an arbitrary constant: $T^*(\mathbf{x}, \mathbf{z}) = \log \frac{p(\mathbf{x}, \mathbf{z})}{q_\theta(\mathbf{z}|\mathbf{x})} + c(\mathbf{x})$. With this choice of $T^*(\mathbf{x}, \mathbf{z})$,*

$$I_{\text{GIWAE}_L}(q_\theta, T^*, K) = I_{\text{IWAE}_L}(q_\theta, K). \tag{8}$$

**Corollary 2.3.** *Suppose the critic function $T_\phi(\mathbf{x}, \mathbf{z})$ is parameterized by $\phi$, and $\exists \phi_0 \, s.t. \, \forall (\mathbf{x}, \mathbf{z}), \, T_{\phi_0}(\mathbf{x}, \mathbf{z}) = \text{const}$. For a given $q_\theta(\mathbf{z}|\mathbf{x})$, let $T_{\phi^*}(\mathbf{x}, \mathbf{z})$ denote the critic function that maximizes the GIWAE lower bound. Using Cor. 2.2, we have*

$$I_{\text{BA}_L}(q_\theta) \leq I_{\text{GIWAE}_L}(q_\theta, T_{\phi^*}, K) \leq I_{\text{IWAE}_L}(q, K) \leq I_{\text{BA}_L}(q_\theta) + \log K. \tag{9}$$

While the GIWAE lower bound on MI does not assume access to the full joint distribution, Cor. 2.2 suggests that the role of the critic function is to learn the true log importance weights for $q_\theta(\mathbf{z}|\mathbf{x})$. Thus, when $p(\mathbf{x}, \mathbf{z})$ is known, IWAE is always preferable to GIWAE. Cor. 2.3 shows that while $I_{\text{GIWAE}_L}(q, T_{\phi^*}, K)$ can improve upon BA, this improvement is at most logarithmic in $K$.

**Relationship with Structured INFONCE**  Since INFONCE and Structured INFONCE (S-INFONCE, see App. B.5, Poole et al. (2019)) are special cases of GIWAE and IWAE that use $p(\mathbf{z})$ as the variational distribution, Cor. 2.3 suggests the following relationship (see App. C.6)

$$0 \leq I_{\text{INFONCE}_L}(T_\phi, K) \leq I_{\text{S-INFONCE}_L}(K) \leq \log K. \tag{10}$$

| | IWAE | Single-Sample AIS | Independent Multi-Sample AIS | Independent Reverse Multi-Sample AIS | Coupled Reverse Multi-Sample AIS |
|---|---|---|---|---|---|
| **Practical?** | ✓ | ✓ | ✓ | ✗ | ✓ |
| **Target EUBO** $I_{LB}$ | | | | | |
| **Proposal ELBO** $I_{UB}$ | | | | | |

Figure 2: Extended state-space probabilistic interpretations of multi-sample AIS bounds. Forward chains are colored in blue, and backward chains are colored in red. Note that ELBOs and EUBOs are obtained by taking the expectation of the log importance weight $\log p_{\text{TGT}}(\cdot)/q_{\text{PROP}}(\cdot)$ under either the proposal or target distribution, and can then be translated to MI bounds.

# 3 MULTI-SAMPLE AIS BOUNDS FOR ESTIMATING MUTUAL INFORMATION

In the previous sections, we derived probabilistic interpretations of extended state space importance sampling bounds using multiple samples $K$, as in IWAE. In this section, we first review AIS, which extends the state space using $T$ intermediate densities. We then show that these approaches are complementary, and derive two practical *Multi-Sample* AIS methods that provide tighter bounds by combining insights from IWAE and AIS.

## 3.1 ANNEALED IMPORTANCE SAMPLING BACKGROUND

AIS (Neal, 2001) constructs a sequence of intermediate distributions $\{\pi_t(\mathbf{z})\}_{t=0}^T$, which bridge between a normalized initial distribution $\pi_0(\mathbf{z}|\mathbf{x})$ and target distribution $\pi_T(\mathbf{z}|\mathbf{x}) = p(\mathbf{z}|\mathbf{x})$ with unnormalized density $\pi_T(\mathbf{x}, \mathbf{z}) = p(\mathbf{x}, \mathbf{z})$ and normalizing constant $\mathcal{Z}_T(\mathbf{x}) = p(\mathbf{x})$. A common choice for intermediate distributions is the geometric mixture path parameterized by $\{\beta_t\}_{t=0}^T$

$$\pi_t(\mathbf{z}|\mathbf{x}) \coloneqq \frac{\pi_0(\mathbf{z}|\mathbf{x})^{1-\beta_t} \, \pi_T(\mathbf{x}, \mathbf{z})^{\beta_t}}{\mathcal{Z}_t(\mathbf{x})}, \quad \text{where} \quad \mathcal{Z}_t(\mathbf{x}) \coloneqq \int \pi_0(\mathbf{z}|\mathbf{x})^{1-\beta_t} \, \pi_T(\mathbf{x}, \mathbf{z})^{\beta_t} \, d\mathbf{z}. \quad (11)$$

In the probabilistic interpretation of AIS, we consider an extended state space proposal $q_{\text{PROP}}^{\text{AIS}}(\mathbf{z}_{0:T}|\mathbf{x})$, obtained by sampling from the initial $\pi_0(\mathbf{z}|\mathbf{x})$ and constructing transitions $\mathcal{T}_t(\mathbf{z}_t|\mathbf{z}_{t-1})$ which leave $\pi_{t-1}(\mathbf{z}|\mathbf{x})$ invariant. The target distribution $p_{\text{TGT}}^{\text{AIS}}(\mathbf{z}_{0:T}|\mathbf{x})$ is given by running the reverse transitions $\tilde{\mathcal{T}}_t(\mathbf{z}_{t-1}|\mathbf{z}_t)$ starting from a target or posterior sample $\pi_T(\mathbf{z}|\mathbf{x})$, as shown in Fig. 2,

$$q_{\text{PROP}}^{\text{AIS}}(\mathbf{z}_{0:T}|\mathbf{x}) \coloneqq \pi_0(\mathbf{z}_0|\mathbf{x}) \prod_{t=1}^T \mathcal{T}_t(\mathbf{z}_t|\mathbf{z}_{t-1}), \quad p_{\text{TGT}}^{\text{AIS}}(\mathbf{x}, \mathbf{z}_{0:T}) \coloneqq \pi_T(\mathbf{x}, \mathbf{z}_T) \prod_{t=1}^T \tilde{\mathcal{T}}_t(\mathbf{z}_{t-1}|\mathbf{z}_t). \quad (12)$$

Taking expectations of the log importance weights under the proposal and target again yields a lower and upper bound on the log partition function $\log p(\mathbf{x})$ (App. E). These single-chain lower and upper bounds translate to upper and lower bounds on MI, $I_{\text{AIS}_U}(\pi_0, T)$ and $I_{\text{AIS}_L}(\pi_0, T)$, which were suggested for MI estimation in the blog post of Sobolev (2019). To characterize the bias reduction for AIS with increasing $T$, we prove the following proposition.

**Proposition 3.1** (Complexity in $T$). *Assuming perfect transitions and a geometric annealing path with linearly-spaced $\{\beta_t\}_{t=1}^T$, the sum of the gaps in the AIS sandwich bounds on MI, $I_{\text{AIS}_U}(\pi_0, T) - I_{\text{AIS}_L}(\pi_0, T)$, reduces linearly with increasing $T$.*

See App. D.1 for a proof. In our experiments in Sec. 5, we will find that this linear bias reduction in $T$ is crucial for achieving tight MI estimation when both $p(\mathbf{z})$ and $p(\mathbf{x}|\mathbf{z})$ are known. However, we can further tighten these AIS bounds using multiple annealing chains ($K > 1$), and we present two practical extended state space approaches in the following sections.

## 3.2 INDEPENDENT MULTI-SAMPLE AIS BOUNDS

To derive *Independent* Multi-Sample AIS (IM-AIS), we construct an extended state space proposal by running $K$ independent AIS forward chains $\mathbf{z}_{0:T}^{(k)} \sim q_{\text{PROP}}^{\text{AIS}}$ in parallel. Similarly to the IWAE upper bound (Eq. (5)), the extended state space target involves selecting a index $s$ uniformly at random, and running a backward AIS chain $\mathbf{z}_{0:T}^{(s)} \sim p_{\text{TGT}}^{\text{AIS}}$ starting from a true posterior sample $\mathbf{z}_T \sim p(\mathbf{z}|\mathbf{x})$. The remaining $K - 1$ samples are obtained by running forward AIS chains, as visualized in Fig. 2

$$q_{\text{PROP}}^{\text{IM-AIS}}(\mathbf{z}_{0:T}^{(1:K)}|\mathbf{x}) := \prod_{k=1}^{K} q_{\text{PROP}}^{\text{AIS}}(\mathbf{z}_{0:T}^{(k)}|\mathbf{x}), \qquad p_{\text{TGT}}^{\text{IM-AIS}}(\mathbf{x}, \mathbf{z}_{0:T}^{(1:K)}) := \frac{1}{K}\sum_{s=1}^{K} p_{\text{TGT}}^{\text{AIS}}(\mathbf{x}, \mathbf{z}_{0:T}^{(s)}) \prod_{k=1, k \neq s}^{K} q_{\text{PROP}}^{\text{AIS}}(\mathbf{z}_{0:T}^{(k)}|\mathbf{x}), \quad (13)$$

where $q_{\text{PROP}}^{\text{AIS}}$ and $p_{\text{TGT}}^{\text{AIS}}$ were defined in Eq. (12). Note that sampling from the extended state space target distribution is practical, as it only requires one sample from the true posterior distribution.

As in Sec. 2.1, taking the expectation of the log unnormalized density ratio under the proposal and target yields lower and upper bounds on $\log p(\mathbf{x})$, respectively,

$$\underbrace{\mathbb{E}_{\mathbf{z}_{0:T}^{(1:K)} \sim q_{\text{PROP}}^{\text{AIS}}}\left[\log \frac{1}{K}\sum_{k=1}^{K}\frac{p_{\text{TGT}}^{\text{AIS}}(\mathbf{x}, \mathbf{z}_{0:T}^{(k)})}{q_{\text{PROP}}^{\text{AIS}}(\mathbf{z}_{0:T}^{(k)}|\mathbf{x})}\right]}_{\text{ELBO}_{\text{IM-AIS}}(\mathbf{x}; \pi_0, K, T)} \leq \log p(\mathbf{x}) \leq \underbrace{\mathbb{E}_{\substack{\mathbf{z}_{0:T}^{(1)} \sim p_{\text{TGT}}^{\text{AIS}} \\ \mathbf{z}_{0:T}^{(2:K)} \sim q_{\text{PROP}}^{\text{AIS}}}}\left[\log \frac{1}{K}\sum_{k=1}^{K}\frac{p_{\text{TGT}}^{\text{AIS}}(\mathbf{x}, \mathbf{z}_{0:T}^{(k)})}{q_{\text{PROP}}^{\text{AIS}}(\mathbf{z}_{0:T}^{(k)}|\mathbf{x})}\right]}_{\text{EUBO}_{\text{IM-AIS}}(\mathbf{x}; \pi_0, K, T)}, \quad (14)$$

which again have KL divergences as the gap in their bounds (see App. E). Independent Multi-Sample AIS reduces to IWAE for $T = 1$, and reduces to single-sample AIS for $K = 1$. Both upper and lower bounds are tight as $K \to \infty$ or $T \to \infty$, and translate to lower and upper bounds on MI as in Sec. 1.1. In App. E.3, we show that the Independent Multi-Sample AIS *lower* bound on MI is limited to logarithmic improvement over single-sample AIS, with $I_{\text{IM-AIS}_L}(\pi_0, K, T) \leq I_{\text{AIS}_L}(\pi_0, T) + \log K$.

## 3.3 COUPLED REVERSE MULTI-SAMPLE AIS BOUNDS

We can exchange the role of the forward and backward annealing chains in Independent Multi-Sample AIS to obtain alternative bounds on the log partition function. We define *Independent Reverse Multi-Sample* AIS (IR-AIS) using the following proposal and target distribution, as shown in Fig. 2.

$$q_{\text{PROP}}^{\text{IR-AIS}}(\mathbf{x}, \mathbf{z}_{0:T}^{(1:K)}) := \frac{1}{K}\sum_{s=1}^{K} q_{\text{PROP}}^{\text{AIS}}(\mathbf{z}_{0:T}^{(s)}|\mathbf{x}) \prod_{k=1, k \neq s}^{K} p_{\text{TGT}}^{\text{AIS}}(\mathbf{x}, \mathbf{z}_{0:T}^{(k)}), \qquad p_{\text{TGT}}^{\text{IR-AIS}}(\mathbf{x}, \mathbf{z}_{0:T}^{(1:K)}) := \prod_{k=1}^{K} p_{\text{TGT}}^{\text{AIS}}(\mathbf{x}, \mathbf{z}_{0:T}^{(k)}).$$

Note that partition function ratio is $\mathcal{Z}_{\text{TGT}}/\mathcal{Z}_{\text{PROP}} = p(\mathbf{x})^K/p(\mathbf{x})^{K-1} = p(\mathbf{x})$. Using these distributions, we derive $\log p(\mathbf{x})$ and MI bounds in App. G. However, the these bounds will be impractical in most settings since they require multiple true posterior samples (see Sec. 1.1).

To address this, we propose *Coupled* Reverse Multi-Sample AIS (CR-AIS). As shown in Fig. 2, the extended state space target distribution initializes $K$ backward chains from a single target sample $\mathbf{z}_T \sim \pi_T(\mathbf{z}|\mathbf{x})$, with the remaining transitions $p_{\text{TGT}}^{\text{AIS}}(\mathbf{z}_{0:T-1}|\mathbf{z}_T)$ matching standard AIS in Eq. (12).

$$p_{\text{TGT}}^{\text{CR-AIS}}(\mathbf{z}_{0:T-1}^{(1:K)}, \mathbf{z}_T, \mathbf{x}) := \pi_T(\mathbf{z}_T, \mathbf{x}) \prod_{k=1}^{K} p_{\text{TGT}}^{\text{AIS}}(\mathbf{z}_{0:T-1}^{(k)}|\mathbf{z}_T, \mathbf{x}). \quad (15)$$

The extended state space proposal is obtained by selecting an index $s$ uniformly at random and running a single forward AIS chain. We then run $K - 1$ backward chains, all starting from the last state of the selected forward chain, as visualized in Fig. 2

$$q_{\text{PROP}}^{\text{CR-AIS}}(\mathbf{z}_{0:T-1}^{(1:K)}, \mathbf{z}_T|\mathbf{x}) := \frac{1}{K}\sum_{s=1}^{K} q_{\text{PROP}}^{\text{AIS}}(\mathbf{z}_{0:T-1}^{(s)}, \mathbf{z}_T|\mathbf{x}) \prod_{k=1, k \neq s}^{K} p_{\text{TGT}}^{\text{AIS}}(\mathbf{z}_{0:T-1}^{(k)}|\mathbf{z}_T, \mathbf{x}). \quad (16)$$

Taking the expected log ratio under the proposal and target yields lower and upper bounds on $\log p(\mathbf{x})$,

$$\underbrace{-\mathbb{E}_{\substack{\mathbf{z}_{0:T-1}^{(1)}, \mathbf{z}_T \sim q_{\text{PROP}}^{\text{AIS}}(\mathbf{z}_{0:T}|\mathbf{x}) \\ \mathbf{z}_{0:T-1}^{(2:K)} \sim p_{\text{TGT}}^{\text{AIS}}(\mathbf{z}_{0:T-1}|\mathbf{z}_T, \mathbf{x})}}\left[\log \frac{1}{K}\sum_{k=1}^{K}\frac{q_{\text{PROP}}^{\text{AIS}}(\cdot)}{p_{\text{TGT}}^{\text{AIS}}(\cdot)}\right]}_{\text{ELBO}_{\text{CR-AIS}}(\mathbf{x}; \pi_0, K, T)} \leq \log p(\mathbf{x}) \leq \underbrace{-\mathbb{E}_{\substack{\mathbf{z}_T \sim \pi_T(\mathbf{z}_T|\mathbf{x}) \\ \mathbf{z}_{0:T-1}^{(1:K)} \sim p_{\text{TGT}}^{\text{AIS}}(\mathbf{z}_{0:T-1}|\mathbf{z}_T, \mathbf{x})}}\left[\log \frac{1}{K}\sum_{k=1}^{K}\frac{q_{\text{PROP}}^{\text{AIS}}(\cdot)}{p_{\text{TGT}}^{\text{AIS}}(\cdot)}\right]}_{\text{EUBO}_{\text{CR-AIS}}(\mathbf{x}; \pi_0, K, T)}.$$

We show in App. H.3 that the Coupled Reverse Multi-Sample AIS *upper* bound on MI is limited to logarithmic improvement over single-sample AIS, with $I_{\text{CR-AIS}_U}(\pi_0, K, T) \geq I_{\text{AIS}_U}(\pi_0, T) - \log K$.

## 3.4 DISCUSSION

**Relationship with BDMC**   While Bidirectional Monte Carlo (BDMC) (Grosse et al., 2015; 2016) was the first method proposing multi-sample log partition function bounds using AIS chains, our probabilistic interpretations provide novel perspective on BDMC. Perhaps surprisingly, we find that BDMC lower and upper bounds do not correspond to the same extended state space proposal and target distributions. In particular, the BDMC lower bound on $\log p(\mathbf{x})$ corresponds to the lower bound of Independent Multi-Sample AIS (Fig. 2 Col. 4, Row 4), while the upper bound of BDMC matches the upper bound of Coupled Reverse Multi-Sample AIS (Fig. 2 Col. 6, Row 3).

**Effect of $K$ and $T$**   We have shown in Prop. 3.1 that Multi-Sample AIS bounds can achieve *linear* bias reduction with increasing $T$, although this computation must be done in serial fashion. While increasing $K$ involves parallel computation, its bias reduction is often only *logarithmic*, as we show for $I_{\text{IWAE}_L}(q_\theta, K)$ (Cor. B.3), $I_{\text{IM-AIS}_L}(\pi_0, K, T)$ (App. E.3), and $I_{\text{CR-AIS}_U}(\pi_0, K, T)$ (App. H.3). Based on these arguments, we recommend increasing $K$ until computation can no longer be parallelized on a given hardware and allocating all remaining resources to increasing $T$.

**Comparing Multi-Sample AIS Bounds**   In App. I Fig. 5, we compare performance of our various Multi-Sample AIS bounds in order to recommend which to use in practice. For the upper bound on MI, we recommend the Independent Multi-Sample AIS ELBO, or the forward direction of BDMC, since it uses independent samples and is not limited to $\log K$ improvement. The results are less conclusive for the lower bounds on MI. While the MI lower bound obtained from $\text{EUBO}_{\text{IM-AIS}}(\mathbf{x}; \pi_0, K, T)$ (RHS of Eq. (14)) can improve upon single-sample AIS by at most $\log K$ (App. E.3), this improvement is easily obtained for low $T$ and may be used to quickly estimate MI of a similar magnitude as $\log K$. The Coupled Reverse AIS lower bound on MI requires moderate values of $T$ to match or marginally improve on Independent Multi-Sample AIS, suggesting that the preferred MI lower bound may differ based on the scale of the true MI and amount of available computation.

## 4   MINE-AIS ESTIMATION OF MUTUAL INFORMATION

For settings where the conditional $p(\mathbf{x}|\mathbf{z})$ is unknown, we propose MINE-AIS, which is inspired by MINE but optimizes a tighter lower bound on MI (App. J). Although this bound involves an intractable log partition function, we present a stable, energy-based training scheme and use our Multi-Sample AIS methods from Sec. 3 to evaluate the bound. Consider an flexible, energy-based distribution $\pi_{\theta,\phi}(\mathbf{z}|\mathbf{x})$ as an approximation to the posterior $p(\mathbf{z}|\mathbf{x})$ (Poole et al., 2019; Arbel et al., 2020)

$$\pi_{\theta,\phi}(\mathbf{z}|\mathbf{x}) = \frac{1}{\mathcal{Z}_{\theta,\phi}(\mathbf{x})} q_\theta(\mathbf{z}|\mathbf{x}) e^{T_\phi(\mathbf{x},\mathbf{z})}, \quad \text{where} \quad \mathcal{Z}_{\theta,\phi}(\mathbf{x}) = \mathbb{E}_{q_\theta(\mathbf{z}|\mathbf{x})}\left[e^{T_\phi(\mathbf{x},\mathbf{z})}\right]. \tag{17}$$

Plugging $\pi_{\theta,\phi}(\mathbf{z}|\mathbf{x})$ into the BA lower bound, we denote the resulting bound as the *Implicit Barber-Agakov Lower bound* (IBAL), since it is often difficult to evaluate explicitly due to the intractable log partition function term. After simplifying in App. J.1, we obtain

$$I(\mathbf{x};\mathbf{z}) \geq I_{\text{BA}_L}(\pi_{\theta,\phi}) = \underbrace{\mathbb{E}_{p(\mathbf{x},\mathbf{z})}\left[\log \frac{q_\theta(\mathbf{z}|\mathbf{x})}{p(\mathbf{z})}\right]}_{I_{\text{BA}_L}(q_\theta)} + \underbrace{\mathbb{E}_{p(\mathbf{x},\mathbf{z})}\left[\log \frac{e^{T_\phi(\mathbf{x},\mathbf{z})}}{\mathbb{E}_{q_\theta(\mathbf{z}|\mathbf{x})}\left[e^{T_\phi(\mathbf{x},\mathbf{z})}\right]}\right]}_{\leq \mathbb{E}_{p(\mathbf{x})}[D_{\text{KL}}[p(\mathbf{z}|\mathbf{x})\|q_\theta(\mathbf{z}|\mathbf{x})]]} =: \text{IBAL}(q_\theta, T_\phi), \tag{18}$$

with the gap of the IBAL equal to $\mathbb{E}_{p(\mathbf{x})}[D_{\text{KL}}[p(\mathbf{z}|\mathbf{x}))\|\pi_{\theta,\phi}(\mathbf{z}|\mathbf{x})]]$. Note that the IBAL generalizes the Unnormalized Barber-Agakov bound from Poole et al. (2019), with $I_{\text{UBA}}(T_\phi) = \text{IBAL}(p(\mathbf{z}), T_\phi)$.

**Proposition 4.1.** *For a given $q_\theta(\mathbf{z}|\mathbf{x})$, the optimal IBAL critic function equals the log importance weights up to a constant $T^*(\mathbf{x}, \mathbf{z}) = \log \frac{p(\mathbf{x},\mathbf{z})}{q_\theta(\mathbf{z}|\mathbf{x})} + c(\mathbf{x})$. For this $T^*$, we have $\text{IBAL}(q_\theta, T^*) = I(\mathbf{x};\mathbf{z})$.*

**Relationship with GIWAE**   We can immediately notice similarities between GIWAE and MINE-AIS, including that their optimal energy functions match and that both bounds include a contrastive term which improves upon the BA lower bound. In fact, we prove the following proposition in App. L.2.

**Proposition 4.2.** *For given $q_\theta(\mathbf{z}|\mathbf{x})$ and $T_\phi(\mathbf{x}, \mathbf{z})$, $\lim_{K\to\infty} I_{\text{GIWAE}_L}(q_\theta, T_\phi, K) = \text{IBAL}(q_\theta, T_\phi)$.*

Thus, we may view the IBAL as the limiting behavior of the finite-sample GIWAE bounds as $K \to \infty$. While Cor. 2.3 shows that $I_{\text{GIWAE}_L}(q_\theta, T)$ can improve $I_{\text{BA}_L}(q_\theta)$ by at most $\log K$ nats, we show in App. L.1 that the IBAL contrastive term is flexible enough to close the entire gap in the BA bound. However, this flexible contrastive term comes at the cost of tractability, as the IBAL in Eq. (18) involves a log partition function $\log \mathbb{E}_{q_\theta(\mathbf{z}|\mathbf{x})}[e^{T_\phi(\mathbf{x},\mathbf{z})}]$ compared to the finite-sum term in GIWAE.

**Energy-Based Training of IBAL**   Although the log partition function $\log \mathcal{Z}_{\theta,\phi}(\mathbf{x})$ in the IBAL is intractable to evaluate, we only require an unbiased estimator of its gradient for training. Differentiating Eq. (18) with respect to the parameters $\theta$ and $\phi$, respectively, we obtain

$$\frac{\partial}{\partial\theta}\text{IBAL}(q_\theta, T_\phi) = \mathbb{E}_{p(\mathbf{x},\mathbf{z})}\left[\frac{\partial}{\partial\theta}\log q_\theta(\mathbf{z}|\mathbf{x})\right] - \mathbb{E}_{p(\mathbf{x})\pi_{\theta,\phi}(\mathbf{z}|\mathbf{x})}\left[\frac{\partial}{\partial\theta}\log q_\theta(\mathbf{z}|\mathbf{x})\right], \tag{19}$$

$$\frac{\partial}{\partial\phi}\text{IBAL}(q_\theta, T_\phi) = \mathbb{E}_{p(\mathbf{x},\mathbf{z})}\left[\frac{\partial}{\partial\phi}T_\phi(\mathbf{x},\mathbf{z})\right] - \mathbb{E}_{p(\mathbf{x})\pi_{\theta,\phi}(\mathbf{z}|\mathbf{x})}\left[\frac{\partial}{\partial\phi}T_\phi(\mathbf{x},\mathbf{z})\right]. \tag{20}$$

| Method | Proposal | MNIST-VAE10 | MNIST-VAE100 | MNIST-GAN10 | MNIST-GAN100 | Method | Proposal | CIFAR-GAN10 | CIFAR-GAN100 |
|---|---|---|---|---|---|---|---|---|---|
| AIS (T=1) | $p(\mathbf{z})$ | (0.00, 1929.84) | (0.00, 5830.52) | (0.00, 786.12) | (0.00, 861.38) | AIS T=1 | $p(\mathbf{z})$ | (0.00, 4035635.75) | (0.00, 4853410.50) |
| | $q(\mathbf{z}\vert\mathbf{x})$ | (21.06, 63.00) | (34.49, 362.13) | (3.67, 314.72) | (2.61, 513.33) | | $q(\mathbf{z}\vert\mathbf{x})$ | (17.30, 403679.22) | (20.17, 2378257.50) |
| AIS (T=500) | $p(\mathbf{z})$ | (34.05, 39.09) | (79.90, 95.17) | **(21.57, 22.47)** | **(25.86, 27.55)** | AIS T=500 | $p(\mathbf{z})$ | (29.52, 33089.90) | (104.51, 63290.40) |
| | $q(\mathbf{z}\vert\mathbf{x})$ | **(34.16, 34.29)** | **(80.19, 82.34)** | **(21.60, 23.06)** | (25.58, 29.53) | | $q(\mathbf{z}\vert\mathbf{x})$ | (48.16, 136.15) | (145.19, 2786.53) |
| AIS (T=30K) | $p(\mathbf{z})$ | **(34.21, 34.21)** | **(80.78, 80.84)** | **(21.97, 22.02)** | **(26.47, 26.52)** | AIS T=100K | $p(\mathbf{z})$ | **(71.87, 73.98)** | (480.26, 488.07) |
| | $q(\mathbf{z}\vert\mathbf{x})$ | **(34.21, 34.21)** | **(80.77, 80.80)** | **(22.01, 22.01)** | **(26.53, 26.54)** | | $q(\mathbf{z}\vert\mathbf{x})$ | **(72.85, 73.54)** | (479.27, 484.84) |
| IWAE (K=1) | $p(\mathbf{z})$ | (0.00, 3827.58) | (0.00, 11501.92) | (0.00, 1630.00) | (0.00, 1740.39) | IWAE K=1 | $p(\mathbf{z})$ | (0.00, 7765695.50) | (0.00, 9916102.00) |
| | $q(\mathbf{z}\vert\mathbf{x})$ | (25.20, **35.34**) | (44.54, 95.63) | (4.23, 57.47) | (3.23, 260.87) | | $q(\mathbf{z}\vert\mathbf{x})$ | (17.45, 77.52) | (20.00, 5346.85) |
| IWAE (K=1K) | $p(\mathbf{z})$ | (6.91, 1197.75) | (6.91, 4234.19) | (6.91, 446.80) | (6.91, 494.73) | IWAE K=1K | $p(\mathbf{z})$ | (6.91, 2044170.75) | (6.91, 2856714.50) |
| | $q(\mathbf{z}\vert\mathbf{x})$ | (31.69, **34.24**) | (51.44, 85.30) | (11.14, 52.73) | (10.14, 201.18) | | $q(\mathbf{z}\vert\mathbf{x})$ | (23.58, **74.00**) | (26.98, 5283.13) |
| IWAE (K=1M) | $p(\mathbf{z})$ | (13.82, 376.89) | (13.82, 2247.73) | (13.81, 81.51) | (13.82, 114.01) | IWAE K=1M | $p(\mathbf{z})$ | (13.82, 710511.63) | (13.82, 1903854.50) |
| | $q(\mathbf{z}\vert\mathbf{x})$ | **(34.10, 34.22)** | (58.35, 83.39) | (17.76, 30.88) | (16.98, 58.04) | | $q(\mathbf{z}\vert\mathbf{x})$ | (30.73, **73.36**) | (33.81, 5271.56) |

Table 1: MI Estimation with IWAE (with varying $K$) and Multi-Sample AIS (with varying $T$) on MNIST (left) and CIFAR (right). Tight estimates, with gap of less than 2 nats, are in bold.

Eq. (19) indicates that to maximize the IBAL as a function of $\theta$ and $\phi$, we need to increase the value of $T_\phi(\mathbf{x}, \mathbf{z})$ or $\log q_\theta(\mathbf{z}|\mathbf{x})$ on samples from $p(\mathbf{x}, \mathbf{z})$, and lower it on samples from $p(\mathbf{x})\pi_{\theta,\phi}(\mathbf{z}|\mathbf{x})$. As is common in training energy-based models, it is difficult to draw samples from $\pi_{\theta,\phi}(\mathbf{z}|\mathbf{x})$. To reduce the cost and variance of the estimated gradient, we initialize chains from true posterior sample $\mathbf{z}_0 \sim p(\mathbf{z}|\mathbf{x})$ as in contrastive divergence training (Hinton, 2002) and run $M$ steps of Hamiltonian Monte Carlo (HMC) transition kernels $\mathcal{T}_{1:M}(\mathbf{z}|\mathbf{z}_0, \mathbf{x})$ (Neal, 2011). See App. M.1 for details.

**Multi-Sample AIS Evaluation of IBAL** After training the critic function using the procedure above, we still need to evaluate the IBAL lower bound on MI. We can easily upper bound $\text{IBAL}(q_\theta, T_\phi)$ using a Multi-Sample AIS lower bound on $\log \mathcal{Z}_{\theta,\phi}(\mathbf{x})$, but this does not ensure a lower bound on MI. In order to upper bound $\log \mathcal{Z}_{\theta,\phi}(\mathbf{x})$ and preserve a lower bound on $I(\mathbf{x}; \mathbf{z})$, Multi-Sample AIS requires true samples from $\pi_{\theta,\phi}(\mathbf{z}|\mathbf{x})$ to initialize backward AIS chains. Since these samples are unavailable, we instead initialize backward chains from the true posterior $p(\mathbf{z}|\mathbf{x})$ instead of $\pi_{\theta,\phi}(\mathbf{z}|\mathbf{x})$. We derive sufficient conditions under which this scheme preserves an upper bound on $\log \mathcal{Z}_{\theta,\phi}(\mathbf{x})$ in App. M.2 and provide empirical validation in App. M.3.

# 5 EXPERIMENTS

In this section, we evaluate our proposed MI bounds on VAEs and GANs trained on MNIST and CIFAR.

## 5.1 MULTI-SAMPLE AIS ESTIMATION OF MUTUAL INFORMATION

We compare Multi-Sample AIS MI estimation against IWAE, since both methods assume the full joint distribution is available. For the initial distribution of AIS or variational distribution of IWAE, we experiment using both the prior $p(\mathbf{z})$ and a learned Gaussian $q_\theta(\mathbf{z}|\mathbf{x})$. Table 1 summarizes our results.

**IWAE** As described in Sec. 2.3, IWAE bounds encompass a wide range of MI estimators. The $K = 1$ bounds with learned $q_\theta(\mathbf{z}|\mathbf{x})$ correspond to BA bounds, while for $K > 1$ and $p(\mathbf{z})$ as the proposal, we obtain Structured INFONCE. While the IWAE upper bound on MI, which uses the $\log p(\mathbf{x})$ lower bound with independent sampling from $q_\theta(\mathbf{z}|\mathbf{x})$, is tight for certain models, we can see that the improvement of the IWAE lower bound on MI is limited by $\log K$. In particular, we need exponentially large sample size to close the gap from the BA lower bound ($K = 1$) to the true MI. For example, on CIFAR GAN100, at least $e^{460}$ total samples are required to match the lower bound estimated by AIS.

**Multi-Sample AIS** We evaluate Multi-Sample AIS with $K = 48$ chains on MNIST and $K = 12$ on CIFAR, and a varying number of intermediate $T$. We show results for the Independent Multi-Sample AIS MI lower bound and Coupled Reverse Multi-Sample AIS upper bound in Table 1. Using large enough values of $T$, Multi-Sample AIS can tightly sandwich large values of ground truth MI for all models and datasets considered. This is in stark contrast to the exponential sample complexity required for the IWAE MI lower bound, and highlights that increasing $T$ in Multi-Sample AIS is a practical way to reduce bias using additional computation. We provide runtime details in App. O.1.3, and provide additional results comparing different Multi-Sample AIS bounds in App. I Fig. 5.

## 5.2 ENERGY-BASED ESTIMATION OF MUTUAL INFORMATION

In this section, we evaluate the family of GIWAE and MINE-AIS bounds, which assume access to a known marginal $p(\mathbf{z})$ but not the conditional $p(\mathbf{x}|\mathbf{z})$. We summarize these bounds in Fig. 3b.

**BA, IWAE, and GIWAE Bounds** Recalling that $I_{\text{GIWAE}_L}(q_\theta, T_\phi, K)$ in Eq. (7) consists of the BA lower bound and a contrastive term, we report the contribution of each term in Fig. 3a. For a fixed $q_\theta(\mathbf{z}|\mathbf{x})$, Cor. 2.3 shows $I_{\text{BA}_L}(q_\theta) \leq I_{\text{GIWAE}_L}(q_\theta, T_{\phi^*}, K) \leq I_{\text{IWAE}_L}(q_\theta, K) \leq I_{\text{BA}_L}(q_\theta) + \log K$.

| Input Used | Bound \ Model | Linear VAE10 | MNIST VAE20 | MNIST GAN20 |
|---|---|---|---|---|
| $p(\mathbf{x}, \mathbf{z})$ | Analytical | 23.23 | N/A | N/A |
| $p(\mathbf{x}, \mathbf{z})$ Joint Samples | AIS Bound on True MI | $(23.23, 23.23)$ | $(65.11, 65.17)$ | $(53.43, 53.50)$ |
| | IWAE LB $(K = 1000)$ | $20.53 + 2.66 = 23.19$ | $38.21 + 6.90 = 45.11$ | $20.97 + 6.91 = 27.88$ |
| | IWAE LB $(K = 100)$ | $21.64 + 1.50 = 23.14$ | $38.86 + 4.61 = 43.47$ | $20.86 + 4.60 = 25.46$ |
| | Structured InfoNCE LB $(K = 1000)$ | 6.91 | 6.91 | 6.91 |
| | Structured InfoNCE LB $(K = 100)$ | 4.61 | 4.61 | 4.61 |
| $p(\mathbf{z})$ Joint Samples | AIS Bound on IBAL (MINE-AIS) | $(23.15, 23.15)$ | $(57.72, 57.74)$ | $(40.79, 40.79)$ |
| | AIS Bound on IBAL (GIWAE $K = 100$) | $(22.87, 22.87)$ | $(44.97, 44.97)$ | $(28.61, 28.62)$ |
| | AIS Bound on IBAL (InfoNCE $K = 100$) | $(11.38, 11.39)$ | $(5.18, 5.18)$ | $(7.42, 7.42)$ |
| | Generalized IWAE LB $(K = 1000)$ | $22.31 + 0.38 = 22.69$ | $37.23 + 6.55 = 43.78$ | $20.50 + 6.72 = 27.22$ |
| | Generalized IWAE LB $(K = 100)$ | $22.48 + 0.39 = 22.87$ | $37.56 + 4.34 = 41.90$ | $20.68 + 4.57 = 25.25$ |
| | Barber-Agakov LB $(K = 1)$ | 22.69 | 37.92 | 21.42 |
| Joint Samples | InfoNCE LB $(K = 1000)$ | 6.91 | 6.91 | 6.91 |
| | InfoNCE LB $(K = 100)$ | 4.61 | 4.61 | 4.61 |

(a)

(b)

$I(\mathbf{x}; \mathbf{z})$

$\mathbb{E}\left[D_{\mathrm{KL}}[p(\mathbf{z}|\mathbf{x}) \| \pi_{\theta,\phi}(\mathbf{z}|\mathbf{x})]\right]$

$\mathrm{IBAL}(q_\theta, T_\phi)$

(MINE-AIS)

$\mathbb{E}\left[D_{\mathrm{KL}}[p_{\mathrm{TGT}}^{\mathrm{GIWAE}}(\mathbf{z}^{(1:K)}, s|\mathbf{x}) \| q_{\mathrm{PROP}}^{\mathrm{GIWAE}}(\mathbf{z}^{(1:K)}, s|\mathbf{x})]\right]$

$\mathbb{E}\left[D_{\mathrm{KL}}[p_{\mathrm{TGT}}^{\mathrm{IWAE}}(\mathbf{z}^{(1:K)}|\mathbf{x}) \| q_{\mathrm{PROP}}^{\mathrm{IWAE}}(\mathbf{z}^{(1:K)}|\mathbf{x})]\right]$

$\mathbb{E}\left[D_{\mathrm{KL}}[p(\mathbf{z}|\mathbf{x}) \| q_\theta(\mathbf{z}|\mathbf{x})]\right]$

$\log K$

$I_{\mathrm{IWAE}_L}(q_\theta, K)$

$\mathbb{E}\left[D_{\mathrm{KL}}[p_{\mathrm{TGT}}^{\mathrm{IWAE}}(s|\mathbf{z}^{(1:K)}, \mathbf{x}) \| q_{\mathrm{PROP}}^{\mathrm{GIWAE}}(s|\mathbf{z}^{(1:K)}, \mathbf{x})]\right]$

$I_{\mathrm{GIWAE}_L}(q_\theta, T_\phi, K)$

$\mathbb{E}\left[D_{\mathrm{KL}}[p_{\mathrm{TGT}}^{\mathrm{IWAE}}(s|\mathbf{z}^{(1:K)}, \mathbf{x}) \| \mathcal{U}(s)]\right]$

$I_{\mathrm{BA}_L}(q_\theta)$

Figure 3: (a) Comparison of energy-based bounds (GIWAE and MINE-AIS) with other MI bounds. (b) Visualizing the gaps of various energy-based lower bounds and their relationships.

Although we perform separate optimizations for each entry in Fig. 3a, we find that these relationships hold in almost all cases. We also observe that GIWAE can approach the performance of IWAE, despite the fact that GIWAE uses a learned $T_\phi(\mathbf{x}, \mathbf{z})$ instead of the optimal critic in IWAE (Cor. 2.2).

For the Linear VAE, the true posterior is in the Gaussian variational family and $I_{\mathrm{BA}_L}(q_\theta)$ is close to the analytical MI. In this case, the contrastive term provides much less than $\log K$ improvement for GIWAE and IWAE, since even the optimal critic function cannot distinguish between $q_\theta(\mathbf{z}|\mathbf{x})$ and $p(\mathbf{z}|\mathbf{x})$. As $K$ increases, we learn a worse $q_\theta$ in almost all cases, as measured by a lower BA term. This allows the contrastive term to achieve closer to its full potential $\log K$, resulting in a higher overall bound. For more complex VAE and GAN posteriors, there is a reduced tradeoff between the terms since the variational family is far enough from the true posterior (in reverse KL divergence) that either GIWAE or IWAE critic functions can approach $\log K$ improvement without significantly lowering the BA term. In all cases, (Structured) INFO-NCE bounds saturate to $\log K$.

**MINE-AIS Bounds** We report MINE-AIS results using a fixed Gaussian $p(\mathbf{z})$ as the base variational distribution and Multi-Sample AIS evaluation of $\mathrm{IBAL}(p(\mathbf{z}), T_\phi)$. We can see in Fig. 3a that MINE-AIS improves over BA due to its flexible, energy-based variational family. To evaluate the quality of the learned $T_\phi(\mathbf{x}, \mathbf{z})$, we compare the IBAL to the Multi-Sample AIS lower bound, which assumes access to $p(\mathbf{x}|\mathbf{z})$ and corresponds to the optimal critic in Prop. 4.1. We find that MINE-AIS underestimates the ground truth MI by $11\%$ and $24\%$ on MNIST-VAE and MNIST-GAN, respectively.

We also observe that MINE-AIS outperforms GIWAE (and IWAE) and by $31\%$ and $49\%$ on MNIST-VAE and MNIST-GAN, respectively. To investigate whether this is due to its improved critic function or a more costly evaluation procedure, in Fig. 3a we report Multi-Sample AIS evaluation of $\mathrm{IBAL}(q_\theta, T_\phi)$ for $q_\theta, T_\phi$ learned by optimizing the GIWAE or INFONCE lower bounds with $K = 100$. GIWAE and INFONCE results only marginally improve, indicating that their critic functions are suboptimal and far from the true log importance weights.[2] We conclude that the improvement of MINE-AIS can primarily be attributed to learning a better critic function using energy-based training.

## 6 CONCLUSION

We have provided a unifying view of mutual information estimation from the perspective of importance sampling. We derived probabilistic interpretations of each bound, which shed light on the limitations of existing estimators and motivated our novel GIWAE, Multi-Sample AIS, and MINE-AIS bounds. When the conditional is not known, our GIWAE bounds highlight how variational bounds can complement contrastive learning to improve lower bounds on MI beyond known $\log K$ limitations. When the full joint distribution is known, we show that our Multi-Sample AIS bounds can tightly estimate large values of MI without exponential sample complexity, and thus should be considered the gold standard for MI estimation in these settings. Finally, MINE-AIS extends Multi-Sample AIS evaluation to unknown conditional densities, and can be viewed as the infinite-sample behavior of GIWAE and existing contrastive bounds. Our MINE-AIS and Multi-Sample AIS methods highlight how MCMC techniques can be used to improve MI estimation when a single analytic marginal or conditional density is available.

---

[2]Note that IWAE and Structured INFONCE use the true log importance weights $T^*(\mathbf{x}, \mathbf{z}) = \log \frac{p(\mathbf{x}, \mathbf{z})}{q_\theta(\mathbf{z}|\mathbf{x})} + c(\mathbf{x})$ (Cor. 2.2). For this optimal critic, $\mathrm{IBAL}(q_\theta, T^*) = I(\mathbf{x}; \mathbf{z}), \forall q_\theta$ (Prop. 4.1) and AIS will sandwich the true MI.

## ACKNOWLEDGEMENTS

The authors thank Shengyang Sun, Guodong Zhang, Vaden Masrani, and Umang Gupta for helpful comments on drafts of this work. We also thank the anonymous reviewers whose comments greatly improved the presentation and encouraged us to derive several additional propositions. MG, RG and AM acknowledge support from the Canada CIFAR AI Chairs program.

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

# Supplementary Material

## Table of Contents

# A   A GENERAL APPROACH FOR DERIVING EXTENDED STATE SPACE BOUNDS ON LOG PARTITION FUNCTIONS

In this section, we give a short proof that the gap in our general extended state space bounds from Sec. 2.1 corresponds to a forward or reverse KL divergence. We derive various upper and lower bounds on $\log p(\mathbf{x})$ using this approach throughout the paper and appendix, and we provide a visual summary in Fig. 4.

First, we consider an extended state space target $p_{\text{TGT}}(\mathbf{x}, \mathbf{z}_{\text{ext}})$ and proposal $q_{\text{PROP}}(\mathbf{x}, \mathbf{z}_{\text{ext}})$ distributions. For all cases discussed in this work, we will choose our target and proposal distributions such that $\log \frac{\mathcal{Z}_{\text{TGT}}(\mathbf{x})}{\mathcal{Z}_{\text{PROP}}(\mathbf{x})} = \log p(\mathbf{x})$. For example, a common construction is to have $\mathcal{Z}_{\text{TGT}}(\mathbf{x}) = \int p_{\text{TGT}}(\mathbf{x}, \mathbf{z}_{\text{ext}}) d\mathbf{z}_{\text{ext}} = p(\mathbf{x})$ and $\mathcal{Z}_{\text{PROP}}(\mathbf{x}) = \int q_{\text{PROP}}(\mathbf{x}, \mathbf{z}_{\text{ext}}) d\mathbf{z}_{\text{ext}} = 1$. Our 'reverse' importance sampling bounds App. F-G construct target and proposal such that $\mathcal{Z}_{\text{TGT}}(\mathbf{x}) = p(\mathbf{x})^K$ and $\mathcal{Z}_{\text{PROP}}(\mathbf{x}) = p(\mathbf{x})^{K-1}$, which still yields $\mathcal{Z}_{\text{TGT}}(\mathbf{x})/\mathcal{Z}_{\text{PROP}}(\mathbf{x}) = p(\mathbf{x})$.

Each pair of extended state-space proposal and target distributions provides both an upper and lower bound on the log partition function. Taking the expected log ratio of unnormalized densities under either the proposal or target distribution, we have

$$\mathbb{E}_{q_{\text{PROP}}(\mathbf{z}_{\text{ext}}|\mathbf{x})} \left[ \log \frac{p_{\text{TGT}}(\mathbf{x}, \mathbf{z}_{\text{ext}})}{q_{\text{PROP}}(\mathbf{x}, \mathbf{z}_{\text{ext}})} \right] \leq \log \frac{\mathcal{Z}_{\text{TGT}}(\mathbf{x})}{\mathcal{Z}_{\text{PROP}}(\mathbf{x})} \leq \mathbb{E}_{p_{\text{TGT}}(\mathbf{z}_{\text{ext}}|\mathbf{x})} \left[ \log \frac{p_{\text{TGT}}(\mathbf{x}, \mathbf{z}_{\text{ext}})}{q_{\text{PROP}}(\mathbf{x}, \mathbf{z}_{\text{ext}})} \right]. \qquad (21)$$

To confirm that these are indeed lower and upper bounds for any $q_{\text{PROP}}$ and $p_{\text{TGT}}$, we can show that the gap in the lower bound in Eq. (21) is the forward KL divergence $D_{\text{KL}}[q_{\text{PROP}} \| p_{\text{TGT}}]$, and the gap in the upper bound is the reverse KL divergence, $D_{\text{KL}}[p_{\text{TGT}} \| q_{\text{PROP}}]$

$$\mathbb{E}_{q_{\text{PROP}}(\mathbf{z}_{\text{ext}}|\mathbf{x})} \left[ \frac{p_{\text{TGT}}(\mathbf{x}, \mathbf{z}_{\text{ext}})}{q_{\text{PROP}}(\mathbf{x}, \mathbf{z}_{\text{ext}})} \right] = \underbrace{\log \frac{\mathcal{Z}_{\text{TGT}}(\mathbf{x})}{\mathcal{Z}_{\text{PROP}}(\mathbf{x})} - D_{\text{KL}}[q_{\text{PROP}}(\mathbf{z}_{\text{ext}}|\mathbf{x}) \| p_{\text{TGT}}(\mathbf{z}_{\text{ext}}|\mathbf{x})]}_{\text{ELBO}(\mathbf{x}; q_{\text{PROP}}, p_{\text{TGT}})} \leq \log \frac{\mathcal{Z}_{\text{TGT}}(\mathbf{x})}{\mathcal{Z}_{\text{PROP}}(\mathbf{x})} \qquad (22)$$

$$\mathbb{E}_{p_{\text{TGT}}(\mathbf{z}_{\text{ext}}|\mathbf{x})} \left[ \frac{p_{\text{TGT}}(\mathbf{x}, \mathbf{z}_{\text{ext}})}{q_{\text{PROP}}(\mathbf{x}, \mathbf{z}_{\text{ext}})} \right] = \underbrace{\log \frac{\mathcal{Z}_{\text{TGT}}(\mathbf{x})}{\mathcal{Z}_{\text{PROP}}(\mathbf{x})} + D_{\text{KL}}[p_{\text{TGT}}(\mathbf{z}_{\text{ext}}|\mathbf{x}) \| q_{\text{PROP}}(\mathbf{z}_{\text{ext}}|\mathbf{x})]}_{\text{EUBO}(\mathbf{x}; q_{\text{PROP}}, p_{\text{TGT}})} \geq \log \frac{\mathcal{Z}_{\text{TGT}}(\mathbf{x})}{\mathcal{Z}_{\text{PROP}}(\mathbf{x})}. \qquad (23)$$

Thus, the bounds in Eq. (21) directly generalize the standard ELBO ($\log p(\mathbf{x}) - D_{\mathrm{KL}}[q_\theta(\mathbf{z}|\mathbf{x}) \| p(\mathbf{z}|\mathbf{x})]$) and EUBO ($\log p(\mathbf{x}) + D_{\mathrm{KL}}[p(\mathbf{z}|\mathbf{x}) \| q_\theta(\mathbf{z}|\mathbf{x})]$), which appear as special cases when $K = 1$, $T = 1$, the proposal distribution is $q_{\mathrm{PROP}}(\mathbf{z}|\mathbf{x}) = q_\theta(\mathbf{z}|\mathbf{x})$, and the target distribution is $p_{\mathrm{TGT}}(\mathbf{z}|\mathbf{x}) = p(\mathbf{z}|\mathbf{x})$. In what follows, our extended state space proposal or target distributions may include $q_\theta(\mathbf{z}|\mathbf{x})$ as an initial or base variational distribution, with the posterior $p(\mathbf{z}|\mathbf{x})$ often appearing within target distributions $p_{\mathrm{TGT}}(\mathbf{x}, \mathbf{z}_{\mathrm{ext}})$.

In Fig. 4, we summarize various extended state space proposal (third column) and target distributions (fourth column). We emphasize that the base variational distribution $q_\theta(\mathbf{z}|\mathbf{x})$ (blue circles) and posterior distribution $p(\mathbf{z}|\mathbf{x})$ (red circles) may be used multiple times, in either or both of the extended state space proposal and target distributions. Similarly, forward AIS chains (blue circles) starting from the initial distribution and the backward AIS chains (shown in red circles) starting from the posterior may be used repeatedly in both the proposal or target. In the next sections, we proceed to derive each of the bounds in Fig. 4 as special cases of this general approach, thus interpreting each importance sampling bound in terms of probabilistic inference in an extended state space.

## B  IMPORTANCE WEIGHTED AUTOENCODER (IWAE)

### B.1  PROBABILISTIC INTERPRETATION AND BOUNDS

Consider a $K$-sample proposal distribution $q_{\mathrm{PROP}}^{\mathrm{IWAE}}(\mathbf{z}^{(1:K)})$ using independent draws from an initial distribution $q_\theta(\mathbf{z}|\mathbf{x})$. The target distribution is defined as a uniform mixture of $K$ components, where each component replaces the initial sample $q_\theta(\mathbf{z}^{(k)}|\mathbf{x})$ in index $k$ with a sample from the target distribution $p(\mathbf{z}|\mathbf{x})$.

$$q_{\mathrm{PROP}}^{\mathrm{IWAE}}(\mathbf{z}^{(1:K)}|\mathbf{x}) = \prod_{s=1}^{K} q_\theta(\mathbf{z}^{(s)}|\mathbf{x}), \qquad p_{\mathrm{TGT}}^{\mathrm{IWAE}}(\mathbf{x}, \mathbf{z}^{(1:K)}) = \frac{1}{K} \sum_{s=1}^{K} p(\mathbf{x}, \mathbf{z}^{(s)}) \prod_{\substack{k=1 \\ k \neq s}}^{K} q_\theta(\mathbf{z}^{(s)}|\mathbf{x}).$$

(24)

Note that the normalizing constant of $p_{\mathrm{TGT}}^{\mathrm{IWAE}}(\mathbf{x}, \mathbf{z}^{(1:K)})$, or $\int p_{\mathrm{TGT}}^{\mathrm{IWAE}}(\mathbf{x}, \mathbf{z}^{(1:K)}) d\mathbf{z}^{(1:K)}$, equals $p(\mathbf{x})$ since $q_\theta(\mathbf{z}|\mathbf{x})$ is normalized and $\int p(\mathbf{x}, \mathbf{z}) d\mathbf{z} = p(\mathbf{x})$. To connect this with the IWAE bound, we show that the log importance ratio reduces to

$$\log \frac{p_{\mathrm{TGT}}^{\mathrm{IWAE}}(\mathbf{x}, \mathbf{z}^{(1:K)})}{q_{\mathrm{PROP}}^{\mathrm{IWAE}}(\mathbf{z}^{(1:K)}|\mathbf{x})} = \log \frac{\frac{1}{K} \sum_{s=1}^{K} p(\mathbf{x}, \mathbf{z}^{(s)}) \prod_{\substack{k=1 \\ k \neq s}}^{K} q_\theta(\mathbf{z}^{(k)}|\mathbf{x})}{\prod_{k=1}^{K} q_\theta(\mathbf{z}^{(k)}|\mathbf{x})}$$

$$= \log \frac{1}{K} \sum_{k=1}^{K} \frac{p(\mathbf{x}, \mathbf{z}^{(k)})}{q_\theta(\mathbf{z}^{(k)}|\mathbf{x})}.$$

(25)

As in Eq. (21), we can obtain lower and upper bounds on $\log p(\mathbf{x})$ by taking expectations under the proposal and target distributions, respectively.

**Alternative Probabilistic Interpretation**   We now present an alternative probabilistic interpretation of IWAE, which is similar to Domke & Sheldon (2018) and will be used as the foundation for our GIWAE bounds in App. C. Consider the following extended state space target distribution,

$$p_{\mathrm{TGT}}^{\mathrm{IWAE}}(\mathbf{x}, \mathbf{z}^{(1:K)}, s) = \frac{1}{K} p(\mathbf{x}, \mathbf{z}^{(s)}) \prod_{\substack{k=1 \\ k \neq s}}^{K} q(\mathbf{z}^{(k)}|\mathbf{x}).$$

(26)

Note that marginalization over $s$ leads to the IWAE mixture target in Eq. (4) or Eq. (24). We consider the extended state space proposal

$$q_{\mathrm{PROP}}^{\mathrm{IWAE}}(\mathbf{z}^{(1:K)}, s|\mathbf{x}) = \left( \prod_{k=1}^{K} q_\theta(\mathbf{z}^{(k)}|\mathbf{x}) \right) q_{\mathrm{PROP}}^{\mathrm{IWAE}}(s|\mathbf{z}^{(1:K)}, \mathbf{x}),$$

(27)

Figure 4: Comparison of the probabilistic extended state-space interpretations of different multi-sample bounds. Forward chains of AIS and variational distributions in IS / IWAE are colored in blue. Backward chains in AIS or posterior distributions in IS / IWAE are colored in red. Note that a lower bound on $\log p(\mathbf{x})$, translates to an upper bound on MI, and vice versa.

where we have defined $q_{\text{PROP}}^{\text{IWAE}}(s|\mathbf{z}^{(1:K)}, \mathbf{x}) = \frac{p(\mathbf{x}, \mathbf{z}^{(s)})}{q_\theta(\mathbf{z}^{(s)}|\mathbf{x})} / \sum_{k=1}^{K} \frac{p(\mathbf{x}, \mathbf{z}^{(k)})}{q_\theta(\mathbf{z}^{(k)}|\mathbf{x})}$.

As desired, the log importance weight match Eq. (25)

$$\log \frac{p_{\text{TGT}}^{\text{IWAE}}(\mathbf{x}, \mathbf{z}^{(1:K)}, s)}{q_{\text{PROP}}^{\text{IWAE}}(\mathbf{z}^{(1:K)}, s|\mathbf{x})} = \log \frac{\frac{1}{K} p(\mathbf{x}, \mathbf{z}^{(s)}) \prod_{\substack{k=1 \\ k \neq s}}^{K} q_\theta(\mathbf{z}^{(k)}|\mathbf{x})}{\prod_{k=1}^{K} q_\theta(\mathbf{z}^{(k)}|\mathbf{x}) \frac{\frac{p(\mathbf{x}, \mathbf{z}^{(s)})}{q_\theta(\mathbf{z}^{(s)}|\mathbf{x})}}{\sum\limits_{k=1}^{K} \frac{p(\mathbf{x}, \mathbf{z}^{(k)})}{q_\theta(\mathbf{z}^{(k)}|\mathbf{x})}}} = \log \frac{1}{K} \sum_{k=1}^{K} \frac{p(\mathbf{x}, \mathbf{z}^{(k)})}{q_\theta(\mathbf{z}^{(k)}|\mathbf{x})} . \qquad (28)$$

**Lower Bound on** $\log p(\mathbf{x})$ **and Upper Bound on MI**    Using the general approach in App. A, taking expectations under $q_{\text{PROP}}^{\text{IWAE}}$ leads to a lower bound on $\log p(\mathbf{x})$

$$\text{ELBO}_{\text{IWAE}}(\mathbf{x}; q_\theta, K) := \mathbb{E}_{\prod_{k=1}^{K} q_\theta(\mathbf{z}^{(k)}|\mathbf{x})} \left[ \log \frac{1}{K} \sum_{k=1}^{K} \frac{p(\mathbf{x}, \mathbf{z}^{(k)})}{q_\theta(\mathbf{z}^{(k)}|\mathbf{x})} \right]. \tag{29}$$

This corresponds to the following upper bound on MI for known $p(\mathbf{x}|\mathbf{z})$

$$I(\mathbf{x}; \mathbf{z}) \leq I_{\text{IWAE}_U}(q_\theta, K) = \mathbb{E}_{p(\mathbf{x},\mathbf{z})}[\log p(\mathbf{x}|\mathbf{z})] - \mathbb{E}_{p(\mathbf{x})}[\text{ELBO}_{\text{IWAE}}(\mathbf{x}; q_\theta, K)] \tag{30}$$

$$= \mathbb{E}_{p(\mathbf{x},\mathbf{z})}[\log p(\mathbf{x}|\mathbf{z})] - \mathbb{E}_{p(\mathbf{x}) \prod_{k=1}^{K} q_\theta(\mathbf{z}^{(k)}|\mathbf{x})} \left[ \log \frac{1}{K} \sum_{k=1}^{K} \frac{p(\mathbf{x}, \mathbf{z}^{(k)})}{q_\theta(\mathbf{z}^{(k)}|\mathbf{x})} \right] \tag{31}$$

**Upper Bound on** $\log p(\mathbf{x})$ **and Lower Bound on MI**    Similarly, with expectations under $p_{\text{TGT}}^{\text{IWAE}}(\mathbf{z}^{(1:K)}|\mathbf{x})$, we obtain an upper bound on $\log p(\mathbf{x})$. Since, for independent draws, the uniform mixture $p_{\text{TGT}}^{\text{IWAE}}$ is invariant to index permutations, we may choose the target sample to be $\mathbf{z}^{(1)}$ and obtain the upper bound of Sobolev & Vetrov (2019):

$$\text{EUBO}_{\text{IWAE}}(\mathbf{x}; q_\theta, K) := \mathbb{E}_{p(\mathbf{z}^{(1)}|\mathbf{x}) \prod_{k=2}^{K} q_\theta(\mathbf{z}^{(k)}|\mathbf{x})} \left[ \log \frac{1}{K} \sum_{i=1}^{K} \frac{p(\mathbf{x}, \mathbf{z}^{(k)})}{q(\mathbf{z}^{(k)}|\mathbf{x})} \right] \geq \log p(\mathbf{x}). \tag{32}$$

Translating this to a lower bound on MI with known $p(\mathbf{x}|\mathbf{z})$,

$$I(\mathbf{x}; \mathbf{z}) \geq I_{\text{IWAE}_L}(q_\theta, K) = \mathbb{E}_{p(\mathbf{x},\mathbf{z})}[\log p(\mathbf{x}|\mathbf{z})] - \mathbb{E}_{p(\mathbf{x})}[\text{EUBO}_{\text{IWAE}}(\mathbf{x}; q_\theta, K)] \tag{33}$$

$$= \mathbb{E}_{p(\mathbf{x},\mathbf{z})}[\log p(\mathbf{x}|\mathbf{z})] - \mathbb{E}_{p(\mathbf{x})p(\mathbf{z}^{(1)}|\mathbf{x}) \prod_{k=2}^{K} q_\theta(\mathbf{z}^{(k)}|\mathbf{x})} \left[ \log \frac{1}{K} \sum_{k=1}^{K} \frac{p(\mathbf{x}, \mathbf{z}^{(k)})}{q_\theta(\mathbf{z}^{(k)}|\mathbf{x})} \right]. \tag{34}$$

### B.2    PROOF OF LOGARITHMIC IMPROVEMENT IN $K$ FOR IWAE EUBO

We first recall results that IWAE bounds tighten with increasing $K$.

**Proposition B.1.** *For given* $q_\theta(\mathbf{z}|\mathbf{x})$, $\text{ELBO}_{\text{IWAE}}(\mathbf{x}; q_\theta, K+1) \geq \text{ELBO}_{\text{IWAE}}(\mathbf{x}; q_\theta, K)$ *(Burda et al., 2016) and* $\text{EUBO}_{\text{IWAE}}(\mathbf{x}; q_\theta, K+1) \leq \text{EUBO}_{\text{IWAE}}(\mathbf{x}; q_\theta, K)$ *(Sobolev & Vetrov, 2019).*

**Proposition B.2** (Improvement of IWAE and ELBO or EUBO)**.** *Let* $p_{\text{TGT}}^{\text{IWAE}}(s|\mathbf{x}, \mathbf{z}^{(1:K)}) = \frac{p(\mathbf{x},\mathbf{z}^{(s)})}{q_\theta(\mathbf{z}^{(s)}|\mathbf{x})} / \sum_{k=1}^{K} \frac{p(\mathbf{x},\mathbf{z}^{(k)})}{q_\theta(\mathbf{z}^{(k)}|\mathbf{x})}$ *denote the normalized importance weights and* $\mathcal{U}(s)$ *indicate the uniform distribution over* $K$ *discrete values. Then, we can characterize the improvement of* $\text{ELBO}_{\text{IWAE}}(\mathbf{x}; q_\theta, K)$ *and* $\text{EUBO}_{\text{IWAE}}(\mathbf{x}; q_\theta, K)$ *over* $\text{ELBO}(\mathbf{x}; q_\theta)$ *and* $\text{EUBO}(\mathbf{x}; q_\theta)$ *using* KL *divergences, as follows*

$$\text{ELBO}_{\text{IWAE}}(\mathbf{x}; q_\theta, K) = \text{ELBO}(\mathbf{x}; q_\theta) + \underbrace{\mathbb{E}_{q_{\text{PROP}}^{\text{IWAE}}(\mathbf{z}^{(1:K)}|\mathbf{x})} \left[ D_{\text{KL}}[\mathcal{U}(s) \| p_{\text{TGT}}^{\text{IWAE}}(s|\mathbf{z}^{(1:K)}, \mathbf{x})] \right]}_{0 \leq \text{KL of uniform from SNIS weights} \leq D_{\text{KL}}[q_\theta(\mathbf{z}|\mathbf{x}) \| p(\mathbf{z}|\mathbf{x})]}, \tag{35}$$

$$\text{EUBO}_{\text{IWAE}}(\mathbf{x}; q_\theta, K) = \text{EUBO}(\mathbf{x}; q_\theta) - \underbrace{\mathbb{E}_{p_{\text{TGT}}^{\text{IWAE}}(\mathbf{z}^{(1:K)}|\mathbf{x})} \left[ D_{\text{KL}}[p_{\text{TGT}}^{\text{IWAE}}(s|\mathbf{z}^{(1:K)}, \mathbf{x}) \| \mathcal{U}(s)] \right]}_{0 \leq \text{KL of SNIS weights from uniform} \leq \log K}. \tag{36}$$

Prop. B.2 demonstrates that the improvement of the IWAE log partition function bounds over its single-sample counterparts is larger for more non-uniform SNIS weights. Notably, the improvement of $\text{EUBO}_{\text{IWAE}}(\mathbf{x}; q_\theta, K)$ over the single-sample $\text{EUBO}(\mathbf{x}; q_\theta)$ is limited by $\log K$.

*Proof.* We first note that the single-sample $\text{ELBO}(\mathbf{x}; q_\theta)$ and $\text{EUBO}(\mathbf{x}; q_\theta)$ are special cases of $\text{ELBO}_{\text{GIWAE}}(\mathbf{x}; q_\theta, T_\phi, K)$ and $\text{EUBO}_{\text{GIWAE}}(\mathbf{x}; q_\theta, T_\phi, K)$ (Lemma C.1) with $T_\phi = \text{const}$. As a result, the gap between $\text{ELBO}(\mathbf{x}; q_\theta)$ and $\text{ELBO}_{\text{IWAE}}(\mathbf{x}; q_\theta, K)$, for example, follows as a special case of the gap between $\text{ELBO}_{\text{GIWAE}}(\mathbf{x}; q_\theta, T_\phi, K)$ and $\text{ELBO}_{\text{IWAE}}(\mathbf{x}; q_\theta, K)$, which we characterize in Lemma C.2 (App. C.4). The result in Prop. B.2 follows directly.

We now justify the range of the KL divergences in Eq. (35) and Eq. (36) referenced in the underbraces.

*Improvement of* $\text{EUBO}_{\text{IWAE}}$: Note $D_{\text{KL}}[p_{\text{TGT}}^{\text{IWAE}}(s|\mathbf{z}^{(1:K)},\mathbf{x})\|\mathcal{U}(s)] = \log K - H(p_{\text{TGT}}^{\text{IWAE}}(s|\mathbf{z}^{(1:K)},\mathbf{x}))$ is bounded above by $\log K$ since the entropy of a discrete random variable is nonnegative.

Thus, the improvement of $\text{EUBO}_{\text{IWAE}}(\mathbf{x}; q_\theta, K)$ over $\text{EUBO}(\mathbf{x}; q_\theta)$ is limited to $\log K$. We prove similar results in Prop. 2.1 and Cor. 2.3 (App. C.4), Prop. E.2 (App. E.3) and Prop. H.2 (App. H.3).

*Improvement of* $\text{ELBO}_{\text{IWAE}}$: On the other hand, the KL divergence $D_{\text{KL}}[\mathcal{U}(s)\|p_{\text{TGT}}^{\text{GIWAE}}(s|\mathbf{z}^{(1:K)},\mathbf{x})]$ is not limited by $\log K$. However, we do know that the improvement of $\text{ELBO}_{\text{IWAE}}(\mathbf{x}; q_\theta, K)$ over $\text{ELBO}(\mathbf{x}; q_\theta)$ will be limited by $D_{\text{KL}}[q_\theta(\mathbf{z}|\mathbf{x})\|p(\mathbf{z}|\mathbf{x})]$, the gap of $\text{ELBO}(\mathbf{x}; q_\theta)$.

$\square$

Prop. B.2 shows that both IWAE upper and lower bounds on $\log p(\mathbf{x})$ improve upon their single-sample counterparts, with the improvement of $\text{EUBO}_{\text{IWAE}}(\mathbf{x}; q_\theta, K)$ over $\text{EUBO}(\mathbf{x}; q_\theta)$ limited by $\log K$. Cor. B.3 follows directly by translating these results to bounds on MI.

**Corollary B.3.** IWAE *bounds on* MI *improve upon the* BA *bounds with the following relationships:*

$$I_{\text{BA}_L}(q_\theta) \leq I_{\text{IWAE}_L}(q_\theta, K) \leq I_{\text{BA}_L}(q_\theta) + \log K, \qquad I_{\text{IWAE}_U}(q_\theta, K) \leq I_{\text{BA}_U}(q_\theta). \qquad (37)$$

*Proof.* Recall from Sec. 1.1 and Sec. 2.2 that $I_{\text{BA}_L}(q_\theta) = \mathbb{E}_{p(\mathbf{x},\mathbf{z})}[\log p(\mathbf{x}|\mathbf{z})] - \mathbb{E}_{p(\mathbf{x})}[\text{EUBO}(\mathbf{x}; q_\theta)]$, and $I_{\text{IWAE}_L}(q_\theta, K) = \mathbb{E}_{p(\mathbf{x},\mathbf{z})}[\log p(\mathbf{x}|\mathbf{z})] - \mathbb{E}_{p(\mathbf{x})}[\text{EUBO}_{\text{IWAE}}(\mathbf{x}; q_\theta, K)]$. Using Prop. B.2, the fact that $\text{EUBO}(\mathbf{x}; q_\theta) - \log K \leq \text{EUBO}_{\text{IWAE}}(\mathbf{x}; q_\theta, K)$ for any $\mathbf{x}$ implies that

$$I_{\text{IWAE}_L}(q_\theta, K) - I_{\text{BA}_L}(q_\theta) = \mathbb{E}_{p(\mathbf{x})}[\text{EUBO}(\mathbf{x}; q_\theta) - \text{EUBO}_{\text{IWAE}}(\mathbf{x}; q_\theta, K)] \leq \log K, \qquad (38)$$

which results in $I_{\text{IWAE}_L}(q_\theta, K) \leq I_{\text{BA}_L}(q_\theta) + \log K$, as desired.

$I_{\text{BA}_L}(q_\theta) \leq I_{\text{IWAE}_L}(q_\theta, K)$ and $I_{\text{IWAE}_U}(q_\theta, K) \leq I_{\text{BA}_U}(q_\theta)$ follow from the fact that IWAE bounds tighten with increasing $K$ in Prop. B.1. $\square$

### B.3 EXPERIMENTAL RESULTS SHOWING LOGARITHMIC IMPROVEMENT FOR $I_{\text{IWAE}_L}(q_\theta, K)$

In Sec. 2.4, we showed that IWAE is a special case of GIWAE, which decomposes into the sum of a variational BA lower bound and a $K$-sample contrastive term. This suggests that the IWAE lower bound on MI, which arises from an upper bound on $\log p(\mathbf{x})$, may be written as

$$I_{\text{IWAE}_L}(q_\theta, K) = \underbrace{\mathbb{E}_{p(\mathbf{x},\mathbf{z})}\left[\log \frac{q_\theta(\mathbf{z}|\mathbf{x})}{p(\mathbf{z})}\right]}_{I_{\text{BA}}(q)} + \underbrace{\mathbb{E}_{p(\mathbf{x})p(\mathbf{z}^{(1)}|\mathbf{x})\prod_{k=2}^K q_\theta(\mathbf{z}^{(k)}|\mathbf{x})}\left[\log \frac{\frac{p(\mathbf{x},\mathbf{z}^{(1)})}{q_\theta(\mathbf{z}^{(1)}|\mathbf{x})}}{\frac{1}{K}\sum_{i=1}^K \frac{p(\mathbf{x},\mathbf{z}^{(k)})}{q_\theta(\mathbf{z}^{(k)}|\mathbf{x})}}\right]}_{0 \leq \text{ contrastive term } \leq \log K}. \qquad (39)$$

This way of writing $I_{\text{IWAE}_L}(q_\theta, K)$ provides additional intuition for the result in Prop. B.2 and Cor. B.3. In particular, the improvement of $I_{\text{IWAE}_L}(q_\theta, K)$ over $I_{\text{BA}_L}(q_\theta)$ is simply the contrastive term, which is limited to $\log K$.

| Method | $\log K$ | Proposal | VAE2 | VAE10 | VAE100 | GAN2 | GAN10 | GAN100 |
|---|---|---|---|---|---|---|---|---|
| IWAE (K=1) | 0 | $p(\mathbf{z})$ | $0+0=0$ | $0+0=0$ | $0+0=0$ | $0+0=0$ | $0+0=0$ | $0+0=0$ |
| | | $q(\mathbf{z}|\mathbf{x})$ | $8.63+0=8.63$ | $25.20+0=25.20$ | $44.54+0=44.54$ | $8.83+0=8.83$ | $4.23+0=4.23$ | $3.23+0=3.23$ |
| IWAE (K=1K) | 6.91 | $p(\mathbf{z})$ | $0+6.81=6.81$ | $0+6.91=6.91$ | $0+6.91=6.91$ | $0+6.88=6.88$ | $0+6.91=6.91$ | $0+6.91=6.91$ |
| | | $q(\mathbf{z}|\mathbf{x})$ | $7.29+1.8=9.09$ | $25.20+6.49=31.69$ | $44.54+6.90=51.44$ | $7.82+2.92=10.74$ | $4.23+6.91=11.14$ | $3.23+6.91=10.14$ |
| IWAE (K=1M) | 13.82 | $p(\mathbf{z})$ | $0+9.09=9.09$ | $0+13.82=13.82$ | $0+13.82=13.82$ | $0+10.76=10.76$ | $0+13.81=13.81$ | $0+13.82=13.82$ |
| | | $q(\mathbf{z}|\mathbf{x})$ | $3.78+5.31=9.09$ | $25.20+8.90=34.10$ | $44.54+13.81=58.35$ | $6.02+4.79=10.81$ | $4.23+13.53=17.76$ | $3.17+13.81=16.98$ |

Table 2: Decomposition of $I_{\text{IWAE}_L}(q_\theta, K)$ into BA term and contrastive term ($< \log K$) on MI estimation for VAE and GAN models trained on MNIST.

Table 2 and Table 3 show the IWAE objective decomposition to the BA term and the contrastive term, on VAEs and GANs trained MNIST and CIFAR-10 dataset. We can see that in all the experiments, the contribution of the contrastive term is always less than or equal to $\log K$.

For MNIST VAEs and GANs with two dimensional latent spaces, the contrastive term may contribute notably less than $\log K$. In these cases, even the optimal critic function $T^*(\mathbf{x}, \mathbf{z}) = \log \frac{p(\mathbf{x},\mathbf{z})}{q_\theta(\mathbf{z}|\mathbf{x})} + c(\mathbf{x})$, as used in IWAE, has difficulty distinguishing posterior and variational samples.

| Method | $\log K$ | Proposal | GAN5 | GAN10 | GAN100 |
|---|---|---|---|---|---|
| IWAE K=1 | 0 | $p(\mathbf{z})$ | $0 + 0 = 0$ | $0 + 0 = 0$ | $0 + 0 = 0$ |
| | | $q(\mathbf{z}|\mathbf{x})$ | $14.53 + 0 = 14.53$ | $17.45 + 0 = 17.45$ | $20.00 + 0 = 20.00$ |
| IWAE K=1k | 6.91 | $p(\mathbf{z})$ | $0 + 6.91 = 6.91$ | $0 + 6.91 = 6.91$ | $0 + 6.91 = 6.91$ |
| | | $q(\mathbf{z}|\mathbf{x})$ | $14.53 + 6.90 = 21.43$ | $16.68 + 6.9 = 23.58$ | $20.07 + 6.91 = 26.98$ |
| IWAE K=1M | 13.82 | $p(\mathbf{z})$ | $0 + 13.82 = 13.82$ | $0 + 13.82 = 13.82$ | $0 + 13.82 = 13.82$ |
| | | $q(\mathbf{z}|\mathbf{x})$ | $14.53 + 13.81 = 28.34$ | $16.92 + 13.81 = 30.73$ | $20.00 + 13.81 = 33.81$ |

Table 3: Decomposition of $I_{\text{IWAE}_L}(q_\theta, K)$ into BA term and contrastive term ($< \log K$) on MI estimation for GAN models trained on CIFAR-10.

However, the contribution of the contrastive term is almost exactly $\log K$ for higher dimensional VAE and GAN models, where the posterior $p(\mathbf{z}|\mathbf{x})$ is more complex and is more easily distinguishable from the variational $q_\theta(\mathbf{z}|\mathbf{x})$. These results highlight the inherent exponential sample complexity of the IWAE lower bound on MI.

## B.4 BIAS REDUCTION IN $K$ FOR IWAE LOWER BOUND ON $\log p(\mathbf{x})$ / UPPER BOUND ON MI

We have seen in App. B.2 Prop. B.2 that the improvement of $\text{EUBO}_{\text{IWAE}}(\mathbf{x}; q_\theta, K)$ and $I_{\text{IWAE}_L}(\mathbf{x}; q_\theta, K)$ over the single-sample $\text{EUBO}(\mathbf{x}; q_\theta)$ and $I_{\text{BA}_L}(\mathbf{x}; q_\theta)$ is limited by $\log K$. This suggests that exponential sample complexity in the gap of the EUBO, $K \propto \exp\{D_{\text{KL}}[p(\mathbf{z}|\mathbf{x})\|q_\theta(\mathbf{z}|\mathbf{x})]\}$, is required to obtain a tight lower bound.

The quantity $D_{\text{KL}}[\mathcal{U}(s)\|p_{\text{TGT}}^{\text{IWAE}}(s|\mathbf{z}^{(1:K)}, \mathbf{x})]$, which measures the improvement of the IWAE lower bound on $\log p(\mathbf{x})$ over the ELBO, is not explicitly limited by $\log K$. However, Chatterjee et al. (2018) suggest that the same exponential sample complexity, $K \propto \exp\{D_{\text{KL}}[p(\mathbf{z}|\mathbf{x})\|q_\theta(\mathbf{z}|\mathbf{x})]\}$, is required for accurate importance sampling estimation with proposal $q_\theta(\mathbf{z}|\mathbf{x})$. Maddison et al. (2017); Domke & Sheldon (2018) find that the bias of the IWAE lower bound in the limit of $K \to \infty$ reduces at the rate of $\mathcal{O}(\frac{1}{2K}\text{Var}[\frac{p(\mathbf{z}|\mathbf{x})}{q_\theta(\mathbf{z}|\mathbf{x})}])$, although the $\text{Var}[\frac{p(\mathbf{x},\mathbf{z})}{q_\theta(\mathbf{z}|\mathbf{x})}]$ term is at least exponential in $D_{\text{KL}}[p(\mathbf{z}|\mathbf{x})\|q_\theta(\mathbf{z}|\mathbf{x})]$ (Song & Ermon, 2019).

This exponential sample complexity for exact estimation of $\log p(\mathbf{x})$ or $I(\mathbf{x}, \mathbf{z})$ is usually impractical for complex target distributions and limited variational families, where $D_{\text{KL}}[p(\mathbf{z}|\mathbf{x})\|q_\theta(\mathbf{z}|\mathbf{x})]$ may be large. This motivates our improved, multi-sample AIS proposals in Sec. 3.2, which achieve more favorable (linear) bias reduction by introducing MCMC transition kernels to bridge between $q_\theta(\mathbf{z}|\mathbf{x})$ and $p(\mathbf{z}|\mathbf{x})$.

## B.5 RELATIONSHIP WITH STRUCTURED INFONCE

We can recognize the Structured INFONCE upper and lower bounds for known $p(\mathbf{x}|\mathbf{z})$ (Poole et al. (2019) Sec. 2.5) as simply applying the standard IWAE bounds, using the marginal $p(\mathbf{z})$ in place of the variational $q_\theta(\mathbf{z}|\mathbf{x})$

$$\mathbb{E}_{p(\mathbf{x})p(\mathbf{z}^{(1)}|\mathbf{x})\prod_{k=2}^{K}p(\mathbf{z}^{(k)})}\left[\log \frac{p(\mathbf{x}|\mathbf{z}^{(1)})}{\frac{1}{K}\sum_{k=1}^{K}p(\mathbf{x}|\mathbf{z}^{(k)})}\right] \leq I(\mathbf{x};\mathbf{z}) \leq \mathbb{E}_{p(\mathbf{x})\prod_{k=1}^{K}p(\mathbf{z}^{(k)})}\left[\log \frac{1}{\frac{1}{K}\sum_{k=1}^{K}p(\mathbf{x}|\mathbf{z}^{(k)})}\right] - H(\mathbf{x}|\mathbf{z}).$$

We refer to the lower bound as $I_{\text{S-INFONCE}_L}(K)$ and the upper bound as $I_{\text{S-INFONCE}_U}(K)$. From Cor. B.3, we obtain an alternative proof that the Structured INFONCE lower bound is upper bounded by $\log K$. Since $I_{\text{BA}_L}(p(\mathbf{z})) \leq I_{\text{IWAE}_L}(p(\mathbf{z}), K) \leq I_{\text{BA}_L}(p(\mathbf{z})) + \log K$ and the BA bound with a prior proposal equals 0 from Eq. (3), we have that $0 \leq I_{\text{S-INFONCE}_L}(K) \leq \log K$.

## C GENERALIZED IWAE

### C.1 PROBABILISTIC INTERPRETATION AND BOUNDS

To derive a probabilistic interpretation for GIWAE, our starting point is to further extend the state space of the IWAE target distribution in Eq. (4), using a uniform index variable $p(s) = \frac{1}{K} \forall s$ that specifies which sample $\mathbf{z}^{(k)}$ is drawn from the posterior $p(\mathbf{z}|\mathbf{x})$. This is shown in Fig. 4, and leads to

a joint distribution over $(\mathbf{x}, \mathbf{z}^{(1:K)}, s)$ as

$$p_{\text{TGT}}^{\text{GIWAE}}(\mathbf{x}, \mathbf{z}^{(1:K)}, s) = \frac{1}{K} p(\mathbf{x}, \mathbf{z}^{(s)}) \prod_{\substack{k=1 \\ k \neq s}}^{K} q(\mathbf{z}^{(k)}|\mathbf{x}), \tag{40}$$

with marginalization over $s$ leading to the mixture in Eq. (4). The posterior over the index variable $s$, which infers the 'positive' sample drawn from $p(\mathbf{z}|\mathbf{x})$ given a set of samples $\mathbf{z}^{(1:K)}$, corresponds to the normalized importance weights

$$p_{\text{TGT}}^{\text{GIWAE}}(s|\mathbf{x}, \mathbf{z}^{(1:K)}) = \frac{\frac{p(\mathbf{x}, \mathbf{z}^{(s)})}{q_\theta(\mathbf{z}^{(s)}|\mathbf{x})}}{\sum_{k=1}^{K} \frac{p(\mathbf{x}, \mathbf{z}^{(k)})}{q_\theta(\mathbf{z}^{(k)}|\mathbf{x})}}, \tag{41}$$

which can be derived from $p_{\text{TGT}}^{\text{GIWAE}}(s|\mathbf{x}, \mathbf{z}^{(1:K)}) = \frac{p_{\text{TGT}}^{\text{GIWAE}}(\mathbf{x}, \mathbf{z}^{(1:K)}, s)}{p_{\text{TGT}}^{\text{GIWAE}}(\mathbf{x}, \mathbf{z}^{(1:K)})} = \frac{\frac{1}{K} p(\mathbf{x}, \mathbf{z}^{(s)}) \prod_{k \neq s} q_\phi(\mathbf{z}^{(k)}|\mathbf{x})}{\frac{1}{K} \sum_{j=1}^{K} p(\mathbf{x}, \mathbf{z}^{(j)}) \prod_{k \neq j} q_\phi(\mathbf{z}^{(k)}|\mathbf{x})}$.

For the GIWAE extended state space proposal distribution, we consider a categorical index variable $q_{\text{PROP}}^{\text{GIWAE}}(s|\mathbf{z}^{(1:K)}, \mathbf{x})$ drawn according SNIS, with weights calculated using the critic function $T_\phi$.

$$q_{\text{PROP}}^{\text{GIWAE}}(\mathbf{z}^{(1:K)}, s|\mathbf{x}) = \left( \prod_{k=1}^{K} q_\theta(\mathbf{z}^{(k)}|\mathbf{x}) \right) q_{\text{PROP}}^{\text{GIWAE}}(s|\mathbf{z}^{(1:K)}, \mathbf{x}), \tag{42}$$

$$\text{where} \quad q_{\text{PROP}}^{\text{GIWAE}}(s|\mathbf{z}^{(1:K)}, \mathbf{x}) = \frac{e^{T_\phi(\mathbf{x}, \mathbf{z}^{(s)})}}{\sum_{k=1}^{K} e^{T_\phi(\mathbf{x}, \mathbf{z}^{(k)})}}. \tag{43}$$

Similarly to Lawson et al. (2019), the variational SNIS distribution $q_{\text{PROP}}^{\text{GIWAE}}(s|\mathbf{z}^{(1:K)}, \mathbf{x})$ is approximating the true SNIS distribution $p_{\text{TGT}}^{\text{GIWAE}}(s|\mathbf{x}, \mathbf{z}^{(1:K)})$. As we show in App. C.4, the optimal critic function is $T_\phi(\mathbf{x}, \mathbf{z}) = \log \frac{p(\mathbf{x}, \mathbf{z})}{q(\mathbf{z}|\mathbf{x})} + c(\mathbf{x})$, which recovers the IWAE probabilistic interpretation (see App. B.1).

**Log Importance Ratio**  To derive bounds on the log partition function, we first calculate the log unnormalized density ratio

$$\log \frac{p_{\text{TGT}}^{\text{GIWAE}}(\mathbf{z}^{(1:K)}, s, \mathbf{x})}{q_{\text{PROP}}^{\text{GIWAE}}(\mathbf{z}^{(1:K)}, s|\mathbf{x})} = \log \frac{\frac{1}{K} p(\mathbf{x}, \mathbf{z}^{(s)}) \prod_{\substack{k=1 \\ k \neq s}}^{K} q(\mathbf{z}^{(k)}|\mathbf{x})}{q(s|\mathbf{z}^{(1:K)}, \mathbf{x}) \prod_{k=1}^{K} q(\mathbf{z}^{(k)}|\mathbf{x})} \tag{44}$$

$$= \log \frac{1}{K} \frac{\sum_k e^{T(\mathbf{x}, \mathbf{z}^{(k)})}}{e^{T(\mathbf{x}, \mathbf{z}^{(s)})}} \frac{p(\mathbf{x}, \mathbf{z}^{(s)})}{q(\mathbf{z}^{(s)}|\mathbf{x})} \tag{45}$$

$$= \log \frac{p(\mathbf{x}, \mathbf{z}^{(s)})}{q(\mathbf{z}^{(s)}|\mathbf{x})} - T(\mathbf{x}, \mathbf{z}^{(s)}) + \log \frac{1}{K} \sum_{k=1}^{K} e^{T(\mathbf{x}, \mathbf{z}^{(k)})}. \tag{46}$$

Taking the expectation of the log unnormalized density ratio under the proposal or target distribution yields a lower or upper bound, respectively, on $\log p(\mathbf{x})$

$$\underbrace{\mathbb{E}_{q_{\text{PROP}}^{\text{GIWAE}}} \left[ \log \frac{p_{\text{TGT}}^{\text{GIWAE}}(\mathbf{z}^{(1:K)}, s, \mathbf{x})}{q_{\text{PROP}}^{\text{GIWAE}}(\mathbf{z}^{(1:K)}, s|\mathbf{x})} \right]}_{\text{ELBO}_{\text{GIWAE}}(\mathbf{x}; q_\theta, T_\phi, K)} \leq \log p(\mathbf{x}) \leq \underbrace{\mathbb{E}_{p_{\text{TGT}}^{\text{GIWAE}}} \left[ \log \frac{p_{\text{TGT}}^{\text{GIWAE}}(\mathbf{z}^{(1:K)}, s, \mathbf{x})}{q_{\text{PROP}}^{\text{GIWAE}}(\mathbf{z}^{(1:K)}, s|\mathbf{x})} \right]}_{\text{EUBO}_{\text{GIWAE}}(\mathbf{x}; q_\theta, T_\phi, K)}. \tag{47}$$

As in Sec. 2.1 and App. A, the gap in the lower and upper bounds can be derived as KL divergences in the extended state space.

**Upper Bound on** $\log p(\mathbf{x})$ **and Lower Bound on MI**   To derive an explicit form for the GIWAE upper bound on $\log p(\mathbf{x})$, we write

$$
\begin{aligned}
\text{EUBO}_{\text{GIWAE}}(q_\theta, T_\phi, K) &= \mathbb{E}_{p_{\text{TGT}}^{\text{GIWAE}}(\mathbf{z}^{(1:K)}, s|\mathbf{x})} \left[ \log \frac{p(\mathbf{x}, \mathbf{z}^{(s)})}{q(\mathbf{z}^{(s)}|\mathbf{x})} - T(\mathbf{x}, \mathbf{z}^{(s)}) + \log \frac{1}{K} \sum_{k=1}^{K} e^{T(\mathbf{x}, \mathbf{z}^{(k)})} \right] \\
&= \mathbb{E}_{p(\mathbf{z}|\mathbf{x})} \left[ \log \frac{p(\mathbf{x}, \mathbf{z})}{q(\mathbf{z}|\mathbf{x})} \right] - \mathbb{E}_{p(\mathbf{z}^{(1)}|\mathbf{x}) \prod_{k=2}^{K} q(\mathbf{z}^{(k)}|\mathbf{x})} \left[ \log \frac{e^{T(\mathbf{x}, \mathbf{z}^{(1)})}}{\frac{1}{K} \sum_{k=1}^{K} e^{T(\mathbf{x}, \mathbf{z}^{(k)})}} \right].
\end{aligned}
\tag{48}
$$

where, in the first term of the second line, we note that $p_{\text{TGT}}^{\text{GIWAE}}(\mathbf{z}^{(1:K)}, s|\mathbf{x})$ specifies that $\mathbf{z}^{(s)} \sim p(\mathbf{z}|\mathbf{x})$. Since $s \sim p(s) = \frac{1}{K}$ is sampled uniformly, we can assume $s = 1$ in the second term due to permutation invariance.

Translating this into a lower bound on MI, we consider $I(\mathbf{x}; \mathbf{z}) = -\mathbb{E}_{p(\mathbf{x})}[\log p(\mathbf{x})] - H(\mathbf{x}|\mathbf{z}) \geq -\mathbb{E}_{p(\mathbf{x})}[\text{EUBO}_{\text{GIWAE}}(q_\theta, T_\phi, K)] - H(\mathbf{x}|\mathbf{z})$. Writing the conditional entropy term over the index $s$,

$$
I(\mathbf{x}; \mathbf{z}) \geq \mathbb{E}_{p(\mathbf{x}, \mathbf{z})}[\log p(\mathbf{x}|\mathbf{z})]
\tag{49}
$$
$$
- \left( \mathbb{E}_{p(\mathbf{x})p(\mathbf{z}|\mathbf{x})} \left[ \log \frac{p(\mathbf{x}, \mathbf{z})}{q(\mathbf{z}|\mathbf{x})} \right] - \mathbb{E}_{p(\mathbf{x})p(\mathbf{z}^{(1)}|\mathbf{x}) \prod_{k=2}^{K} q(\mathbf{z}^{(k)}|\mathbf{x})} \left[ \log \frac{e^{T_\phi(\mathbf{x}, \mathbf{z}^{(1)})}}{\frac{1}{K} \sum_{k=1}^{K} e^{T_\phi(\mathbf{x}, \mathbf{z}^{(k)})}} \right] \right)
$$
$$
= \mathbb{E}_{p(\mathbf{x}, \mathbf{z})} \left[ \log \frac{q(\mathbf{z}|\mathbf{x})}{p(\mathbf{z})} \right] + \mathbb{E}_{p(\mathbf{x})p(\mathbf{z}^{(1)}|\mathbf{x}) \prod_{k=2}^{K} q_\theta(\mathbf{z}^{(k)}|\mathbf{x})} \left[ \log \frac{e^{T_\phi(\mathbf{x}, \mathbf{z}^{(1)})}}{\frac{1}{K} \sum_{i=1}^{K} e^{T_\phi(\mathbf{x}, \mathbf{z}^{(k)})}} \right],
\tag{50}
$$

which matches Eq. (7) from the main text.

## C.2   GIWAE UPPER BOUND ON MI DOES NOT PROVIDE BENEFIT OVER IWAE

To derive an explicit form for the GIWAE lower bound on $\log p(\mathbf{x})$,

$$
\text{ELBO}_{\text{GIWAE}}(\mathbf{x}; q_\theta, T_\phi, K) = \mathbb{E}_{q_{\text{PROP}}^{\text{GIWAE}}(\mathbf{z}^{(1:K)}, s|\mathbf{x})} \left[ \log \frac{p(\mathbf{x}, \mathbf{z}^{(s)})}{q(\mathbf{z}^{(s)}|\mathbf{x})} - T_\phi(\mathbf{x}, \mathbf{z}^{(s)}) + \log \frac{1}{K} \sum_{k=1}^{K} e^{T_\phi(\mathbf{x}, \mathbf{z}^{(k)})} \right]
$$
$$
\tag{51}
$$

Translating this to an upper bound on MI yields

$$
I(\mathbf{x}; \mathbf{z}) \leq \mathbb{E}_{p(\mathbf{z}^{(s)}|\mathbf{x})} \left[ \log p(\mathbf{x}|\mathbf{z}^{(s)}) \right] - \left( \mathbb{E}_{\prod_{k=1}^{K} q(\mathbf{z}^{(k)}|\mathbf{x})} \left[ \sum_{s=1}^{K} \frac{e^{T_\phi(\mathbf{x}, \mathbf{z}^{(s)})}}{\sum_{k=1}^{K} e^{T_\phi(\mathbf{x}, \mathbf{z}^{(k)})}} \log \frac{p(\mathbf{x}, \mathbf{z}^{(s)})}{q(\mathbf{z}^{(s)}|\mathbf{x})} \right] \right.
\tag{52}
$$
$$
\left. - \mathbb{E}_{q_{\text{PROP}}^{\text{GIWAE}}(\mathbf{z}^{(1:K)}, s|\mathbf{x})} \left[ \log \frac{e^{T(\mathbf{x}, \mathbf{z}^{(s)})}}{\frac{1}{K} \sum_{k=1}^{K} e^{T(\mathbf{x}, \mathbf{z}^{(k)})}} \right] \right)
$$

Thus, in Eq. (52), knowledge of the full joint density is required to evaluate both the conditional entropy and $\log \frac{p(\mathbf{x}, \mathbf{z}^{(s)})}{q(\mathbf{z}^{(s)}|\mathbf{x})}$ terms. If both $p(\mathbf{z})$ and $p(\mathbf{x}|\mathbf{z})$ are known, then we will show in Cor. 2.2 below that the optimal critic or negative energy function in GIWAE yields the true importance weights, and the resulting MI or $\log p(\mathbf{x})$ bounds matches the IWAE bounds.

We thus conclude that the GIWAE upper bound on MI and lower bound on $\log p(\mathbf{x})$ does not provide any benefit over IWAE in practice. However, $\text{ELBO}_{\text{GIWAE}}(\mathbf{x}; q_\theta, T_\phi, K)$ is still useful for analysis, as our proof of Lemma C.1 below allows us characterize the gap between $\text{ELBO}_{\text{IWAE}}(\mathbf{x}; q_\theta, K)$ and $\text{ELBO}(\mathbf{x}; q_\theta)$ in Prop. B.2.

### C.3 ELBO AND EUBO ARE SPECIAL CASES OF GIWAE LOG PARTITION FUNCTION BOUNDS

**Lemma C.1.** *The single-sample* ELBO *and* EUBO *are special cases of* GIWAE*, with*

$$\text{ELBO}(\mathbf{x}; q_\theta) = \text{ELBO}_{\text{GIWAE}}(\mathbf{x}; q_\theta, T_{\phi_0} = const, K),$$
$$\text{EUBO}(\mathbf{x}; q_\theta) = \text{EUBO}_{\text{GIWAE}}(\mathbf{x}; q_\theta, T_{\phi_0} = const, K).$$

*In both cases, the* SNIS *sampling distribution ([Eq. (43)](#)) is uniform* $q_{\text{PROP}}^{\text{GIWAE}}(\mathbf{z}^{(1:K)}, s|\mathbf{x}) = \frac{1}{K} = \mathcal{U}(s)$.

*Proof.* We consider the GIWAE probabilistic interpretation ([App. C.1](#)) for $T_{\phi_0} = \text{const}$. We refer to this extended state space proposal as $q_{\text{PROP}}^{\text{BA}}$, since it leads to $I_{\text{BA}_L}(q_\theta)$ and $I_{\text{BA}_U}(q_\theta)$ bounds on MI.

$$q_{\text{PROP}}^{\text{BA}}(\mathbf{z}^{(1:K)}, s|\mathbf{x}) = \left( \prod_{k=1}^K q_\theta(\mathbf{z}^{(k)}|\mathbf{x}) \right) q_{\text{PROP}}^{\text{BA}}(s|\mathbf{z}^{(1:K)}, \mathbf{x}), \tag{53}$$

$$\text{where } q_{\text{PROP}}^{\text{BA}}(s|\mathbf{z}^{(1:K)}, \mathbf{x}) = \frac{e^{T_{\phi_0}(\mathbf{x}, \mathbf{z}^{(s)})}}{\sum_{k=1}^K e^{T_{\phi_0}(\mathbf{x}, \mathbf{z}^{(k)})}} = \mathcal{U}(s) = \frac{1}{K}. \tag{54}$$

Note that the SNIS sampling distribution $q_{\text{PROP}}^{\text{BA}}(s|\mathbf{z}^{(1:K)}, \mathbf{x})$ will be uniform, which matches $p(s) = \frac{1}{K}$ in the GIWAE extended state space target distribution

$$p_{\text{TGT}}^{\text{GIWAE}}(\mathbf{z}^{(1:K)}, s|\mathbf{x}) = \frac{1}{K} p(\mathbf{z}^{(s)}|\mathbf{x}) \prod_{\substack{k=1 \\ k \neq s}}^K q_\theta(\mathbf{z}^{(k)}|\mathbf{x}) \tag{55}$$

Now, taking the log unnormalized importance weights, we obtain

$$\log \frac{p_{\text{TGT}}^{\text{GIWAE}}(\mathbf{z}^{(1:K)}, s|\mathbf{x})}{q_{\text{PROP}}^{\text{BA}}(\mathbf{z}^{(1:K)}, s|\mathbf{x})} = \log \frac{\frac{1}{K} p(\mathbf{z}^{(s)}|\mathbf{x}) \prod_{\substack{k=1 \\ k \neq s}}^K q_\theta(\mathbf{z}^{(k)}|\mathbf{x})}{\frac{1}{K} \prod_{k=1}^K q_\theta(\mathbf{z}^{(k)}|\mathbf{x})} = \log \frac{p(\mathbf{x}, \mathbf{z}^{(s)})}{q_\theta(\mathbf{z}^{(s)}|\mathbf{x})} \tag{56}$$

Taking expectations with respect to $q_{\text{PROP}}^{\text{BA}}(\mathbf{z}^{(1:K)}, s|\mathbf{x})$ or $p_{\text{TGT}}^{\text{GIWAE}}(\mathbf{z}^{(1:K)}, s|\mathbf{x})$ leads to $\text{ELBO}(\mathbf{x}; q_\theta)$ and $\text{EUBO}(\mathbf{x}; q_\theta)$ respectively, as in [Sec. 2-2.2](#). □

### C.4 PROOF OF RELATIONSHIP BETWEEN IWAE AND GIWAE PROBABILISTIC INTERPRETATIONS ([PROP. 2.1](#))

We first prove a lemma which relates both the GIWAE lower and upper bounds on $\log p(\mathbf{x})$ to the respective IWAE bounds. From this lemma, [Prop. B.2](#) follows directly and relates the ELBO and EUBO (which are special cases of $\text{ELBO}_{\text{GIWAE}}$ and $\text{EUBO}_{\text{GIWAE}}$) to $\text{ELBO}_{\text{IWAE}}$ and $\text{EUBO}_{\text{IWAE}}$.

**Lemma C.2.** *We can characterize the difference between* IWAE *and* GIWAE *bounds on* $\log p(\mathbf{x})$ *using* KL *divergences between their respective* SNIS *distributions.*

$$\text{ELBO}_{\text{IWAE}}(\mathbf{x}; q_\theta, K) = \text{ELBO}_{\text{GIWAE}}(\mathbf{x}; q_\theta, T_\phi, K) + \mathbb{E}_{q_{\text{PROP}}^{\text{GIWAE}}(\mathbf{z}^{(1:K)}|\mathbf{x})} \left[ D_{\text{KL}} \left[ q_{\text{PROP}}^{\text{GIWAE}}(s|\mathbf{z}^{(1:K)}, \mathbf{x}) \| p_{\text{TGT}}^{\text{IWAE}}(s|\mathbf{z}^{(1:K)}, \mathbf{x}) \right] \right] \tag{57}$$

$$\text{EUBO}_{\text{IWAE}}(\mathbf{x}; q_\theta, K) = \text{EUBO}_{\text{GIWAE}}(\mathbf{x}; q_\theta, T_\phi, K) - \mathbb{E}_{p_{\text{TGT}}^{\text{IWAE}}(\mathbf{z}^{(1:K)}|\mathbf{x})} \left[ D_{\text{KL}} \left[ p_{\text{TGT}}^{\text{IWAE}}(s|\mathbf{z}^{(1:K)}, \mathbf{x}) \| q_{\text{PROP}}^{\text{GIWAE}}(s|\mathbf{z}^{(1:K)}, \mathbf{x}) \right] \right] \tag{58}$$

*Proof.* Recall from [Sec. 2.1](#) or [App. A](#) that the gap of the lower bound $\text{ELBO}_{\text{GIWAE}}(\mathbf{x}; q_\theta, T_\phi, K)$ is $D_{\text{KL}}[q_{\text{PROP}}^{\text{GIWAE}}(\mathbf{z}^{(1:K)}, s|\mathbf{x}) \| p_{\text{TGT}}^{\text{GIWAE}}(\mathbf{z}^{(1:K)}, s, \mathbf{x})]$, while the gap of the upper bound $\text{EUBO}_{\text{GIWAE}}(\mathbf{x}; q_\theta, T_\phi, K)$ is $D_{\text{KL}}[p_{\text{TGT}}^{\text{GIWAE}}(\mathbf{z}^{(1:K)}, s|\mathbf{x}) \| q_{\text{PROP}}^{\text{GIWAE}}(\mathbf{z}^{(1:K)}, s|\mathbf{x})]$. We will expand these KL divergences to reveal the relationship between the GIWAE bounds and IWAE bounds.

First, recall from [Eq. (41)](#) that the posterior over the index variable $s$, or target SNIS distribution, is

$$p_{\text{TGT}}^{\text{GIWAE}}(s|\mathbf{z}^{(1:K)}, \mathbf{x}) = \frac{p_{\text{TGT}}^{\text{GIWAE}}(\mathbf{z}^{(1:K)}, s, \mathbf{x})}{\sum_{s=1}^K p_{\text{TGT}}^{\text{GIWAE}}(\mathbf{z}^{(1:K)}, s, \mathbf{x})} = \frac{\frac{1}{K} p(\mathbf{x}, \mathbf{z}^{(s)}) \prod_{\substack{k=1 \\ k \neq s}}^K q_\theta(\mathbf{z}^{(k)}|\mathbf{x})}{\frac{1}{K} \sum_{s=1}^K p(\mathbf{x}, \mathbf{z}^{(s)}) \prod_{\substack{k=1 \\ k \neq s}}^K q_\theta(\mathbf{z}^{(k)}|\mathbf{x})} = \frac{\frac{p(\mathbf{x}, \mathbf{z}^{(s)})}{q(\mathbf{z}^{(s)}|\mathbf{x})}}{\sum_{s=1}^K \frac{p(\mathbf{x}, \mathbf{z}^{(s)})}{q(\mathbf{z}^{(s)}|\mathbf{x})}}.$$

The joint distribution then factorizes as $p_{\text{TGT}}^{\text{GIWAE}}(\mathbf{z}^{(1:K)}, s, \mathbf{x}) = p_{\text{TGT}}^{\text{GIWAE}}(\mathbf{z}^{(1:K)}, \mathbf{x}) \cdot p_{\text{TGT}}^{\text{GIWAE}}(s|\mathbf{z}^{(1:K)}, \mathbf{x})$.

*ELBO Case*: Using this factorization of $p_{\text{TGT}}^{\text{GIWAE}}(\mathbf{z}^{(1:K)}, s, \mathbf{x})$, we can rewrite the gap of $\text{ELBO}_{\text{GIWAE}}(\mathbf{x}; q_\theta, T_\phi, K)$ as follows

$$D_{\text{KL}}[q_{\text{PROP}}^{\text{GIWAE}}(\mathbf{z}^{(1:K)}, s|\mathbf{x})\|p_{\text{TGT}}^{\text{GIWAE}}(\mathbf{z}^{(1:K)}, s|\mathbf{x})] \tag{59}$$

$$= \mathbb{E}_{q_{\text{PROP}}^{\text{GIWAE}}(\mathbf{z}^{(1:K)}, s|\mathbf{x})}\left[\log \frac{q_{\text{PROP}}^{\text{GIWAE}}(\mathbf{z}^{(1:K)}|\mathbf{x})}{p_{\text{TGT}}^{\text{GIWAE}}(\mathbf{z}^{(1:K)}|\mathbf{x})} \frac{q_{\text{PROP}}^{\text{GIWAE}}(s|\mathbf{z}^{(1:K)}, \mathbf{x})}{p_{\text{TGT}}^{\text{GIWAE}}(s|\mathbf{z}^{(1:K)}, \mathbf{x})}\right]$$

$$= D_{\text{KL}}\left[\prod_{k=1}^{K} q_\theta(\mathbf{z}^{(k)}|\mathbf{x}) \middle\| \frac{1}{K}\sum_{s=1}^{K} p(\mathbf{z}^{(s)}|\mathbf{x})\prod_{\substack{k=1\\k\neq s}}^{K} q_\theta(\mathbf{z}^{(k)}|\mathbf{x})\right] + \mathbb{E}_{q_{\text{PROP}}^{\text{GIWAE}}}\left[D_{\text{KL}}[q_{\text{PROP}}^{\text{GIWAE}}(s|\mathbf{z}^{(1:K)}, \mathbf{x})\|p_{\text{TGT}}^{\text{GIWAE}}(s|\mathbf{z}^{(1:K)}, \mathbf{x})]\right]$$

$$= D_{\text{KL}}[q_{\text{PROP}}^{\text{IWAE}}(\mathbf{z}^{(1:K)}|\mathbf{x})\|p_{\text{TGT}}^{\text{IWAE}}(\mathbf{z}^{(1:K)}|\mathbf{x})] + \mathbb{E}_{q_{\text{PROP}}^{\text{GIWAE}}}\left[D_{\text{KL}}[q_{\text{PROP}}^{\text{GIWAE}}(s|\mathbf{z}^{(1:K)}, \mathbf{x})\|p_{\text{TGT}}^{\text{GIWAE}}(s|\mathbf{z}^{(1:K)}, \mathbf{x})]\right], \tag{60}$$

where we can recognize the first term as the gap in $\text{ELBO}_{\text{IWAE}}(\mathbf{x}; q_\theta, K)$. Noting that $\text{ELBO}_{\text{IWAE}}(\mathbf{x}; q_\theta, K) - \text{ELBO}_{\text{GIWAE}}(\mathbf{x}; q_\theta, T_\phi, K) = D_{\text{KL}}[q_{\text{PROP}}^{\text{GIWAE}}(\mathbf{z}^{(1:K)}, s|\mathbf{x})\|p_{\text{TGT}}^{\text{GIWAE}}(\mathbf{z}^{(1:K)}, s|\mathbf{x})] - D_{\text{KL}}[q_{\text{PROP}}^{\text{IWAE}}(\mathbf{z}^{(1:K)}|\mathbf{x})\|p_{\text{TGT}}^{\text{IWAE}}(\mathbf{z}^{(1:K)}|\mathbf{x})]$, we obtain Eq. (57), as desired

$$\text{ELBO}_{\text{IWAE}}(\mathbf{x}; q_\theta, K) = \text{ELBO}_{\text{GIWAE}}(\mathbf{x}; q_\theta, T_\phi, K) + \mathbb{E}_{q_{\text{PROP}}^{\text{GIWAE}}(\mathbf{z}^{(1:K)}|\mathbf{x})}\left[D_{\text{KL}}[q_{\text{PROP}}^{\text{GIWAE}}(s|\mathbf{z}^{(1:K)}, \mathbf{x})\|p_{\text{TGT}}^{\text{GIWAE}}(s|\mathbf{z}^{(1:K)}, \mathbf{x})]\right].$$

*EUBO Case*: For $\text{ELBO}_{\text{GIWAE}}(\mathbf{x}; q_\theta, T_\phi, K)$, the derivations follow in a similar fashion to Eq. (59)-Eq. (60), but using the reverse KL divergence and expectations under $p_{\text{TGT}}^{\text{GIWAE}}(\mathbf{z}^{(1:K)}, s|\mathbf{x})$. □

Translating Lemma C.2 to a statement relating IWAE and GIWAE bounds on MI, we obtain the following proposition.

**Proposition 2.1** (Improvement of IWAE over GIWAE). *For a given $q_\theta(\mathbf{z}|\mathbf{x})$ and any $T_\phi(\mathbf{x}, \mathbf{z})$,*

$$I_{\text{IWAE}_L}(q_\theta, K) = I_{\text{GIWAE}_L}(q_\theta, T_\phi, K) + \mathbb{E}_{p(\mathbf{x})p_{\text{TGT}}^{\text{GIWAE}}(\mathbf{z}^{(1:K)}|\mathbf{x})}\left[D_{\text{KL}}[p_{\text{TGT}}^{\text{GIWAE}}(s|\mathbf{z}^{(1:K)}, \mathbf{x})\|q_{\text{PROP}}^{\text{GIWAE}}(s|\mathbf{z}^{(1:K)}, \mathbf{x})]\right].$$

*Proof.* The result follows directly from Lemma C.2. First, note that $p_{\text{TGT}}^{\text{IWAE}}(\mathbf{x}, \mathbf{z}^{(1:K)}, s) = p_{\text{TGT}}^{\text{GIWAE}}(\mathbf{x}, \mathbf{z}^{(1:K)}, s)$. Then, using the upper bounds on $\log p(\mathbf{x})$ and taking outer expectations with respect to $p(\mathbf{x})$, we have

$$I_{\text{IWAE}_L}(q_\theta, K) - I_{\text{GIWAE}_L}(q_\theta, T_\phi, K) = -H(\mathbf{x}|\mathbf{z}) - \mathbb{E}_{p(\mathbf{x})}\left[\text{EUBO}_{\text{IWAE}}(\mathbf{x}; q_\theta, K)\right] \tag{61}$$
$$+ H(\mathbf{x}|\mathbf{z}) + \mathbb{E}_{p(\mathbf{x})}\left[\text{EUBO}_{\text{GIWAE}}(\mathbf{x}; q_\theta, T_\phi, K)\right]$$

$$= \mathbb{E}_{p(\mathbf{x})p_{\text{TGT}}^{\text{IWAE}}(\mathbf{z}^{(1:K)}|\mathbf{x})}\left[D_{\text{KL}}[p_{\text{TGT}}^{\text{IWAE}}(s|\mathbf{z}^{(1:K)}, \mathbf{x}) \| q_{\text{PROP}}^{\text{GIWAE}}(s|\mathbf{z}^{(1:K)}, \mathbf{x})]\right]. \tag{62}$$

□

## C.5 Proof of GIWAE Optimal Critic Function and Logarithmic Improvement (Cor. 2.2 and Cor. 2.3)

We now prove Cor. 2.2 and Cor. 2.3 from the main text, with a corollary stating the results for the special case of INFONCE in App. C.6.

**Corollary 2.2.** *For a given $q_\theta(\mathbf{z}|\mathbf{x})$ and $K > 1$, the optimal critic function is the true log importance weight up to an arbitrary constant: $T^*(\mathbf{x}, \mathbf{z}) = \log \frac{p(\mathbf{x}, \mathbf{z})}{q_\theta(\mathbf{z}|\mathbf{x})} + c(\mathbf{x})$. With this choice of $T^*(\mathbf{x}, \mathbf{z})$,*

$$I_{\text{GIWAE}_L}(q_\theta, T^*, K) = I_{\text{IWAE}_L}(q_\theta, K). \tag{8}$$

*Proof.* Using Prop. 2.1, we can see that the gap in the GIWAE and IWAE bounds, which corresponds to the posterior KL divergence over the index variable $s$, will equal zero iff

$$q_{\text{PROP}}^{\text{GIWAE}}(s|\mathbf{z}^{(1:K)}, \mathbf{x}) = p_{\text{TGT}}^{\text{GIWAE}}(s|\mathbf{z}^{(1:K)}, \mathbf{x}) \implies \frac{e^{T(\mathbf{x}, \mathbf{z}^{(s)})}}{\sum_{k=1}^{K} e^{T(\mathbf{x}, \mathbf{z}^{(k)})}} = \frac{\frac{p(\mathbf{x}, \mathbf{z}^{(s)})}{q_\theta(\mathbf{z}^{(s)}|\mathbf{x})}}{\sum_{s=1}^{K} \frac{p(\mathbf{x}, \mathbf{z}^{(s)})}{q_\theta(\mathbf{z}^{(s)}|\mathbf{x})}} \tag{63}$$

This condition also ensures that the overall GIWAE proposal $p_{\text{TGT}}^{\text{GIWAE}}(\mathbf{z}^{(1:K)}, s|\mathbf{x})$ (Eq. (40)) and target $q_{\text{PROP}}^{\text{GIWAE}}(\mathbf{z}^{(1:K)}, s|\mathbf{x})$ (Eq. (42)) distributions match. We will show that any $T(\mathbf{x}, \mathbf{z})$ which satisfies Eq. (63) has the form

$$T^*(\mathbf{x}, \mathbf{z}) = \log \frac{p(\mathbf{x}, \mathbf{z})}{q_\theta(\mathbf{z}|\mathbf{x})} + c(\mathbf{x}). \tag{64}$$

Let $f(\mathbf{x}, \mathbf{z}) = \log \frac{p(\mathbf{x}, \mathbf{z})}{q_\theta(\mathbf{z}|\mathbf{x})} + g(\mathbf{x}, \mathbf{z})$, which represents an arbitrary choice of critic function. We will show that $g(\mathbf{x}, \mathbf{z})$ must be constant with respect to $\mathbf{z}$

$$\frac{e^{\log \frac{p(\mathbf{x}, \mathbf{z}^{(s)})}{q_\theta(\mathbf{z}^{(s)}|\mathbf{x})} + g(\mathbf{x}, \mathbf{z}^{(s)})}}{\sum_{k=1}^{K} e^{\log \frac{p(\mathbf{x}, \mathbf{z}^{(s)})}{q_\theta(\mathbf{z}^{(s)}|\mathbf{x})} + g(\mathbf{x}, \mathbf{z}^{(s)})}} = \frac{\frac{p(\mathbf{x}, \mathbf{z}^{(s)})}{q_\theta(\mathbf{z}^{(s)}|\mathbf{x})}}{\sum_{k=1}^{K} \frac{p(\mathbf{x}, \mathbf{z}^{(k)})}{q_\theta(\mathbf{z}^{(k)}|\mathbf{x})}}$$

$$\implies e^{\log \frac{p(\mathbf{x}, \mathbf{z}^{(s)})}{q_\theta(\mathbf{z}^{(s)}|\mathbf{x})}} \cdot e^{g(\mathbf{x}, \mathbf{z}^{(s)})} \cdot \sum_{k=1}^{K} \frac{p(\mathbf{x}, \mathbf{z}^{(k)})}{q_\theta(\mathbf{z}^{(k)}|\mathbf{x})} = \frac{p(\mathbf{x}, \mathbf{z}^{(s)})}{q_\theta(\mathbf{z}^{(s)}|\mathbf{x})} \cdot \sum_{k=1}^{K} e^{\log \frac{p(\mathbf{x}, \mathbf{z}^{(k)})}{q_\theta(\mathbf{z}^{(k)}|\mathbf{x})} + g(\mathbf{x}, \mathbf{z}^{(k)})}$$

$$\sum_{k=1}^{K} \frac{p(\mathbf{x}, \mathbf{z}^{(k)})}{q_\theta(\mathbf{z}^{(k)}|\mathbf{x})} \frac{p(\mathbf{x}, \mathbf{z}^{(s)})}{q_\theta(\mathbf{z}^{(s)}|\mathbf{x})} e^{g(\mathbf{x}, \mathbf{z}^{(s)})} = \sum_{k=1}^{K} \frac{p(\mathbf{x}, \mathbf{z}^{(k)})}{q_\theta(\mathbf{z}^{(k)}|\mathbf{x})} \frac{p(\mathbf{x}, \mathbf{z}^{(s)})}{q_\theta(\mathbf{z}^{(s)}|\mathbf{x})} e^{g(\mathbf{x}, \mathbf{z}^{(k)})}$$

$$\implies g(\mathbf{x}, \mathbf{z}) = c(\mathbf{x})$$

where $g(\mathbf{x}, \mathbf{z}) = c(\mathbf{x})$ is required in order to ensure that $g(\mathbf{x}, \mathbf{z}^{(s)}) = g(\mathbf{x}, \mathbf{z}^{(k)})$ for arbitrary choices of $\mathbf{z}$ samples.

This form for the optimal critic function $T^*(\mathbf{x}, \mathbf{z})$ in Eq. (64) implies that learning $T_\phi(\mathbf{x}, \mathbf{z})$ in GIWAE becomes unnecessary when the density ratio $\frac{p(\mathbf{x}, \mathbf{z})}{q_\theta(\mathbf{z}|\mathbf{x})}$ is available in closed form, as is assumed in the IWAE bound.

For this choice of $T^*(\mathbf{x}, \mathbf{z})$, the value of the GIWAE objective matches the IWAE lower bound on MI.

$$I_{\text{GIWAE}}(q_\theta, T^*, K) = \mathbb{E}_{p(\mathbf{x}, \mathbf{z})} \left[ \log \frac{q(\mathbf{z}|\mathbf{x})}{p(\mathbf{z})} \right] + \mathbb{E}_{p(\mathbf{x}, \mathbf{z}^{(1)}) \prod_{k=2}^{K} q_\theta(\mathbf{z}^{(k)}|\mathbf{x})} \left[ \log \frac{e^{\log \frac{p(\mathbf{x}, \mathbf{z}^{(1)})}{q(\mathbf{z}^{(1)}|\mathbf{x})}} \cdot e^{c(\mathbf{x})}}{\frac{1}{K} \sum_{k=1}^{K} e^{\log \frac{p(\mathbf{x}, \mathbf{z}^{(k)})}{q_\theta(\mathbf{z}^{(k)}|\mathbf{x})}} \cdot e^{c(\mathbf{x})}} \right]$$

$$= \mathbb{E}_{p(\mathbf{x}, \mathbf{z})} \left[ \log \frac{\cancel{q(\mathbf{z}|\mathbf{x})}}{p(\mathbf{z})} + \log \frac{p(\mathbf{x}, \mathbf{z})}{\cancel{q(\mathbf{z}|\mathbf{x})}} \right] - \mathbb{E}_{p(\mathbf{x}, \mathbf{z}^{(1)}) \prod_{k=2}^{K} q_\theta(\mathbf{z}^{(k)}|\mathbf{x})} \left[ \log \frac{1}{K} \sum_{i=1}^{K} \frac{p(\mathbf{x}, \mathbf{z}^{(k)})}{q_\theta(\mathbf{z}^{(k)}|\mathbf{x})} \right] \tag{65}$$

$$= -H(\mathbf{x}|\mathbf{z}) + \left( -\mathbb{E}_{p(\mathbf{x}) p(\mathbf{z}^{(s)}|\mathbf{x}) \prod_{k=2}^{K} q_\theta(\mathbf{z}^{(k)}|\mathbf{x})} \left[ \log \frac{1}{K} \sum_{i=1}^{K} \frac{p(\mathbf{x}, \mathbf{z}^{(k)})}{q_\theta(\mathbf{z}^{(k)}|\mathbf{x})} \right] \right)$$

$$= I_{\text{IWAE}_L}(q_\theta, K).$$

The second term in Eq. (65) is exactly the negative of the IWAE upper bound on $\log p(\mathbf{x})$ in Eq. (34). Combined with the entropy $-H(\mathbf{x}|\mathbf{z})$, we obtain the $I_{\text{IWAE}_L}(q_\theta, K)$ lower bound on MI as desired. $\square$

**Corollary 2.3.** *Suppose the critic function $T_\phi(\mathbf{x}, \mathbf{z})$ is parameterized by $\phi$, and $\exists \phi_0 \text{ s.t. } \forall (\mathbf{x}, \mathbf{z}), T_{\phi_0}(\mathbf{x}, \mathbf{z}) = \text{const}$. For a given $q_\theta(\mathbf{z}|\mathbf{x})$, let $T_{\phi^*}(\mathbf{x}, \mathbf{z})$ denote the critic function that maximizes the GIWAE lower bound. Using Cor. 2.2, we have*

$$I_{\text{BA}_L}(q_\theta) \leq I_{\text{GIWAE}_L}(q_\theta, T_{\phi^*}, K) \leq I_{\text{IWAE}_L}(q, K) \leq I_{\text{BA}_L}(q_\theta) + \log K. \tag{9}$$

*Proof.* We begin by showing that $I_{\text{BA}_L}(q_\theta) \leq I_{\text{GIWAE}_L}(q_\theta, T_{\phi^*}, K)$. This follows from the assumption that there exists $\phi_0$ in the parameter space of the neural network $T_\phi$ such that $T_{\phi_0} = \text{const}$. With this $\phi_0$, we would have $I_{\text{BA}_L}(q_\theta) = I_{\text{GIWAE}_L}(q_\theta, T_{\phi_0}, K)$ as in Lemma C.1. Thus, the optimal $\phi^*$ in the parameter space can only improve upon the BA bound.

Next, for any given $\phi$ (including $\phi^*$), we have $I_{\text{GIWAE}_L}(q_\theta, T_\phi, K) \leq I_{\text{IWAE}_L}(q_\theta, K)$, since IWAE uses the (unconstrained) optimal critic function $T^* = \log \frac{p(\mathbf{x}, \mathbf{z})}{q_\theta(\mathbf{z}|\mathbf{x})} + c(\mathbf{x})$ (Cor. 2.2). The final inequality follows from Prop. B.2, which shows that $I_{\text{IWAE}_L}(q_\theta, K)$ improves by at most $\log K$ over $I_{\text{BA}_L}(q_\theta)$. These relationships are visualized in Fig. 3b. $\square$

### C.6 PROPERTIES OF INFONCE

**Corollary C.3.** *Using the prior* $q_\theta(\mathbf{z}|\mathbf{x}) = p(\mathbf{z})$ *as in* INFONCE,

(a) *For the optimal critic function,* Cor. 2.2 *implies* $T^*(\mathbf{x}, \mathbf{z}) = \log p(\mathbf{x}|\mathbf{z}) + c(\mathbf{x})$, *and*

$$I_{\text{GIWAE}_L}(p(\mathbf{z}), T^*, K) = I_{\text{INFONCE}_L}(T^*, K) = I_{\text{S-INFONCE}_L}(K). \tag{66}$$

(b) *For an arbitrary critic function* $T_\phi(\mathbf{x}, \mathbf{z})$, Cor. 2.3 *translates to*

$$0 \le I_{\text{INFONCE}_L}(T_\phi, K) \le I_{\text{S-INFONCE}_L}(K) \le \log K. \tag{67}$$

For INFONCE, note that using the prior as the proposal does allow the critic to admit an efficient bi-linear implementation $T_\phi(\mathbf{x}, \mathbf{z}) = f_{\phi_\mathbf{x}}(\mathbf{x})^T f_{\phi_\mathbf{z}}(\mathbf{z})$, which requires only $N + K$ forward passes instead of $NK$ for GIWAE, where $N$ is the batch size and $K$ is the total number of positive and negative samples.

## D SINGLE-SAMPLE AIS

### D.1 PROOF OF PROP. 3.1 (COMPLEXITY IN $T$ FOR SINGLE-SAMPLE AIS)

In this section, we prove Prop. 3.1, which relates the sum of the gaps in the single-sample AIS upper and lower bounds to the symmetrized KL divergence between the endpoint distributions. We extend this result to IM-AIS in Cor. E.1 and CR-AIS in Cor. H.3.

**Proposition 3.1** (Complexity in $T$). *Assuming perfect transitions and a geometric annealing path with linearly-spaced* $\{\beta_t\}_{t=1}^T$, *the sum of the gaps in the* AIS *sandwich bounds on* MI, $I_{\text{AIS}_U}(\pi_0, T) - I_{\text{AIS}_L}(\pi_0, T)$, *reduces linearly with increasing* $T$.

In particular, we will show that

$$\text{EUBO}_{\text{AIS}}(\mathbf{x}; \pi_0, T) - \text{ELBO}_{\text{AIS}}(\mathbf{x}; \pi_0, T) \le \frac{1}{T} \big( D_{\text{KL}}[\pi_T(\mathbf{z}|\mathbf{x}) \| \pi_0(\mathbf{z})] + D_{\text{KL}}[\pi_0(\mathbf{z}) \| \pi_T(\mathbf{z}|\mathbf{x})] \tag{68}$$

where the left hand side also corresponds to the sum of the gaps $D_{\text{KL}}[q_{\text{PROP}}^{\text{AIS}}(\mathbf{z}_{0:T}|\mathbf{x}) \| p_{\text{TGT}}^{\text{AIS}}(\mathbf{z}_{0:T}|\mathbf{x})] + D_{\text{KL}}[p_{\text{TGT}}^{\text{AIS}}(\mathbf{z}_{0:T}|\mathbf{x}) \| q_{\text{PROP}}^{\text{AIS}}(\mathbf{z}_{0:T}|\mathbf{x})]$. This result translates to mutual information bounds as in Sec. 1.1.

In contrast to Grosse et al. (2013) Thm. 1, our linear bias reduction result holds for finite $T$.

*Proof.* For linear scheduling and perfect transitions, we simplify the difference in the single-sample upper and lower bounds as

$$\delta_L^{T,K=1} + \delta_U^{T,K=1} = \text{EUBO}_{\text{AIS}}(\mathbf{x}; \pi_0, T) - \text{ELBO}_{\text{AIS}}(\mathbf{x}; \pi_0, T)$$

$$= \mathbb{E}_{\mathbf{z}_{0:T} \sim p_{\text{TGT}}^{\text{AIS}}} \left[ \log \frac{p_{\text{TGT}}^{\text{AIS}}(\mathbf{x}, \mathbf{z}_{0:T})}{q_{\text{PROP}}^{\text{AIS}}(\mathbf{z}_{0:T}|\mathbf{x})} \right] - \mathbb{E}_{\mathbf{z}_{0:T} \sim q_{\text{PROP}}^{\text{AIS}}} \left[ \log \frac{p_{\text{TGT}}^{\text{AIS}}(\mathbf{x}, \mathbf{z}_{0:T})}{q_{\text{PROP}}^{\text{AIS}}(\mathbf{z}_{0:T}|\mathbf{x})} \right]$$

$$= \mathbb{E}_{\mathbf{z}_{0:T} \sim p_{\text{TGT}}^{\text{AIS}}} \left[ \log \frac{\pi_T(\mathbf{z}_T|\mathbf{x}) \prod_{t=1}^T \tilde{\mathcal{T}}_t(\mathbf{z}_{t-1}|\mathbf{z}_t)}{\pi_0(\mathbf{z}_0|\mathbf{x}) \prod_{t=1}^T \mathcal{T}_t(\mathbf{z}_t|\mathbf{z}_{t-1})} \right] - \mathbb{E}_{\mathbf{z}_{0:T} \sim q_{\text{PROP}}^{\text{AIS}}} \left[ \log \frac{\pi_T(\mathbf{z}_T|\mathbf{x}) \prod_{t=1}^T \tilde{\mathcal{T}}_t(\mathbf{z}_{t-1}|\mathbf{z}_t)}{\pi_0(\mathbf{z}_0|\mathbf{x}) \prod_{t=1}^T \mathcal{T}_t(\mathbf{z}_t|\mathbf{z}_{t-1})} \right]$$

$$= \mathbb{E}_{\mathbf{z}_{0:T} \sim p_{\text{TGT}}^{\text{AIS}}} \left[ \log \prod_{t=1}^T \left( \frac{\tilde{\pi}_T(\mathbf{x}, \mathbf{z}_t)}{\tilde{\pi}_0(\mathbf{z}_t|\mathbf{x})} \right)^{\beta_t - \beta_{t-1}} \right] - \mathbb{E}_{\mathbf{z}_{0:T} \sim q_{\text{PROP}}^{\text{AIS}}} \left[ \log \prod_{t=1}^T \left( \frac{\tilde{\pi}_T(\mathbf{x}, \mathbf{z}_t)}{\tilde{\pi}_0(\mathbf{z}_t|\mathbf{x})} \right)^{\beta_t - \beta_{t-1}} \right]$$

$$= \mathbb{E}_{\mathbf{z}_{0:T} \sim p_{\text{TGT}}^{\text{AIS}}} \left[ \sum_{t=1}^T (\beta_t - \beta_{t-1}) \log \frac{\tilde{\pi}_T(\mathbf{x}, \mathbf{z}_t)}{\tilde{\pi}_0(\mathbf{z}_t|\mathbf{x})} \right] - \mathbb{E}_{\mathbf{z}_{0:T} \sim q_{\text{PROP}}^{\text{AIS}}} \left[ \sum_{t=1}^T (\beta_t - \beta_{t-1}) \log \frac{\tilde{\pi}_T(\mathbf{x}, \mathbf{z}_t)}{\tilde{\pi}_0(\mathbf{z}_t|\mathbf{x})} \right]. \tag{69}$$

$$\overset{(1)}{=} \sum_{t=1}^T \mathbb{E}_{\pi_{\beta_t}(\mathbf{z})} \left[ (\beta_t - \beta_{t-1}) \log \frac{\tilde{\pi}_T(\mathbf{x}, \mathbf{z})}{\tilde{\pi}_0(\mathbf{z}|\mathbf{x})} \right] - \sum_{t=1}^T \mathbb{E}_{\pi_{\beta_{t-1}}(\mathbf{z})} \left[ (\beta_t - \beta_{t-1}) \log \frac{\tilde{\pi}_T(\mathbf{x}, \mathbf{z})}{\tilde{\pi}_0(\mathbf{z}|\mathbf{x})} \right]$$

$$\overset{(2)}{=} \frac{1}{T} \mathbb{E}_{\pi_T(\mathbf{z})} \left[ \log \frac{\tilde{\pi}_T(\mathbf{x}, \mathbf{z})}{\tilde{\pi}_0(\mathbf{z}|\mathbf{x})} \right] - \frac{1}{T} \mathbb{E}_{\pi_0(\mathbf{z})} \left[ \log \frac{\tilde{\pi}_T(\mathbf{x}, \mathbf{z})}{\tilde{\pi}_0(\mathbf{z}|\mathbf{x})} \right]$$

$$= \frac{1}{T} \big( D_{\text{KL}}[\pi_0 \| \pi_T] + D_{\text{KL}}[\pi_T \| \pi_0] \big),$$

where in (2), we use the linear annealing schedule $\beta_t - \beta_{t-1} = \frac{1}{T}$ $\forall t$ and note that intermediate terms cancel in telescoping fashion. In (1), we have used the assumption of perfect transitions (PT), which is common in analysis of AIS (Neal, 2001; Grosse et al., 2013). In this case, the AIS proposal and target distributions have the following factorial form

$$\mathbf{z}_{0:T} \sim q_{\text{PROP}}^{\text{AIS}}(\mathbf{z}_{0:T}^{(1:K)}|\mathbf{x}) \stackrel{\text{(PT)}}{=} \pi_0(\mathbf{z}_0) \prod_{t=1}^{T} \pi_{\beta_{t-1}}(\mathbf{z}_t), \tag{70}$$

$$\mathbf{z}_{0:T} \sim p_{\text{TGT}}^{\text{AIS}}(\mathbf{z}_{0:T}^{(1:K)}|\mathbf{x}) \stackrel{\text{(PT)}}{=} \pi_T(\mathbf{z}_T) \prod_{t=1}^{T} \pi_{\beta_{t-1}}(\mathbf{z}_{t-1}). \tag{71}$$

In other words, for $1 \leq t \leq T$, perfect transitions results in independent, exact samples from $\mathbf{z}_t \sim \pi_{\beta_{t-1}}(\mathbf{z})$ in the forward direction, and $\mathbf{z}_t \sim \pi_{\beta_t}(\mathbf{z})$ in the reverse direction. Using the factorized structure of Eq. (70) and Eq. (71), the expectations over the extended state space simplify to a sum of expectations at each $\mathbf{z}_t$.

The above proves the proposition for the case of single sample AIS, but should also hold for our tighter multi-sample AIS bounds. We extend this result to Independent Multi-Sample AIS in App. E.2 below, and extend the result to Coupled Reverse Multi-Sample AIS in App. H.4. □

# E    INDEPENDENT MULTI-SAMPLE AIS

## E.1    PROBABILISTIC INTERPRETATION AND BOUNDS

Our Independent Multi-Sample AIS (IM-AIS) are identical to the standard IWAE bounds in App. B, but using AIS forward $q_{\text{PROP}}^{\text{AIS}}(\mathbf{z}_{0:T}|\mathbf{x})$ and backward $p_{\text{TGT}}^{\text{AIS}}(\mathbf{z}_{0:T}|\mathbf{x})$ chains of length $T$ instead of the endpoint distributions only $q_\theta(\mathbf{z}|\mathbf{x})$ and $p(\mathbf{z}|\mathbf{x})$.

In particular, we construct an extended state space proposal by running $K$ independent AIS forward chains $\mathbf{z}_{0:T}^{(k)} \sim q_{\text{PROP}}^{\text{AIS}}$ in parallel. As in the IWAE upper bound (Eq. (5)), the extended state space target involves selecting a single index $s$ uniformly at random, and running a backward AIS chain $\mathbf{z}_{0:T}^{(s)} \sim p_{\text{TGT}}^{\text{AIS}}$ starting from a true posterior sample $\mathbf{z}_T \sim p(\mathbf{z}|\mathbf{x})$. The remaining $K-1$ samples are obtained by running forward AIS chains, as visualized in Fig. 2

$$q_{\text{PROP}}^{\text{IM-AIS}}(\mathbf{z}_{0:T}^{(1:K)}|\mathbf{x}) := \prod_{k=1}^{K} q_{\text{PROP}}^{\text{AIS}}(\mathbf{z}_{0:T}^{(k)}|\mathbf{x}), \quad p_{\text{TGT}}^{\text{IM-AIS}}(\mathbf{x}, \mathbf{z}_{0:T}^{(1:K)}) := \frac{1}{K} \sum_{s=1}^{K} p_{\text{TGT}}^{\text{AIS}}(\mathbf{x}, \mathbf{z}_{0:T}^{(s)}) \prod_{\substack{k=1 \\ k \neq s}}^{K} q_{\text{PROP}}^{\text{AIS}}(\mathbf{z}_{0:T}^{(k)}|\mathbf{x}).$$

where $q_{\text{PROP}}^{\text{AIS}}$ and $p_{\text{TGT}}^{\text{AIS}}$ were defined in Eq. (12). Expanding the log unnormalized density ratio,

$$\log \frac{p_{\text{TGT}}^{\text{IM-AIS}}(\mathbf{x}, \mathbf{z}_{0:T}^{(1:K)})}{q_{\text{PROP}}^{\text{IM-AIS}}(\mathbf{z}_{0:T}^{(1:K)}|\mathbf{x})} = \log \frac{\frac{1}{K} \sum_{s=1}^{K} p_{\text{TGT}}^{\text{AIS}}(\mathbf{x}, \mathbf{z}_{0:T}^{(s)}) \prod_{\substack{k=1 \\ k \neq s}}^{K} q_{\text{PROP}}^{\text{AIS}}(\mathbf{z}_{0:T}^{(k)}|\mathbf{x})}{\prod_{k=1}^{K} q_{\text{PROP}}^{\text{AIS}}(\mathbf{z}_{0:T}^{(k)}|\mathbf{x})} \tag{72}$$

$$= \log \frac{1}{K} \sum_{k=1}^{K} \frac{p_{\text{TGT}}^{\text{AIS}}(\mathbf{x}, \mathbf{z}_{0:T}^{(k)})}{q_{\text{PROP}}^{\text{AIS}}(\mathbf{z}_{0:T}^{(k)}|\mathbf{x})} \tag{73}$$

which is similar to the IWAE ratio but involves AIS chains. Taking the expectation under the proposal and target as in App. A recovers the lower and upper bounds in Eq. (14). The gap in the lower bound is $D_{\text{KL}}[q_{\text{PROP}}^{\text{IM-AIS}}(\mathbf{z}_{0:T}^{(1:K)}|\mathbf{x})\|p_{\text{TGT}}^{\text{IM-AIS}}(\mathbf{z}_{0:T}^{(1:K)}|\mathbf{x})]$ and the gap in the upper bound is $D_{\text{KL}}[p_{\text{TGT}}^{\text{IM-AIS}}(\mathbf{z}_{0:T}^{(1:K)}|\mathbf{x}))\|q_{\text{PROP}}^{\text{IM-AIS}}(\mathbf{z}_{0:T}^{(1:K)}|\mathbf{x})]$.

## E.2    PROOF OF LINEAR BIAS REDUCTION IN $T$ FOR IM-AIS

**Corollary E.1** (Complexity in $T$ for Independent Multi-Sample AIS Bound)**.** *Assuming perfect transitions and a geometric annealing path with linearly-spaced $\{\beta_t\}_{t=0}^{T}$, the sum of the gaps in the Independent Multi-Sample AIS sandwich bounds on MI, $I_{\text{IM-AIS}_U}(\pi_0, T) - I_{\text{IM-AIS}_L}(\pi_0, T)$, reduces linearly with increasing $T$.*

*Proof.* Using identical proof techniques as for IWAE (Burda et al. (2016); Sobolev & Vetrov (2019), Prop. B.1), we can show that our Independent Multi-Sample AIS bounds $\text{ELBO}_{\text{IM-AIS}}(\mathbf{x}; \pi_0, T, K-1) \leq \text{ELBO}_{\text{IM-AIS}}(\mathbf{x}; \pi_0, T, K)$ and $\text{EUBO}_{\text{IM-AIS}}(\mathbf{x}; \pi_0, T, K) \leq \text{EUBO}_{\text{IM-AIS}}(\mathbf{x}; \pi_0, T, K-1)$ improve with increasing $K$. Thus, the bias of our multi-sample bounds is less than the bias of the single-sample bounds, so the inequality in Eq. (68) and linear bias reduction in Prop. 3.1 also hold for Independent Multi-Sample AIS. We characterize this improvement in Prop. E.2 below. $\square$

### E.3 PROOF OF LOGARITHMIC IMPROVEMENT OF IM-AIS EUBO

**Proposition E.2** (Improvement of Independent Multi-Sample AIS over Single-Sample AIS). *Let* $p_{\text{TGT}}^{\text{IM-AIS}}(s|\mathbf{x}, \mathbf{z}_{0:T}^{(1:K)}) = \frac{p_{\text{TGT}}^{\text{AIS}}(\mathbf{x}, \mathbf{z}_{0:T}^{(s)})}{q_{\text{PROP}}^{\text{AIS}}(\mathbf{z}_{0:T}^{(s)}|\mathbf{x})} / \sum_{k=1}^{K} \frac{p_{\text{TGT}}^{\text{AIS}}(\mathbf{x}, \mathbf{z}_{0:T}^{(k)})}{q_{\text{PROP}}^{\text{AIS}}(\mathbf{z}_{0:T}^{(k)}|\mathbf{x})}$ *denote the normalized importance weights over* AIS *chains, and let* $\mathcal{U}(s)$ *indicate the uniform distribution over* $K$ *discrete values. Then, we can characterize the improvement of the Independent Multi-Sample* AIS *bounds on* $\log p(\mathbf{x})$, $\text{ELBO}_{\text{IM-AIS}}(\mathbf{x}; \pi_0, T, K)$ *and* $\text{EUBO}_{\text{IM-AIS}}(\mathbf{x}; \pi_0, T, K)$, *over the single-sample* AIS *bounds* $\text{ELBO}_{\text{AIS}}(\mathbf{x}; \pi_0, T)$ *and* $\text{EUBO}_{\text{AIS}}(\mathbf{x}; \pi_0, T)$ *using* KL *divergences, as follows*

$$\text{ELBO}_{\text{IM-AIS}}(\mathbf{x}; \pi_0, T, K) = \text{ELBO}_{\text{AIS}}(\mathbf{x}; \pi_0, T) + \underbrace{\mathbb{E}_{q_{\text{PROP}}^{\text{IM-AIS}}(\mathbf{z}_{0:T}^{(1:K)}|\mathbf{x})}\left[D_{\text{KL}}\left[\mathcal{U}(s)\big\|p_{\text{TGT}}^{\text{IM-AIS}}(s|\mathbf{z}_{0:T}^{(1:K)}, \mathbf{x})\right]\right]}_{0 \leq \text{ KL of uniform from SNIS weights} \leq D_{\text{KL}}[q_{\text{PROP}}^{\text{AIS}}(\mathbf{z}_{0:T}|\mathbf{x})\|p_{\text{TGT}}^{\text{AIS}}(\mathbf{z}_{0:T}|\mathbf{x})]},$$
(74)

$$\text{EUBO}_{\text{IM-AIS}}(\mathbf{x}; \pi_0, T, K) = \text{EUBO}_{\text{AIS}}(\mathbf{x}; \pi_0, T) - \underbrace{\mathbb{E}_{p_{\text{TGT}}^{\text{IM-AIS}}(\mathbf{z}_{0:T}^{(1:K)}|\mathbf{x})}\left[D_{\text{KL}}\left[p_{\text{TGT}}^{\text{IM-AIS}}(s|\mathbf{z}_{0:T}^{(1:K)}, \mathbf{x})\big\|\mathcal{U}(s)\right]\right]}_{0 \leq \text{ KL of SNIS weights from uniform} \leq \log K}.$$
(75)

*Proof.* The result follows directly from App. C.4 Prop. 2.1 by viewing Independent Multi-Sample AIS as IWAE with an AIS proposal as in App. E.1.

$\square$

## F REVERSE IWAE

In this section, we propose Reverse IWAE (RIWAE), which is an impractical alternative to standard IWAE. However, we use this as the basis for our Independent Reverse (App. G) and Coupled Reverse Multi-Sample AIS bounds (Sec. 3.3 and App. H).

### F.1 PROBABILISTIC INTERPRETATION AND BOUNDS

Similarly to simple reverse importance sampling, $\frac{1}{p(\mathbf{x})} = \mathbb{E}_{p(\mathbf{z}|\mathbf{x})}\left[\frac{q_\theta(\mathbf{z}|\mathbf{x})}{p(\mathbf{x},\mathbf{z})}\right]$, we consider $K$ independent posterior samples in an extended state space target distribution. The proposal distribution for our importance sampling scheme is a mixture of $K-1$ posterior distributions and one variational distribution,

$$q_{\text{PROP}}^{\text{RIWAE}}(\mathbf{x}, \mathbf{z}^{(1:K)}) = \frac{1}{K}\sum_{s=1}^{K} q_\theta(\mathbf{z}^{(s)}|\mathbf{x}) \prod_{\substack{k=1 \\ k \neq s}}^{K} p(\mathbf{x}, \mathbf{z}^{(k)}), \qquad p_{\text{TGT}}^{\text{RIWAE}}(\mathbf{x}, \mathbf{z}^{(1:K)}) = \prod_{k=1}^{K} p(\mathbf{x}, \mathbf{z}^{(k)}). \qquad (76)$$

Note that we have normalization constants of $\int p_{\text{TGT}}^{\text{RIWAE}}(\mathbf{x}, \mathbf{z}^{(1:K)})d\mathbf{z}^{(1:K)} = p(\mathbf{x})^K$ and $\int q_{\text{PROP}}^{\text{RIWAE}}(\mathbf{x}, \mathbf{z}^{(1:K)})d\mathbf{z}^{(1:K)} = p(\mathbf{x})^{K-1}$ since the transition kernels of the reverse chains do not change the normalization.

We visualize this sampling scheme in Fig. 4. Similarly to Eq. (25), the log unnormalized density ratio simplifies to

$$
\log \frac{p_{\text{TGT}}^{\text{RIWAE}}(\mathbf{x}, \mathbf{z}^{(1:K)})}{q_{\text{PROP}}^{\text{RIWAE}}(\mathbf{x}, \mathbf{z}^{(1:K)})} = \log \frac{\prod\limits_{k=1}^{K} p(\mathbf{x}, \mathbf{z}^{(k)})}{\frac{1}{K} \sum\limits_{s=1}^{K} q_\theta(\mathbf{z}^{(s)}|\mathbf{x}) \prod\limits_{\substack{k=1 \\ k \neq s}}^{K} p(\mathbf{x}, \mathbf{z}^{(k)})}
\tag{77}
$$

$$
= -\log \frac{1}{K} \sum_{k=1}^{K} \frac{q_\theta(\mathbf{z}^{(k)}|\mathbf{x})}{p(\mathbf{x}, \mathbf{z}^{(k)})}.
\tag{78}
$$

Taking the expectation under the proposal and target distributions yield lower and upper bounds on $\log p(\mathbf{x})$. These translate to upper and lower bounds on $\log p(\mathbf{x})$ which are different that standard IWAE bounds in Eq. (5),

$$
\mathbb{E}_{q_\theta(\mathbf{z}^{(1)}|\mathbf{x}) \prod\limits_{k=2}^{K} p(\mathbf{z}^{(k)}|\mathbf{x})} \left[ -\log \frac{1}{K} \sum_{k=1}^{K} \frac{q_\theta(\mathbf{z}^{(k)}|\mathbf{x})}{p(\mathbf{x}, \mathbf{z}^{(k)})} \right] \leq \log p(\mathbf{x}) \leq \mathbb{E}_{\prod\limits_{k=1}^{K} p(\mathbf{z}^{(k)}|\mathbf{x})} \left[ -\log \frac{1}{K} \sum_{k=1}^{K} \frac{q_\theta(\mathbf{z}^{(k)}|\mathbf{x})}{p(\mathbf{x}, \mathbf{z}^{(k)})} \right].
$$

Note these bounds are *impractical* as they would require more than one true posterior sample from $p(\mathbf{z}|\mathbf{x})$. However, we use them in Sec. 3.3 and App. H to derive practical multi-sample reverse AIS bounds.

## F.2 IMPROVEMENT OF RIWAE OVER ELBO AND EUBO

**Proposition F.1** (Improvement of Reverse IWAE over ELBO and EUBO). *Let* $q_{\text{PROP}}^{\text{RIWAE}}(s|\mathbf{x}, \mathbf{z}^{(1:K)}) = \frac{q_\theta(\mathbf{z}^{(s)}|\mathbf{x})}{p(\mathbf{x}, \mathbf{z}^{(s)}|\mathbf{x})} / \sum\limits_{k=1}^{K} \frac{q_\theta(\mathbf{z}^{(k)}|\mathbf{x})}{p(\mathbf{x}, \mathbf{z}^{(k)}|\mathbf{x})}$ *denote the normalized reverse importance sampling weights, or the posterior over the index variable in Eq. (76). Let* $\mathcal{U}(s)$ *indicate the uniform distribution over* $K$ *discrete values. Then, we can characterize the improvement of the Reverse IWAE bounds on* $\log p(\mathbf{x})$, $\text{ELBO}_{\text{RIWAE}}(\mathbf{x}; q_\theta, K)$ *and* $\text{EUBO}_{\text{RIWAE}}(\mathbf{x}; q_\theta, K)$, *over the single-sample bounds* $\text{ELBO}(\mathbf{x}; q_\theta)$ *and* $\text{EUBO}(\mathbf{x}; q_\theta)$ *using KL divergences, as follows*

$$
\text{ELBO}_{\text{RIWAE}}(\mathbf{x}; q_\theta, K) = \text{ELBO}(\mathbf{x}; q_\theta) + \underbrace{\mathbb{E}_{q_{\text{PROP}}^{\text{RIWAE}}(\mathbf{z}^{(1:K)}|\mathbf{x})} \left[ D_{\text{KL}} \left[ q_{\text{PROP}}^{\text{RIWAE}}(s|\mathbf{x}, \mathbf{z}^{(1:K)}) \big\| \mathcal{U}(s) \right] \right]}_{0 \leq \text{KL of uniform from SNIS weights} \leq \log K},
$$

$$
\text{EUBO}_{\text{RIWAE}}(\mathbf{x}; q_\theta, K) = \text{EUBO}(\mathbf{x}; q_\theta) - \underbrace{\mathbb{E}_{p_{\text{TGT}}^{\text{RIWAE}}(\mathbf{z}^{(1:K)}|\mathbf{x})} \left[ D_{\text{KL}} \left[ \mathcal{U}(s) \big\| q_{\text{PROP}}^{\text{RIWAE}}(s|\mathbf{x}, \mathbf{z}^{(1:K)}) \right] \right]}_{0 \leq \text{KL of SNIS weights from uniform} \leq D_{\text{KL}}[p(\mathbf{z}|\mathbf{x}) \| q_\theta(\mathbf{z}|\mathbf{x})]}.
$$

*Proof.* The proof follows similarly as Prop. B.2 or Lemma C.2. □

## G INDEPENDENT REVERSE MULTI-SAMPLE AIS

### G.1 PROBABILISTIC INTERPRETATION AND BOUNDS

In similar fashion to Reverse IWAE, we now use $K$ independent reverse AIS chains to form an extended state space target distribution $p_{\text{TGT}}^{\text{IR-AIS}}(\mathbf{z}_{0:T}^{(1:K)}, \mathbf{x})$, and a mixture proposal distribution $q_{\text{PROP}}^{\text{IR-AIS}}(\mathbf{z}_{0:T}^{(1:K)}, \mathbf{x})$ which includes a single forward AIS chain (see Fig. 2)

$$
q_{\text{PROP}}^{\text{IR-AIS}}(\mathbf{x}, \mathbf{z}_{0:T}^{(1:K)}) = \frac{1}{K} \sum_{s=1}^{K} q_{\text{PROP}}^{\text{AIS}}(\mathbf{z}_{0:T}^{(s)}|\mathbf{x}) \prod_{\substack{k=1 \\ k \neq s}}^{K} p_{\text{TGT}}^{\text{AIS}}(\mathbf{x}, \mathbf{z}_{0:T}^{(k)}),
$$

$$
p_{\text{TGT}}^{\text{IR-AIS}}(\mathbf{x}, \mathbf{z}_{0:T}^{(1:K)}) = \prod_{k=1}^{K} p_{\text{TGT}}^{\text{AIS}}(\mathbf{x}, \mathbf{z}_{0:T}^{(k)}).
\tag{79}
$$

Similarly to Eq. (77)-(78), the log unnormalized density ratio becomes

$$
\log \frac{p_{\text{TGT}}^{\text{IR-AIS}}(\mathbf{z}_{0:T}^{(1:K)}, \mathbf{x})}{q_{\text{PROP}}^{\text{IR-AIS}}(\mathbf{z}_{0:T}^{(1:K)}, \mathbf{x})} = \log \frac{\prod\limits_{k=1}^{K} p_{\text{TGT}}^{\text{AIS}}(\mathbf{x}, \mathbf{z}_{0:T}^{(k)})}{\frac{1}{K} \sum\limits_{k=1}^{K} q_{\text{PROP}}^{\text{AIS}}(\mathbf{z}_{0:T}^{(k)}|\mathbf{x}) \prod\limits_{t=1}^{T} p_{\text{TGT}}^{\text{AIS}}(\mathbf{x}, \mathbf{z}_{0:T}^{(k)})} \tag{80}
$$

$$
= -\log \frac{1}{K} \sum_{k=1}^{K} \frac{q_{\text{PROP}}^{\text{AIS}}(\mathbf{z}_{0:T}^{(k)}, \mathbf{x})}{p_{\text{TGT}}^{\text{AIS}}(\mathbf{x}, \mathbf{z}_{0:T}^{(k)})}. \tag{81}
$$

Taking expectations of the log unnormalized density ratio under the proposal and target, respectively, yield lower and upper bounds on $\log p(\mathbf{x})$

$$
\underbrace{\mathbb{E}_{\substack{\mathbf{z}_{0:T}^{(1)} \sim q_{\text{PROP}}^{\text{AIS}} \\ \mathbf{z}_{0:T}^{(2:K)} \sim p_{\text{TGT}}^{\text{AIS}}}} \left[ -\log \frac{1}{K} \sum_{k=1}^{K} \frac{q_{\text{PROP}}^{\text{AIS}}(\mathbf{z}_{0:T}^{(k)}|\mathbf{x})}{p_{\text{TGT}}^{\text{AIS}}(\mathbf{x}, \mathbf{z}_{0:T}^{(k)})} \right]}_{\text{ELBO}_{\text{IR-AIS}}(\mathbf{x}; \pi_0, K, T)} \leq \log p(\mathbf{x}) \leq \underbrace{\mathbb{E}_{\mathbf{z}_{0:T}^{(1:K)} \sim p_{\text{TGT}}^{\text{AIS}}} \left[ -\log \frac{1}{K} \sum_{k=1}^{K} \frac{q_{\text{PROP}}^{\text{AIS}}(\mathbf{z}_{0:T}^{(k)}|\mathbf{x})}{p_{\text{TGT}}^{\text{AIS}}(\mathbf{x}, \mathbf{z}_{0:T}^{(k)})} \right]}_{\text{EUBO}_{\text{IR-AIS}}(\mathbf{x}; \pi_0, K, T)}.
$$
$$\tag{82}$$

However, Independent Reverse Multi-Sample AIS may be impractical in common settings, since it is infeasible to have access to more than one true posterior sample.

## G.2 Proof of Logarithmic Improvement in $K$ for Independent Reverse AIS

**Proposition G.1** (Improvement of Independent Reverse Multi-Sample AIS over Single-Sample AIS).
*Let* $q_{\text{PROP}}^{\text{IR-AIS}}(s|\mathbf{x}, \mathbf{z}_{0:T}^{(1:K)}) = \frac{q_{\text{PROP}}^{\text{AIS}}(\mathbf{z}_{0:T}^{(s)}|\mathbf{x})}{p_{\text{TGT}}^{\text{AIS}}(\mathbf{x}, \mathbf{z}_{0:T}^{(s)})} / \sum\limits_{k=1}^{K} \frac{q_{\text{PROP}}^{\text{AIS}}(\mathbf{z}_{0:T}^{(k)}|\mathbf{x})}{p_{\text{TGT}}^{\text{AIS}}(\mathbf{x}, \mathbf{z}_{0:T}^{(k)})}$ *denote the normalized reverse importance sampling weights over* AIS *chains, and let* $\mathcal{U}(s)$ *indicate the uniform distribution over* $K$ *discrete values. Then, we can characterize the improvement of the Independent Reverse Multi-Sample* AIS *bounds on* $\log p(\mathbf{x})$, $\text{ELBO}_{\text{IR-AIS}}(\mathbf{x}; \pi_0, K, T)$ *and* $\text{EUBO}_{\text{IR-AIS}}(\mathbf{x}; \pi_0, K, T)$, *over the single-sample* $\text{ELBO}_{\text{AIS}}(\mathbf{x}; \pi_0, T)$ *and* $\text{ELBO}_{\text{AIS}}(\mathbf{x}; \pi_0, T)$ *using* KL *divergences, as follows*

$$
\text{ELBO}_{\text{IR-AIS}}(\mathbf{x}; \pi_0, K, T) = \text{ELBO}_{\text{AIS}}(\mathbf{x}; \pi_0, T) + \underbrace{\mathbb{E}_{q_{\text{PROP}}^{\text{IR-AIS}}(\mathbf{z}_{0:T}^{(1:K)}|\mathbf{x})} \left[ D_{\text{KL}}[q_{\text{PROP}}^{\text{IR-AIS}}(s|\mathbf{x}, \mathbf{z}_{0:T}^{(1:K)}) \| \mathcal{U}(s)] \right]}_{0 \leq \text{KL } of \text{ uniform from SNIS weights} \leq \log K},
$$

$$
\text{EUBO}_{\text{IR-AIS}}(\mathbf{x}; \pi_0, K, T) = \text{EUBO}_{\text{AIS}}(\mathbf{x}; \pi_0, T) - \underbrace{\mathbb{E}_{p_{\text{TGT}}^{\text{IR-AIS}}(\mathbf{z}_{0:T}^{(1:K)}|\mathbf{x})} \left[ D_{\text{KL}}[\mathcal{U}(s) \| q_{\text{PROP}}^{\text{IR-AIS}}(s|\mathbf{x}, \mathbf{z}_{0:T}^{(1:K)})] \right]}_{0 \leq \text{KL } of \text{ SNIS weights from uniform} \leq D_{\text{KL}}[p_{\text{TGT}}^{\text{AIS}}(\mathbf{z}_{0:T}|\mathbf{x}) \| q_{\text{PROP}}^{\text{AIS}}(\mathbf{z}_{0:T}|\mathbf{x})]}.
$$

*Proof.* The proof follows similarly as Prop. B.2 or Lemma C.2. □

## H  Coupled Reverse Multi-Sample AIS

### H.1  Probabilistic Interpretation and Bounds

The Coupled Reverse Multi-Sample AIS extended state space target distribution in Fig. 2 initializes $K$ backward chains from a *single* target sample $\mathbf{z}_T \sim \pi_T(\mathbf{z}|\mathbf{x})$, which makes the bound useful in practical situations. We denote the remaining transitions as $p_{\text{TGT}}^{\text{AIS}}(\mathbf{z}_{0:T-1}|\mathbf{z}_T, \mathbf{x})$, since they are identical to standard AIS in Eq. (12). Thus, the CR-AIS extended state space target distribution is

$$
p_{\text{TGT}}^{\text{CR-AIS}}(\mathbf{x}, \mathbf{z}_{0:T-1}^{(1:K)}, \mathbf{z}_T) = \pi_T(\mathbf{x}, \mathbf{z}_T) \prod_{k=1}^{K} p_{\text{TGT}}^{\text{AIS}}(\mathbf{z}_{0:T-1}^{(k)}|\mathbf{z}_T, \mathbf{x}). \tag{83}
$$

The extended state space proposal is obtained by selecting an index $s$ uniformly at random and running a single forward AIS chain. We then run $K - 1$ backward chains, all starting from the last

state of the selected forward chain,

$$q_{\text{PROP}}^{\text{CR-AIS}}(\mathbf{z}_{0:T-1}^{(1:K)}, \mathbf{z}_T|\mathbf{x}) := \frac{1}{K} \sum_{s=1}^{K} q_{\text{PROP}}^{\text{AIS}}(\mathbf{z}_{0:T-1}^{(s)}, \mathbf{z}_T|\mathbf{x}) \prod_{\substack{k=1 \\ k \neq s}}^{K} p_{\text{TGT}}^{\text{AIS}}(\mathbf{z}_{0:T-1}^{(k)}|\mathbf{z}_T, \mathbf{x}). \qquad (84)$$

See Fig. 2 or Fig. 4 for a graphical model description.

We can construct log-partition bounds using the log unnormalized density ratio

$$\log \frac{p_{\text{TGT}}^{\text{CR-AIS}}(\mathbf{z}_{0:T-1}^{(1:K)}, \mathbf{z}_T, \mathbf{x})}{q_{\text{PROP}}^{\text{CR-AIS}}(\mathbf{z}_{0:T-1}^{(1:K)}, \mathbf{z}_T|\mathbf{x})} = \log \frac{\pi_T(\mathbf{z}_T, \mathbf{x}) \prod\limits_{k=1}^{K} p_{\text{TGT}}^{\text{AIS}}(\mathbf{z}_{0:T-1}^{(k)}|\mathbf{z}_T, \mathbf{x})}{\frac{1}{K} \sum\limits_{s=1}^{K} q_{\text{PROP}}^{\text{AIS}}(\mathbf{z}_{0:T-1}^{(s)}, \mathbf{z}_T|\mathbf{x}) \prod\limits_{\substack{k=1 \\ k \neq s}}^{K} p_{\text{TGT}}^{\text{AIS}}(\mathbf{z}_{0:T-1}^{(k)}|\mathbf{z}_T)} \qquad (85)$$

$$= -\log \frac{1}{K} \sum_{k=1}^{K} \frac{q_{\text{PROP}}^{\text{AIS}}(\mathbf{z}_{0:T-1}^{(k)}, \mathbf{z}_T|\mathbf{x})}{p_{\text{TGT}}^{\text{AIS}}(\mathbf{z}_{0:T-1}^{(k)}|\mathbf{z}_T)\pi_T(\mathbf{z}_T, \mathbf{x})}. \qquad (86)$$

Taking the expected log ratio under the proposal and target yields lower and upper bounds on $\log p(\mathbf{x})$,

$$\text{ELBO}_{\text{CR-AIS}}(\mathbf{x}; \pi_0, K, T) := -\mathbb{E}_{\substack{\mathbf{z}_{0:T-1}^{(1)}, \mathbf{z}_T \sim q_{\text{PROP}}^{\text{AIS}}(\mathbf{z}_{0:T}|\mathbf{x}) \\ \mathbf{z}_{0:T-1}^{(2:K)} \sim p_{\text{TGT}}^{\text{AIS}}(\mathbf{z}_{0:T-1}|\mathbf{z}_T, \mathbf{x})}} \left[ \log \frac{1}{K} \sum_{k=1}^{K} \frac{q_{\text{PROP}}^{\text{AIS}}(\mathbf{z}_{0:T-1}^{(k)}, \mathbf{z}_T|\mathbf{x})}{p_{\text{TGT}}^{\text{AIS}}(\mathbf{x}, \mathbf{z}_{0:T-1}^{(k)}, \mathbf{z}_T)} \right] \leq \log p(\mathbf{x})$$

$$\text{EUBO}_{\text{CR-AIS}}(\mathbf{x}; \pi_0, K, T) := -\mathbb{E}_{\substack{\mathbf{z}_T \sim \pi_T(\mathbf{z}_T|\mathbf{x}) \\ \mathbf{z}_{0:T-1}^{(1:K)} \sim p_{\text{TGT}}^{\text{AIS}}(\mathbf{z}_{0:T-1}|\mathbf{z}_T, \mathbf{x})}} \left[ \log \frac{1}{K} \sum_{k=1}^{K} \frac{q_{\text{PROP}}^{\text{AIS}}(\mathbf{z}_{0:T-1}^{(k)}, \mathbf{z}_T|\mathbf{x})}{p_{\text{TGT}}^{\text{AIS}}(\mathbf{x}, \mathbf{z}_{0:T-1}^{(k)}, \mathbf{z}_T)} \right] \geq \log p(\mathbf{x}).$$

## H.2 PROOF THAT CR-AIS BOUNDS TIGHTEN WITH INCREASING $K$

In this section, we prove that Coupled Reverse Multi-Sample AIS bounds get tighter with increasing $K$. Our proof provides an alternative perspective to Burda et al. (2016); Sobolev & Vetrov (2019) for showing the monotonic improvement of IWAE or Independent Multi-Sample AIS with $K$. We will also characterize the improvement of CR-AIS bounds over single-sample AIS bounds in Prop. H.2, as a direct consequence of this lemma.

**Lemma H.1** (CR-AIS Bounds Tighten with Increasing $K$). *Coupled Reverse Multi-Sample* AIS *bounds get tighter with increasing number of samples $K$. In other words, for any $K > 1$,*

$$\text{ELBO}_{\text{CR-AIS}}(\mathbf{x}; \pi_0, T, K-1) \leq \text{ELBO}_{\text{CR-AIS}}(\mathbf{x}; \pi_0, T, K) \leq \log p(\mathbf{x}), \qquad (87)$$

$$\text{EUBO}_{\text{CR-AIS}}(\mathbf{x}; \pi_0, T, K-1) \geq \text{EUBO}_{\text{CR-AIS}}(\mathbf{x}; \pi_0, T, K) \geq \log p(\mathbf{x}). \qquad (88)$$

*Proof.* Our proof will proceed by introducing an additional set of $M$ index variables $s_{1:M}$ into the $K$-sample probabilistic interpretation of CR-AIS in Eq. (83)-(84), with $M < K$. We will show that the KL divergence in this *joint* state space (including $s_{1:M}$) is equal to the gap of the $M$-sample $\text{ELBO}_{\text{CR-AIS}}(\mathbf{x}; \pi_0, T, M)$ or $\text{EUBO}_{\text{CR-AIS}}(\mathbf{x}; \pi_0, T, M)$. We then show that marginalizing over $s_{1:M}$ yields the gap of the $K$-sample $\text{ELBO}_{\text{CR-AIS}}(\mathbf{x}; \pi_0, T, K)$ or $\text{EUBO}_{\text{CR-AIS}}(\mathbf{x}; \pi_0, T, K)$. Since marginalization cannot increase the KL divergence, we will have shown that for any $M < K$, $D_{\text{KL}}[q_{\text{PROP}}^{\text{CR-AIS}}(\mathbf{z}_{0:T-1}^{(1:K)}, \mathbf{z}_T|\mathbf{x})\|p_{\text{TGT}}^{\text{CR-AIS}}(\mathbf{z}_{0:T-1}^{(1:K)}, \mathbf{z}_T|\mathbf{x})]] \leq D_{\text{KL}}[q_{\text{PROP}}^{\text{CR-AIS}}(\mathbf{z}_{0:T-1}^{(1:M)}, \mathbf{z}_T|\mathbf{x})\|p_{\text{TGT}}^{\text{CR-AIS}}(\mathbf{z}_{0:T-1}^{(1:M)}, \mathbf{z}_T|\mathbf{x})]]$ and thus $\text{ELBO}_{\text{CR-AIS}}(\mathbf{x}; \pi_0, T, M) \leq \text{ELBO}_{\text{CR-AIS}}(\mathbf{x}; \pi_0, T, K)$. Identical reasoning holds for the $\text{EUBO}_{\text{CR-AIS}}$.

**Sub-Sampling Probabilistic Interpretation** Let $\mathcal{U}(s_1, ..., s_M)$ indicate the probability of drawing $M < K$ sample indices uniformly without replacement (i.e. each $s_m$ is distinct). In the CR-AIS target distribution, there is no distinction between the indices $\{s_{1:M}\}$ and $k \notin \{s_{1:M}\}$ after drawing $\mathbf{z}_T \sim \pi_T(\mathbf{x}, \mathbf{z})$. We can write

$$p_{\text{TGT}}^{\text{CR-AIS}}(\mathbf{x}, s_{1:M}, \mathbf{z}_{0:T-1}^{(1:K)}, \mathbf{z}_T) = \mathcal{U}(s_1, ..., s_M) \cdot \pi_T(\mathbf{x}, \mathbf{z}_T) \prod_{k=1}^{K} p_{\text{TGT}}^{\text{AIS}}(\mathbf{z}_{0:T-1}^{(k)}|\mathbf{z}_T, \mathbf{x}), \qquad (89)$$

which, after marginalization over $s_{1:M}$, clearly matches $p_{\text{TGT}}^{\text{CR-AIS}}(\mathbf{x}, \mathbf{z}_{0:T-1}^{(1:K)}, \mathbf{z}_T)$.

We also draw $s_{1:M} \sim \mathcal{U}(s_1, \dots, s_M)$ for the extended state space proposal. Next, we select an index $m$ uniformly at random from $\{1, \dots, M\}$, which is used to to specify which chain $\mathbf{z}_{0:T-1}^{(s_m)}$ is run in the forward direction to obtain $\mathbf{z}_T$, as in Eq. (84). After marginalizing over $m$, we obtain the following mixture proposal distribution

$$q_{\text{PROP}}^{\text{CR-AIS}}(s_{1:M}, \mathbf{z}_{0:T-1}^{(1:K)}, \mathbf{z}_T | \mathbf{x}) := \mathcal{U}(s_1, \dots, s_M) \cdot \left( \frac{1}{M} \sum_{m=1}^{M} q_{\text{PROP}}^{\text{AIS}}(\mathbf{z}_{0:T-1}^{(s_m)}, \mathbf{z}_T | \mathbf{x}) \prod_{\substack{j=1 \\ j \neq m}}^{M} p_{\text{TGT}}^{\text{AIS}}(\mathbf{z}_{0:T-1}^{(s_j)} | \mathbf{z}_T, \mathbf{x}) \right)$$

$$\cdot \prod_{\substack{k=1 \\ k \notin \{s_{1:M}\}}}^{K} p_{\text{TGT}}^{\text{AIS}}(\mathbf{z}_{0:T-1}^{(k)} | \mathbf{z}_T, \mathbf{x}). \tag{90}$$

We will consider marginalizing over $s_{1:M}$ below, but first write the KL divergence in the extended state space which includes $s_{1:M}$. For example, the forward KL divergence matches the gap of the $M$-sample $\text{ELBO}_{\text{CR-AIS}}(\mathbf{x}; \pi_0, T, M)$,

$$D_{\text{KL}}[q_{\text{PROP}}^{\text{CR-AIS}}(s_{1:M}, \mathbf{z}_{0:T-1}^{(1:K)}, \mathbf{z}_T | \mathbf{x}) \| p_{\text{TGT}}^{\text{CR-AIS}}(s_{1:M}, \mathbf{z}_{0:T-1}^{(1:K)}, \mathbf{z}_T | \mathbf{x})] \tag{91}$$

$$= \mathbb{E}_{q_{\text{PROP}}^{\text{CR-AIS}}} \left[ \log \frac{\cancel{\mathcal{U}(s_{1:M})} \cdot \left( \frac{1}{M} \sum_{m=1}^{M} q_{\text{PROP}}^{\text{AIS}}(\mathbf{z}_{0:T-1}^{(s_m)}, \mathbf{z}_T | \mathbf{x}) \prod_{\substack{j=1 \\ j \neq m}}^{M} p_{\text{TGT}}^{\text{AIS}}(\mathbf{z}_{0:T-1}^{(s_j)} | \mathbf{z}_T, \mathbf{x}) \right) \cdot \cancel{\prod_{\substack{k=1 \\ k \notin \{s_{1:M}\}}}^{K} p_{\text{TGT}}^{\text{AIS}}(\mathbf{z}_{0:T-1}^{(k)} | \mathbf{z}_T, \mathbf{x})}}{\cancel{\mathcal{U}(s_{1:M})} \cdot \pi_T(\mathbf{x}, \mathbf{z}_T) \prod_{k \in \{s_{1:M}\}} p_{\text{TGT}}^{\text{AIS}}(\mathbf{z}_{0:T-1}^{(k)} | \mathbf{z}_T, \mathbf{x}) \cancel{\prod_{\substack{k=1 \\ k \notin \{s_{1:M}\}}}^{K} p_{\text{TGT}}^{\text{AIS}}(\mathbf{z}_{0:T-1}^{(k)} | \mathbf{z}_T, \mathbf{x})}} \right]$$

$$= D_{\text{KL}} \left[ q_{\text{PROP}}^{\text{CR-AIS}}(\mathbf{z}_{0:T-1}^{(1:M)}, \mathbf{z}_T | \mathbf{x}) \| p_{\text{TGT}}^{\text{CR-AIS}}(\mathbf{z}_{0:T-1}^{(1:M)}, \mathbf{z}_T | \mathbf{x}) \right], \tag{92}$$

which matches the $M$-sample probabilistic interpretation of CR-AIS from App. H.1. Identical reasoning holds for the case of $\text{EUBO}_{\text{CR-AIS}}(\mathbf{x}; \pi_0, T, K) \leq \text{EUBO}_{\text{CR-AIS}}(\mathbf{x}; \pi_0, T, M)$ using the reverse KL divergence.

**Marginalization over** $s_{1:M}$ We have already seen from Eq. (89) that the marginal $\sum_{s_{1:M}} p_{\text{TGT}}^{\text{CR-AIS}}(\mathbf{x}, s_{1:M}, \mathbf{z}_{0:T-1}^{(1:K)}, \mathbf{z}_T)$ matches the $K$-sample target distribution $p_{\text{TGT}}^{\text{CR-AIS}}(\mathbf{x}, \mathbf{z}_{0:T-1}^{(1:K)}, \mathbf{z}_T)$.

We would now like to marginalize over $s_{1:M}$ in $q_{\text{PROP}}^{\text{CR-AIS}}(s_{1:M}, \mathbf{z}_{0:T-1}^{(1:K)}, \mathbf{z}_T | \mathbf{x})$. Combining the two product terms in Eq. (90),

$$\sum_{s_{1:M}} q_{\text{PROP}}^{\text{CR-AIS}}(s_{1:M}, \mathbf{z}_{0:T-1}^{(1:K)}, \mathbf{z}_T | \mathbf{x}) = \mathbb{E}_{\mathcal{U}(s_{1:M})} \left[ \frac{1}{M} \sum_{m=1}^{M} q_{\text{PROP}}^{\text{AIS}}(\mathbf{z}_{0:T-1}^{(s_m)}, \mathbf{z}_T | \mathbf{x}) \prod_{\substack{k=1 \\ k \neq s_m}}^{K} p_{\text{TGT}}^{\text{AIS}}(\mathbf{z}_{0:T-1}^{(k)} | \mathbf{z}_T, \mathbf{x}) \right]$$

$$\tag{93}$$

$$= \frac{1}{M} \sum_{m=1}^{M} \mathbb{E}_{\mathcal{U}(s_{1:M})} \left[ q_{\text{PROP}}^{\text{AIS}}(\mathbf{z}_{0:T-1}^{(s_m)}, \mathbf{z}_T | \mathbf{x}) \prod_{\substack{k=1 \\ k \neq s_m}}^{K} p_{\text{TGT}}^{\text{AIS}}(\mathbf{z}_{0:T-1}^{(k)} | \mathbf{z}_T, \mathbf{x}) \right]$$

$$\tag{94}$$

$$\overset{(1)}{=} \frac{1}{M} \sum_{m=1}^{M} \mathbb{E}_{\mathcal{U}(s_m)} \left[ q_{\text{PROP}}^{\text{AIS}}(\mathbf{z}_{0:T-1}^{(s_m)}, \mathbf{z}_T | \mathbf{x}) \prod_{\substack{k=1 \\ k \neq s_m}}^{K} p_{\text{TGT}}^{\text{AIS}}(\mathbf{z}_{0:T-1}^{(k)} | \mathbf{z}_T, \mathbf{x}) \right] \tag{95}$$

$$\overset{(2)}{=} \frac{1}{M} \sum_{m=1}^{M} \frac{1}{K} \sum_{j=1}^{K} \left[ q_{\text{PROP}}^{\text{AIS}}(\mathbf{z}_{0:T-1}^{(j)}, \mathbf{z}_T | \mathbf{x}) \prod_{\substack{k=1 \\ k \neq j}}^{K} p_{\text{TGT}}^{\text{AIS}}(\mathbf{z}_{0:T-1}^{(k)} | \mathbf{z}_T, \mathbf{x}) \right] \tag{96}$$

$$= \frac{1}{K} \sum_{j=1}^{K} \left[ q_{\text{PROP}}^{\text{AIS}}(\mathbf{z}_{0:T-1}^{(j)}, \mathbf{z}_T | \mathbf{x}) \prod_{\substack{k=1 \\ k \neq j}}^{K} p_{\text{TGT}}^{\text{AIS}}(\mathbf{z}_{0:T-1}^{(k)} | \mathbf{z}_T, \mathbf{x}) \right], \tag{97}$$

where in (1), we can write the marginal $\mathcal{U}(s_m)$ since the terms inside expectation do not explicitly depend on the other indices in $s_{1:M}$. In (2), we use the fact that the marginal over any $s_m$ is uniform.

Thus, we have shown that the marginal distributions match the standard $K$-sample target and proposal distributions of CR-AIS, with

$$D_{\text{KL}}\Big[\sum_{s_{1:M}} q_{\text{PROP}}^{\text{CR-AIS}}(s_{1:M}, \mathbf{z}_{0:T-1}^{(1:K)}, \mathbf{z}_T|\mathbf{x}) \Big\| \sum_{s_{1:M}} p_{\text{TGT}}^{\text{CR-AIS}}(s_{1:M}, \mathbf{z}_{0:T-1}^{(1:K)}, \mathbf{z}_T|\mathbf{x})\Big] = \underbrace{D_{\text{KL}}[q_{\text{PROP}}^{\text{CR-AIS}}(\mathbf{z}_{0:T-1}^{(1:K)}, \mathbf{z}_T|\mathbf{x})\|p_{\text{TGT}}^{\text{CR-AIS}}(\mathbf{z}_{0:T-1}^{(1:K)}, \mathbf{z}_T|\mathbf{x})]}_{\text{gap of ELBO}_{\text{CR-AIS}}(\mathbf{x}; \pi_0, T, K)}$$

and similar reasoning for the EUBO using the reverse KL divergence.

**Conclusion of Proof**   Since marginalization can not increase the KL divergence, we have

$$\underbrace{D_{\text{KL}}[q_{\text{PROP}}^{\text{CR-AIS}}(\mathbf{z}_{0:T-1}^{(1:K)}, \mathbf{z}_T|\mathbf{x})\|p_{\text{TGT}}^{\text{CR-AIS}}(\mathbf{z}_{0:T-1}^{(1:K)}, \mathbf{z}_T|\mathbf{x})]}_{\text{gap of ELBO}_{\text{CR-AIS}}(\mathbf{x}; \pi_0, T, K)} \le \underbrace{D_{\text{KL}}[q_{\text{PROP}}^{\text{CR-AIS}}(s_{1:M}, \mathbf{z}_{0:T-1}^{(1:K)}, \mathbf{z}_T|\mathbf{x})\|p_{\text{TGT}}^{\text{CR-AIS}}(s_{1:M}, \mathbf{z}_{0:T-1}^{(1:K)}, \mathbf{z}_T|\mathbf{x})]}_{\text{gap of ELBO}_{\text{CR-AIS}}(\mathbf{x}; \pi_0, T, M) \text{ (Eq. (92))}}.$$

(98)

This shows that $\text{ELBO}_{\text{CR-AIS}}(\mathbf{x}; \pi_0, T, K)$ is tighter than $\text{ELBO}_{\text{CR-AIS}}(\mathbf{x}; \pi_0, T, M)$, with $\text{ELBO}_{\text{CR-AIS}}(\mathbf{x}; \pi_0, T, K) \ge \text{ELBO}_{\text{CR-AIS}}(\mathbf{x}; \pi_0, T, M)$. Identical reasoning holds for the case of $\text{EUBO}_{\text{CR-AIS}}(\mathbf{x}; \pi_0, T, K) \le \text{EUBO}_{\text{CR-AIS}}(\mathbf{x}; \pi_0, T, M)$ using the reverse KL divergence. Choosing $M = K - 1$ proves that the bounds tighten as $K$ increases, as desired. □

**Characterizing the Improvement of $K$-Sample over $M$-Sample CR-AIS**   Note that we can explicitly write the gap of the inequality Eq. (98) using the chain rule for joint probability, for example $q_{\text{PROP}}^{\text{CR-AIS}}(s_{1:M}, \mathbf{z}_{0:T-1}^{(1:K)}, \mathbf{z}_T|\mathbf{x}) = q_{\text{PROP}}^{\text{CR-AIS}}(\mathbf{z}_{0:T-1}^{(1:K)}, \mathbf{z}_T|\mathbf{x}) q_{\text{PROP}}^{\text{CR-AIS}}(s_{1:M}|\mathbf{z}_{0:T-1}^{(1:K)}, \mathbf{z}_T, \mathbf{x})$.

The gap in Eq. (98) also corresponds to the gap between $\text{ELBO}_{\text{CR-AIS}}(\mathbf{x}; \pi_0, T, K)$ and $\text{ELBO}_{\text{CR-AIS}}(\mathbf{x}; \pi_0, T, M)$, so we can write

$$\text{ELBO}_{\text{CR-AIS}}(\mathbf{x}; \pi_0, T, K) - \text{ELBO}_{\text{CR-AIS}}(\mathbf{x}; \pi_0, T, M)$$
$$= D_{\text{KL}}[q_{\text{PROP}}^{\text{CR-AIS}}(s_{1:M}|\mathbf{z}_{0:T-1}^{(1:K)}, \mathbf{z}_T, \mathbf{x})\|p_{\text{TGT}}^{\text{CR-AIS}}(s_{1:M}|\mathbf{z}_{0:T-1}^{(1:K)}, \mathbf{z}_T, \mathbf{x})]. \quad (99)$$

Below, we analyze the posterior over the index variable for $M = 1$ as a special case. This allows us to characterize the gap between $\text{ELBO}_{\text{CR-AIS}}(\mathbf{x}; \pi_0, T, K)$ and the single-sample $\text{ELBO}_{\text{AIS}}(\mathbf{x}; \pi_0, T) = \text{ELBO}_{\text{CR-AIS}}(\mathbf{x}; \pi_0, T, K = 1)$.

### H.3   PROOF OF LOGARITHMIC IMPROVEMENT IN $K$ FOR CR-AIS ELBO

**Proposition H.2** (Improvement of Coupled Reverse Multi-Sample AIS over Single-Sample AIS). *Let*

$$q_{\text{PROP}}^{\text{CR-AIS}}(s|\mathbf{x}, \mathbf{z}_{0:T-1}^{(1:K)}, \mathbf{z}_T) = \frac{\dfrac{q_{\text{PROP}}^{\text{AIS}}(\mathbf{z}_{0:T-1}^{(s)}, \mathbf{z}_T|\mathbf{x})}{p_{\text{TGT}}^{\text{AIS}}(\mathbf{x}, \mathbf{z}_{0:T-1}^{(s)}|\mathbf{z}_T)}}{\displaystyle\sum_{k=1}^{K} \frac{q_{\text{PROP}}^{\text{AIS}}(\mathbf{z}_{0:T-1}^{(k)}, \mathbf{z}_T|\mathbf{x})}{p_{\text{TGT}}^{\text{AIS}}(\mathbf{x}, \mathbf{z}_{0:T-1}^{(k)}|\mathbf{z}_T)}} \quad (100)$$

*denote the normalized importance weights over the AIS chains used in Coupled Reverse Multi-Sample AIS, and let $\mathcal{U}(s)$ indicate the uniform distribution over $K$ discrete values. Then, we can characterize the improvement of the Coupled Reverse Multi-Sample AIS bounds on $\log p(\mathbf{x})$, $\text{ELBO}_{\text{CR-AIS}}(\mathbf{x}; \pi_0, T, K)$ and $\text{EUBO}_{\text{CR-AIS}}(\mathbf{x}; \pi_0, T, K)$, over the single-sample AIS bounds $\text{ELBO}_{\text{AIS}}(\mathbf{x}; \pi_0, T)$ and $\text{EUBO}_{\text{AIS}}(\mathbf{x}; \pi_0, T)$ using KL divergences, as follows*

$$\text{ELBO}_{\text{CR-AIS}}(\mathbf{x}; \pi_0, T, K) = \text{ELBO}_{\text{AIS}}(\mathbf{x}; \pi_0, T) + \underbrace{\mathbb{E}_{q_{\text{PROP}}^{\text{CR-AIS}}(\mathbf{z}_{0:T-1}^{(1:K)}, \mathbf{z}_T|\mathbf{x})}\Big[D_{\text{KL}}\big[q_{\text{PROP}}^{\text{CR-AIS}}(s|\mathbf{x}, \mathbf{z}_{0:T-1}^{(1:K)}, \mathbf{z}_T)\big\|\mathcal{U}(s)\big]\Big]}_{0 \le \text{KL of uniform from SNIS weights} \le \log K},$$

$$\text{EUBO}_{\text{CR-AIS}}(\mathbf{x}; \pi_0, T, K) = \text{EUBO}_{\text{AIS}}(\mathbf{x}; \pi_0, T) - \underbrace{\mathbb{E}_{p_{\text{TGT}}^{\text{CR-AIS}}(\mathbf{z}_{0:T-1}^{(1:K)}, \mathbf{z}_T|\mathbf{x})}\Big[D_{\text{KL}}\big[\mathcal{U}(s)\big\|q_{\text{PROP}}^{\text{CR-AIS}}(s|\mathbf{x}, \mathbf{z}_{0:T-1}^{(1:K)}, \mathbf{z}_T)\big]\Big]}_{0 \le \text{KL of SNIS weights from uniform} \le D_{\text{KL}}[p_{\text{TGT}}^{\text{AIS}}(\mathbf{z}_{0:T}|\mathbf{x})\|q_{\text{PROP}}^{\text{AIS}}(\mathbf{z}_{0:T}|\mathbf{x})]}.$$

*Proof.* Using $M = 1$ in Eq. (99) above, we obtain

$$\text{ELBO}_{\text{CR-AIS}}(\mathbf{x}; \pi_0, T, K) - \text{ELBO}_{\text{AIS}}(\mathbf{x}; \pi_0, T, M = 1)$$
$$= D_{\text{KL}}\big[q_{\text{PROP}}^{\text{CR-AIS}}(s|\mathbf{z}_{0:T-1}^{(1:K)}, \mathbf{z}_T, \mathbf{x})\big\|p_{\text{TGT}}^{\text{CR-AIS}}(s|\mathbf{z}_{0:T-1}^{(1:K)}, \mathbf{z}_T, \mathbf{x})\big] \quad (101)$$

From Eq. (90), we can write the posterior over the index variable as

$$q_{\text{PROP}}^{\text{CR-AIS}}(s|\mathbf{z}_{0:T-1}^{(1:K)}, \mathbf{z}_T, \mathbf{x}) = \frac{\frac{q_{\text{PROP}}^{\text{AIS}}(\mathbf{z}_{0:T-1}^{(s)}, \mathbf{z}_T|\mathbf{x})}{p_{\text{TGT}}^{\text{AIS}}(\mathbf{x}, \mathbf{z}_{0:T-1}^{(s)}|\mathbf{z}_T)}}{\sum_{k=1}^{K} \frac{q_{\text{PROP}}^{\text{AIS}}(\mathbf{z}_{0:T-1}^{(k)}, \mathbf{z}_T|\mathbf{x})}{p_{\text{TGT}}^{\text{AIS}}(\mathbf{x}, \mathbf{z}_{0:T-1}^{(k)}|\mathbf{z}_T)}} \tag{102}$$

For the target distribution in Eq. (89), the posterior $p_{\text{TGT}}^{\text{CR-AIS}}(s|\mathbf{z}_{0:T-1}^{(1:K)}, \mathbf{z}_T, \mathbf{x}) = \mathcal{U}(s)$ is uniform. Thus, we can characterize the improvement of the $K$-sample CR-AIS ELBO over its single-chain AIS

$$\text{ELBO}_{\text{CR-AIS}}(\mathbf{x}; \pi_0, T, K) - \text{ELBO}_{\text{AIS}}(\mathbf{x}; \pi_0, T) = \mathbb{E}_{q_{\text{PROP}}^{\text{CR-AIS}}(\mathbf{z}_{0:T-1}^{(1:K)}, \mathbf{z}_T|\mathbf{x})} \left[ D_{\text{KL}}[q_{\text{PROP}}^{\text{CR-AIS}}(s|\mathbf{x}, \mathbf{z}_{0:T-1}^{(1:K)}, \mathbf{z}_T) \| \mathcal{U}(s)] \right]. \tag{103}$$

The proof follows identically for the EUBO.

For the ELBO, the KL divergence with the uniform distribution in the second argument is bounded above by $\log K$. For the EUBO, the improvement of CR-AIS is limited by the gap in the single-chain $\text{EUBO}_{\text{AIS}}(\mathbf{x}; \pi_0, T)$, which corresponds to $D_{\text{KL}}[p_{\text{TGT}}^{\text{AIS}}(\mathbf{z}_{0:T}|\mathbf{x}) \| q_{\text{PROP}}^{\text{AIS}}(\mathbf{z}_{0:T}|\mathbf{x})]$. ☐

### H.4 PROOF OF LINEAR BIAS REDUCTION IN $T$ FOR CR-AIS

**Corollary H.3** (Complexity in $T$ for Coupled Reverse Multi-Sample AIS Bound). *Assuming perfect transitions and a geometric annealing path with linearly-spaced $\{\beta_t\}_{t=0}^{T}$, the sum of the gaps in the Coupled Reverse Multi-Sample AIS sandwich bounds on MI, $I_{\text{CR-AIS}_U}(\pi_0, K, T) - I_{\text{CR-AIS}_L}(\pi_0, K, T)$, reduces linearly with increasing $T$.*

*Proof.* Since we have shown in Prop. H.2 that $\text{ELBO}_{\text{CR-AIS}}(\pi_0, K, T)$ and $\text{EUBO}_{\text{CR-AIS}}(\pi_0, K, T)$ are tighter than $\text{ELBO}_{\text{AIS}}(\pi_0, T)$ and $\text{EUBO}_{\text{AIS}}(\pi_0, T)$, the bias in the CR-AIS sandwich bounds $\text{EUBO}_{\text{CR-AIS}}(\pi_0, K, T) - \text{ELBO}_{\text{CR-AIS}}(\pi_0, K, T)$ will be less than that of single-sample AIS. Thus, inequality in Eq. (68) and linear bias reduction in Prop. 3.1 (see App. D.1) also hold for CR-AIS, which implies linear bias reduction under perfect transitions and linear scheduling. ☐

## I COMPARISON OF MULTI-SAMPLE AIS BOUNDS

In Fig. 5, we compare the performance of our various Multi-Sample AIS bounds for MI estimation of a Linear VAE with 10 latent variables and random weights, and VAE and GAN models with 20 latent variables trained on MNIST.

To obtain an upper bound on MI, we recommend using the Independent Multi-Sample AIS ELBO for log partition function estimation. This corresponds to the forward direction of BDMC and achieves the best performance in all cases. This upper bound uses independent samples and is not limited to $\log K$ improvement, in contrast to the Coupled Reverse Multi-Sample AIS upper bound on MI (Prop. H.2).

The results are less conclusive for the Multi-Sample AIS lower bounds on MI, where either the Independent Multi-Sample AIS EUBO or Coupled Reverse AIS EUBO may be preferable for log partition function estimation. Recall that these bounds have different sources of stochasticity that provide improvement over single-chain AIS. The stochasticity in the Independent Multi-Sample AIS lower bound on MI comes from $K - 1$ independent negative forward chains which, by App. E.3 Prop. E.2, can only lead to $\log K$ improvement over single-sample AIS. However, these gains are easily attained for low values of $T$. For example, with two total AIS distributions, which corresponds to simple importance sampling with $\pi_0(\mathbf{z}) = p(\mathbf{z})$, the Independent Multi-Sample AIS lower bound on MI reduces to Structured INFO-NCE and saturates to $\log K$. This may be useful for quickly estimating values of MI at a similar order of magnitude as $\log K$.

The stochasticity in the Coupled Reverse AIS lower bound on MI is induced by MCMC transitions in $K$ coupled backward chains. While this does not formally limit the improvement over single-sample AIS, we see in Fig. 5 that at least moderate values of $T$ may be needed to match or marginally improve upon Independent Multi-Sample AIS. These observations suggest that the preferred lower bounds on MI may vary based on the scale of the true MI and the amount of computation available.

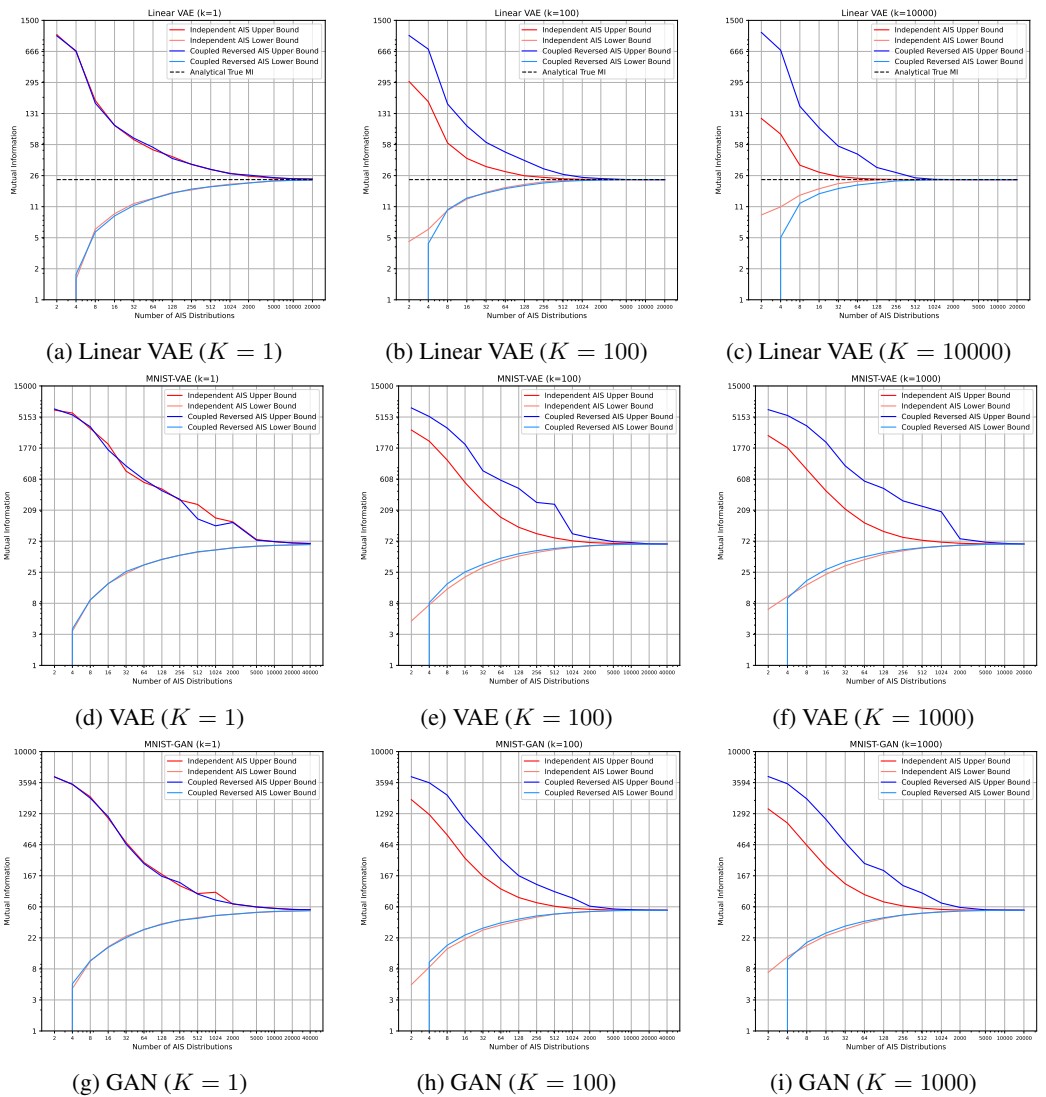

Figure 5: Comparing Multi-Sample AIS sandwich bounds and evaluating the effect of $K$ and $T$ for MI estimation in deep generative models on MNIST.

## J  GENERALIZED MUTUAL INFORMATION NEURAL ESTIMATION (GMINE)

In this section, we provide probabilistic interpretations for the Mutual Information Neural Estimation (MINE) lower bounds on MI from Belghazi et al. (2018), which allows us to derive novel *Generalized* MINE bounds. In a similar spirit to GIWAE, Generalized MINE uses a base variational distribution $q_\theta(\mathbf{z}|\mathbf{x})$ to tighten the MINE-DV or MINE-F bound and can be evaluated when $p(\mathbf{z})$ is available.

See Fig. 6 for a summary of Generalized MINE bounds and their relationships. We discuss probabilistic interpretations in this section, and provide complementary interpretations in terms of conjugate dual representations of the KL divergence in App. K.

We begin by deriving a probabilistic interpretation for the IBAL lower bound in Sec. 4, which we will show is closely related to the probabilistic interpretations of MINE. Our MINE-AIS method is designed to optimize and evaluate the IBAL lower bound on MI, with detailed discussion in Sec. 4 and App. M.

$$I(\mathbf{x}; \mathbf{z}) = \mathbb{E}_{p(\mathbf{x},\mathbf{z})}\left[\log \frac{q_\theta(\mathbf{z}|\mathbf{x})}{p(\mathbf{z})}\right] + \mathbb{E}_{p(\mathbf{x})}\left[D_{\mathrm{KL}}[p(\mathbf{z}|\mathbf{x})||q_\theta(\mathbf{z}|\mathbf{x})]\right]$$

$\mathbb{E}\left[D_{\mathrm{KL}}\left[p(\mathbf{z}|\mathbf{x})\|\frac{1}{Z(\mathbf{x})}q_\theta(\mathbf{z}|\mathbf{x})e^{T_\phi(\mathbf{x},\mathbf{z})}\right]\right]$

$$\mathrm{IBAL}(q_\theta, T_\phi) = \mathbb{E}_{p(\mathbf{x},\mathbf{z})}\left[\log \frac{q_\theta(\mathbf{z}|\mathbf{x})}{p(\mathbf{z})}\right] + \mathbb{E}_{p(\mathbf{x},\mathbf{z})}\left[T_\phi(\mathbf{x},\mathbf{z})\right] - \mathbb{E}_{p(\mathbf{x})}\left[\log\left[\mathbb{E}_{q_\theta(\mathbf{z}|\mathbf{x})}e^{T_\phi(\mathbf{x},\mathbf{z})}\right]\right]$$

$D_{\mathrm{KL}}[p(\mathbf{x})\|\pi_{\theta,\phi}(\mathbf{x})]$

$D_{\mathrm{KL}}\left[p(\mathbf{x},\mathbf{z})\|\frac{1}{Z}p(\mathbf{x})q_\theta(\mathbf{z}|\mathbf{x})e^{T_\phi(\mathbf{x},\mathbf{z})}\right]$

$$I_{\mathrm{GMINE\text{-}DV}}(q_\theta, T_\phi) = \mathbb{E}_{p(\mathbf{x},\mathbf{z})}\left[\log \frac{q_\theta(\mathbf{z}|\mathbf{x})}{p(\mathbf{z})}\right] + \mathbb{E}_{p(\mathbf{x},\mathbf{z})}\left[T_\phi(\mathbf{x},\mathbf{z})\right] - \log \mathbb{E}_{p(\mathbf{x})q_\theta(\mathbf{z}|\mathbf{x})}\left[e^{T_\phi(\mathbf{x},\mathbf{z})}\right]$$

$\mathbb{E}\left[D_{\mathrm{GKL}}\left[\frac{1}{Z(\mathbf{x})}q_\theta(\mathbf{z}|\mathbf{x})e^{T_\phi(\mathbf{x},\mathbf{z})}\|q_\theta(\mathbf{z}|\mathbf{x})e^{T_\phi(\mathbf{x},\mathbf{z})-1}\right]\right]$

$D_{\mathrm{GKL}}\left[\frac{1}{Z}p(\mathbf{x})q_\theta(\mathbf{z}|\mathbf{x})e^{T_\phi(\mathbf{x},\mathbf{z})}\|p(\mathbf{x})q_\theta(\mathbf{z}|\mathbf{x})e^{T_\phi(\mathbf{x},\mathbf{z})-1}\right]$

$\mathbb{E}\left[D_{\mathrm{GKL}}\left[p(\mathbf{z}|\mathbf{x})\|q_\theta(\mathbf{z}|\mathbf{x})e^{T_\phi(\mathbf{x},\mathbf{z})-1}\right]\right]$

$$I_{\mathrm{GMINE\text{-}F}}(q_\theta, T_\phi) = \mathbb{E}_{p(\mathbf{x},\mathbf{z})}\left[\log \frac{q_\theta(\mathbf{z}|\mathbf{x})}{p(\mathbf{z})}\right] + \mathbb{E}_{p(\mathbf{x},\mathbf{z})}\left[T_\phi(\mathbf{x},\mathbf{z})\right] - \mathbb{E}_{p(\mathbf{x})q_\theta(\mathbf{z}|\mathbf{x})}\left[e^{T_\phi(\mathbf{x},\mathbf{z})-1}\right]$$

$\mathbb{E}\left[D_{\mathrm{KL}}[p(\mathbf{z}|\mathbf{x})\|q_\theta(\mathbf{z}|\mathbf{x})]\right]$

$$I_{\mathrm{BA}}(q_\theta) = \mathbb{E}_{p(\mathbf{x},\mathbf{z})}\left[\log \frac{q_\theta(\mathbf{z}|\mathbf{x})}{p(\mathbf{z})}\right]$$

Figure 6: Generalized Energy Based Bounds. Arrows indicate the gaps in each MI lower bound or its relationship to other lower bounds. $D_{\mathrm{GKL}}(\cdot\|\cdot)$ represents the *generalized* KL divergence between two unnormalized densities (see App. J.3). All bounds are written in terms of a base variational distribution $q_\theta(\mathbf{z}|\mathbf{x})$, which may be chosen to be the marginal $p(\mathbf{z})$ as in MINE-DV and MINE-F.

### J.1 PROBABILISTIC INTERPRETATION OF IBAL

For an energy-based posterior approximation

$$\pi_{\theta,\phi}(\mathbf{z}|\mathbf{x}) := \frac{1}{\mathcal{Z}_{\theta,\phi}(\mathbf{x})}q_\theta(\mathbf{z}|\mathbf{x})e^{T_\phi(\mathbf{x},\mathbf{z})}, \quad \text{where} \quad \mathcal{Z}_{\theta,\phi}(\mathbf{x}) = \mathbb{E}_{q_\theta(\mathbf{z}|\mathbf{x})}\left[e^{T_\phi(\mathbf{x},\mathbf{z})}\right], \quad (104)$$

we consider the BA lower bound on MI,

$$I_{\mathrm{BA}_L}(\pi_{\theta,\phi}) = I(\mathbf{x},\mathbf{z}) - \mathbb{E}_{p(\mathbf{x})}\left[D_{\mathrm{KL}}[p(\mathbf{z}|\mathbf{x})\|\pi_{\theta,\phi}(\mathbf{z}|\mathbf{x})]\right] \quad (105)$$

$$= \mathbb{E}_{p(\mathbf{x},\mathbf{z})}\left[\log \frac{p(\mathbf{x},\mathbf{z})}{p(\mathbf{x})p(\mathbf{z})}\right] - \mathbb{E}_{p(\mathbf{x},\mathbf{z})}\left[\log p(\mathbf{z}|\mathbf{x})\right] + \mathbb{E}_{p(\mathbf{x},\mathbf{z})}\left[T(\mathbf{x},\mathbf{z})\right] - \mathbb{E}_{p(\mathbf{x},\mathbf{z})}\left[\log \mathcal{Z}_{\theta,\phi}\right]$$

$$= \mathbb{E}_{p(\mathbf{x},\mathbf{z})}\left[\log \frac{q_\theta(\mathbf{z}|\mathbf{x})}{p(\mathbf{z})}\right] + \mathbb{E}_{p(\mathbf{x},\mathbf{z})}\left[T_\phi(\mathbf{x},\mathbf{z})\right] - \mathbb{E}_{p(\mathbf{x})}\left[\log \mathcal{Z}_{\theta,\phi}(\mathbf{x})\right]$$

$$= \underbrace{\mathbb{E}_{p(\mathbf{x},\mathbf{z})}\left[\log \frac{q_\theta(\mathbf{z}|\mathbf{x})}{p(\mathbf{z})}\right]}_{I_{\mathrm{BA}_L}(q_\theta)} + \underbrace{\mathbb{E}_{p(\mathbf{x},\mathbf{z})}\left[\log \frac{e^{T_\phi(\mathbf{x},\mathbf{z})}}{\mathbb{E}_{q_\theta(\mathbf{z}|\mathbf{x})}\left[e^{T_\phi(\mathbf{x},\mathbf{z})}\right]}\right]}_{\text{contrastive term}}$$

$$=: \mathrm{IBAL}(q_\theta, T_\phi),$$

where the gap in the mutual information bound is $\mathbb{E}_{p(\mathbf{x})}\left[D_{\mathrm{KL}}[p(\mathbf{z}|\mathbf{x})\|\pi_{\theta,\phi}(\mathbf{z}|\mathbf{x})]\right]$. Note that $\mathrm{IBAL}(q_\theta, T_\phi)$ includes $I_{\mathrm{BA}_L}(q_\theta)$ as one of its terms, where we refer to $q_\theta(\mathbf{z}|\mathbf{x})$ as the base variational distribution. We visualize relationships between various energy-based bounds in Fig. 3b and Fig. 6.

**Proposition 4.1.** *For a given $q_\theta(\mathbf{z}|\mathbf{x})$, the optimal IBAL critic function equals the log importance weights up to a constant $T^*(\mathbf{x},\mathbf{z}) = \log \frac{p(\mathbf{x},\mathbf{z})}{q_\theta(\mathbf{z}|\mathbf{x})} + c(\mathbf{x})$. For this $T^*$, we have $\mathrm{IBAL}(q_\theta, T^*) = I(\mathbf{x};\mathbf{z})$.*

See App. L.1 for the proof.

### J.2 PROBABILISTIC INTERPRETATION OF GENERALIZED MINE-DV

We can interpret the MINE-DV bound of Belghazi et al. (2018) as arising from an energy-based variational approximation $\pi_{\theta,\phi}(\mathbf{x},\mathbf{z})$ of the *full joint* distribution $p(\mathbf{x},\mathbf{z})$ as follows

$$\pi_{\theta,\phi}(\mathbf{x},\mathbf{z}) := \frac{1}{\mathcal{Z}}p(\mathbf{x})q_\theta(\mathbf{z}|\mathbf{x})e^{T_\phi(\mathbf{x},\mathbf{z})}, \quad \text{where} \quad \mathcal{Z} = \mathbb{E}_{p(\mathbf{x})q_\theta(\mathbf{z}|\mathbf{x})}\left[e^{T_\phi(\mathbf{x},\mathbf{z})}\right], \quad (106)$$

$q_\theta(\mathbf{z}|\mathbf{x})$ is a base variational distribution, and $T_\phi$ is a critic or negative energy function.

Note that for the induced marginal $\pi_{\theta,\phi}(\mathbf{x}) := \int \pi_{\theta,\phi}(\mathbf{x}, \mathbf{z}) d\mathbf{z} \neq p(\mathbf{x})$ due to the contribution of the critic function. Instead, we have

$$\pi_{\theta,\phi}(\mathbf{x}) = \frac{1}{\mathcal{Z}} p(\mathbf{x}) \mathcal{Z}(\mathbf{x}), \tag{107}$$

where $\mathcal{Z}(\mathbf{x}) = \int q(\mathbf{z}|\mathbf{x}) e^{T_\phi(\mathbf{x},\mathbf{z})} d\mathbf{z}$ and $\mathcal{Z} = \int p(\mathbf{x}) q(\mathbf{z}|\mathbf{x}) e^{T_\phi(\mathbf{x},\mathbf{z})} d\mathbf{x} d\mathbf{z} = \mathbb{E}_{p(\mathbf{x})}[\mathcal{Z}(\mathbf{x})]$.

Subtracting the joint KL divergence $D_{\mathrm{KL}}[p(\mathbf{x}, \mathbf{z}) \| \pi_{\theta,\phi}(\mathbf{x}, \mathbf{z})]$ from $I(\mathbf{x}; \mathbf{z})$, we obtain the *Generalized* MINE-DV lower bound on MI

$$I(\mathbf{x}; \mathbf{z}) \geq I(\mathbf{x}; \mathbf{z}) - D_{\mathrm{KL}}\left[p(\mathbf{x}, \mathbf{z}) \middle\| \frac{1}{\mathcal{Z}} p(\mathbf{x}) q_\theta(\mathbf{z}|\mathbf{x}) e^{T_\phi(\mathbf{x},\mathbf{z})}\right] \tag{108}$$

$$= \mathbb{E}_{p(\mathbf{x},\mathbf{z})}\left[\log \frac{q_\theta(\mathbf{z}|\mathbf{x})}{p(\mathbf{z})}\right] + \mathbb{E}_{p(\mathbf{x},\mathbf{z})}\left[T_\phi(\mathbf{x}, \mathbf{z})\right] - \log \mathbb{E}_{p(\mathbf{x}) q_\theta(\mathbf{z}|\mathbf{x})}\left[e^{T_\phi(\mathbf{x},\mathbf{z})}\right] \tag{109}$$

$$=: I_{\mathrm{GMINE\text{-}DV}}(q_\theta, T_\phi).$$

By construction, Eq. (108) shows that the gap in Generalized MINE-DV is $D_{\mathrm{KL}}[p(\mathbf{x}, \mathbf{z}) \| \frac{1}{\mathcal{Z}} p(\mathbf{x}) q_\theta(\mathbf{z}|\mathbf{x}) e^{T_\phi(\mathbf{x},\mathbf{z})}]$, which has a probabilistic interpretation in terms of the approximate joint distribution $\pi_{\theta,\phi}(\mathbf{x}, \mathbf{z})$.

**Relationship with MINE-DV**   For $q_\theta(\mathbf{z}|\mathbf{x}) = p(\mathbf{z})$, we obtain the MINE-DV bound (Belghazi et al., 2018; Poole et al., 2019) as a special case with $I_{\mathrm{MINE\text{-}DV}}(T_\phi) = I_{\mathrm{GMINE\text{-}DV}}(p(\mathbf{z}), T_\phi)$. In particular, the joint base distribution in Eq. (106) corresponds to the product of marginals $p(\mathbf{x}) p(\mathbf{z})$. Our probabilistic interpretation shows that the gap in MINE-DV corresponds to $D_{\mathrm{KL}}[p(\mathbf{x}, \mathbf{z}) \| \frac{1}{\mathcal{Z}} p(\mathbf{x}) p(\mathbf{z}) e^{T_\phi(\mathbf{x},\mathbf{z})}]$.

We expect that the Generalized MINE, with a learned variational distribution $q_\theta(\mathbf{z}|\mathbf{x})$, to obtain tighter bounds than MINE-DV. We can guarantee that $I_{\mathrm{GMINE\text{-}DV}}(q_\theta^*, T_\phi) \geq I_{\mathrm{MINE\text{-}DV}}(T_\phi)$ for the optimal $q_\theta^*$ with a given $T_\phi$, so long as $p(\mathbf{z})$ is in the variational family (i.e., $\exists \theta_0$ such that $q_{\theta_0}(\mathbf{z}|\mathbf{x}) = p(\mathbf{z})$).

**Relationship with BA Bound**   Choosing a constant critic function $T_{\phi_0}(\mathbf{x}, \mathbf{z}) = \text{const}$, we can see that $\pi_{\theta,\phi_0}(\mathbf{x}, \mathbf{z}) = p(\mathbf{x}) q_\theta(\mathbf{z}|\mathbf{x})$ and $I_{\mathrm{GMINE\text{-}DV}}(q_\theta, T_{\phi_0} = \text{const}) = I_{\mathrm{BA}_L}(q_\theta)$ for a given $q_\theta(\mathbf{z}|\mathbf{x})$.

**Optimal Critic Function**   For a given $q_\theta(\mathbf{z}|\mathbf{x})$, the optimal critic function of Generalized MINE-DV corresponds to the log importance weight between the target $p(\mathbf{x}, \mathbf{z})$ and the joint base distribution $p(\mathbf{x}) q_\theta(\mathbf{z}|\mathbf{x})$, plus a constant

$$T^*(\mathbf{x}, \mathbf{z}) = \log \frac{p(\mathbf{x}, \mathbf{z})}{p(\mathbf{x}) q_\theta(\mathbf{z}|\mathbf{x})} + c. \tag{110}$$

For the optimal critic function associated with a given $q_\theta(\mathbf{z}|\mathbf{x})$, we have $I_{\mathrm{MINE\text{-}DV}}(q_\theta, T^*) = I(\mathbf{x}; \mathbf{z})$.

**Relationship with IBAL**   We can use our probabilistic interpretation to show that the IBAL is tighter than Generalized MINE-DV, with $\mathrm{IBAL}(q_\theta, T_\phi) \geq I_{\mathrm{GMINE\text{-}DV}}(q_\theta, T_\phi)$. Subtracting the gaps in the bounds in Eq. (105) and Eq. (108),

$$\mathrm{IBAL}(q_\theta, T_\phi) = I_{\mathrm{GMINE\text{-}DV}}(q_\theta, T_\phi) + \underbrace{D_{\mathrm{KL}}[p(\mathbf{x}, \mathbf{z}) \| \pi_{\theta,\phi}(\mathbf{x}, \mathbf{z})]}_{\text{gap in GMINE-DV}(q_\theta, T_\phi)} - \underbrace{\mathbb{E}_{p(\mathbf{x})}\left[D_{\mathrm{KL}}[p(\mathbf{z}|\mathbf{x}) \| \pi_{\theta,\phi}(\mathbf{z}|\mathbf{x})]\right]}_{\text{gap in IBAL}(q_\theta, T_\phi)}$$

$$= I_{\mathrm{GMINE\text{-}DV}}(q_\theta, T_\phi) + \mathbb{E}_{p(\mathbf{x})}\left[\log \frac{p(\mathbf{x})}{\pi_{\theta,\phi}(\mathbf{x})}\right] \tag{111}$$

$$= I_{\mathrm{GMINE\text{-}DV}}(q_\theta, T_\phi) + D_{\mathrm{KL}}[p(\mathbf{x}) \| \pi_{\theta,\phi}(\mathbf{x})] \tag{112}$$

$$\geq I_{\mathrm{GMINE\text{-}DV}}(q_\theta, T_\phi). \tag{113}$$

We see that the difference between $\mathrm{IBAL}(q_\theta, T_\phi)$ and $I_{\mathrm{GMINE\text{-}DV}}(q_\theta, T_\phi)$ corresponds to a marginal KL divergence in the $\mathbf{x}$ space, where $\pi_{\theta,\phi}(\mathbf{x})$ is defined in Eq. (107).

As in Poole et al. (2019), we can also interpret the gap between the IBAL$(q_\theta, T_\phi)$ and $I_{\text{GMINE-DV}}(q_\theta, T_\phi)$ as an application of Jensen's inequality, with

$$D_{\text{KL}}[p(\mathbf{x})\|\pi_{\theta,\phi}(\mathbf{x})] = D_{\text{KL}}\left[p(\mathbf{x})\middle\|\frac{1}{\mathcal{Z}}p(\mathbf{x})\mathcal{Z}(\mathbf{x})\right] \tag{114}$$

$$= \log \mathcal{Z} - \mathbb{E}_{p(\mathbf{x})}[\log \mathcal{Z}(\mathbf{x})] \tag{115}$$

$$= \log \mathbb{E}_{p(\mathbf{x})}[\mathcal{Z}(\mathbf{x})] - \mathbb{E}_{p(\mathbf{x})}[\log \mathcal{Z}(\mathbf{x})], \tag{116}$$

since $\log \mathbb{E}_{p(\mathbf{x})}[\mathcal{Z}(\mathbf{x})] \geq \mathbb{E}_{p(\mathbf{x})}[\log \mathcal{Z}(\mathbf{x})]$.

## J.3 PROBABILISTIC INTERPRETATION OF GENERALIZED MINE-F

The BA lower bound and its corresponding EUBO upper bound on $\log p(\mathbf{x})$ are derived using a KL divergence between the true and approximate posterior, which are normalized conditional distributions over $\mathbf{z}$ given $\mathbf{x}$. In this section, we interpret the MINE-F bound (Belghazi et al., 2018) as arising from a generalized notion of the KL divergence between possibly *unnormalized* density functions. In particular, our probabilistic interpretation will involve an unnormalized $\tilde{\pi}_{\theta,\phi}(\mathbf{z}|\mathbf{x})$ which seeks to approximate the true (normalized) posterior $p(\mathbf{z}|\mathbf{x})$.

**Generalized KL Divergence (GKL)** First, we state the definition of the *generalized* KL divergence, which takes unnormalized measures as input arguments and corresponds to the limiting behavior of the $\alpha$-divergence (Cichocki & Amari (2010))

$$D_{\text{GKL}}[\tilde{r}(\mathbf{z})\|\tilde{s}(\mathbf{z})] = \int \tilde{r}(\mathbf{z})\log\frac{\tilde{r}(\mathbf{z})}{\tilde{s}(\mathbf{z})}d\mathbf{z} - \int \tilde{r}(\mathbf{z})d\mathbf{z} + \int \tilde{s}(\mathbf{z})d\mathbf{z}. \tag{117}$$

As in the case of the standard KL divergence, this quantity is nonnegative, convex in either argument, and vanishes for $\tilde{r}(\mathbf{z}) = \tilde{s}(\mathbf{z})$. It is also a member of both the family of Bregman divergences and $f$-divergences (Amari, 2009). If $r(\mathbf{z})$ and $s(\mathbf{z})$ are normalized, then $D_{\text{GKL}}[r(\mathbf{z})\|s(\mathbf{z})] = D_{\text{KL}}[r(\mathbf{z})\|s(\mathbf{z})]$. Finally, if $r(\mathbf{z})$ is normalized and $\tilde{s}(\mathbf{z})$ is unnormalized, one can easily confirm that

$$D_{\text{GKL}}[r(\mathbf{z})\|\tilde{s}(\mathbf{z})] = D_{\text{KL}}[r(\mathbf{z})\|s(\mathbf{z})] + D_{\text{GKL}}[s(\mathbf{z})\|\tilde{s}(\mathbf{z})] \tag{118}$$
$$\geq D_{\text{KL}}[r(\mathbf{z})\|s(\mathbf{z})].$$

**Generalized EUBO (GEUBO)** Since the Generalized KL divergence is always nonnegative, we define a Generalized EUBO by adding the Generalized KL divergence between the normalized true posterior and an unnormalized approximate posterior $\tilde{\pi}(\mathbf{z}|\mathbf{x})$.

$$\text{GEUBO}(\mathbf{x}; \tilde{\pi}) := \log p(\mathbf{x}) + D_{\text{GKL}}[p(\mathbf{z}|\mathbf{x})\|\tilde{\pi}(\mathbf{z}|\mathbf{x})] \tag{119}$$

$$= \mathbb{E}_{p(\mathbf{z}|\mathbf{x})}\left[\log\frac{p(\mathbf{x},\mathbf{z})}{\tilde{\pi}(\mathbf{z}|\mathbf{x})}\right] - 1 + \mathcal{Z}_{\tilde{\pi}}(\mathbf{x}), \tag{120}$$

where we define $\mathcal{Z}_{\tilde{\pi}}(\mathbf{x}) := \int \tilde{\pi}(\mathbf{z}|\mathbf{x})d\mathbf{z}$. In general, we have $\text{EUBO}(\mathbf{x}; \pi) \leq \text{GEUBO}(\mathbf{x}; \tilde{\pi})$ where $\pi$ is the normalized distribution of $\tilde{\pi}$, with equality if $\tilde{\pi}$ is normalized.

**Generalized BA (GBA)** Using the Generalized EUBO in place of $\log p(\mathbf{x})$, we obtain the following lower bound on MI, which we denote as the Generalized BA lower bound

$$I(\mathbf{x}; \mathbf{z}) \geq I(\mathbf{x}; \mathbf{z}) - \mathbb{E}_{p(\mathbf{x})}\left[D_{\text{GKL}}[p(\mathbf{z}|\mathbf{x})\|\tilde{\pi}(\mathbf{z}|\mathbf{x})]\right]$$

$$= \mathbb{E}_{p(\mathbf{x},\mathbf{z})}\left[\log\frac{\tilde{\pi}(\mathbf{z}|\mathbf{x})}{p(\mathbf{z})}\right] + 1 - \mathbb{E}_{p(\mathbf{x})}\left[\mathcal{Z}_{\tilde{\pi}}(\mathbf{x})\right] \tag{121}$$

$$=: I_{\text{GBA}_L}(\tilde{\pi}).$$

**Generalized MINE-F (GMINE-F)** We now consider an unnormalized, energy-based approximation to the true posterior, involving a base variational distribution $q_\theta(\mathbf{z}|\mathbf{x})$ and a learned critic function $T_\phi(\mathbf{x}, \mathbf{z})$

$$\tilde{\pi}_{\theta,\phi}(\mathbf{z}|\mathbf{x}) := q_\theta(\mathbf{z}|\mathbf{x})e^{T_\phi(\mathbf{x},\mathbf{z})-1}. \tag{122}$$

Using this unnormalized approximate posterior in the GBA lower bound in Eq. (121), we obtain the *Generalized* MINE-F lower bound

$$I(\mathbf{x}; \mathbf{z}) \geq I(\mathbf{x}; \mathbf{z}) - \mathbb{E}_{p(\mathbf{x})}\left[D_{\text{GKL}}[p(\mathbf{z}|\mathbf{x}) \| \tilde{\pi}_{\theta,\phi}(\mathbf{z}|\mathbf{x})]\right] \tag{123}$$

$$= \mathbb{E}_{p(\mathbf{x},\mathbf{z})}\left[\log \frac{q_\theta(\mathbf{z}|\mathbf{x})}{p(\mathbf{z})}\right] + \mathbb{E}_{p(\mathbf{x},\mathbf{z})}\left[T_\phi(\mathbf{x},\mathbf{z})\right] - \mathbb{E}_{p(\mathbf{x})q_\theta(\mathbf{z}|\mathbf{x})}\left[e^{T_\phi(\mathbf{x},\mathbf{z})-1}\right] \tag{124}$$

$$=: I_{\text{GMINE-F}}(q_\theta, T_\phi). \tag{125}$$

By construction, we can see that the gap in $I_{\text{GMINE-F}}(q_\theta, T_\phi)$ is equal to the Generalized KL divergence $\mathbb{E}_{p(\mathbf{x})}\left[D_{\text{GKL}}[p(\mathbf{z}|\mathbf{x}) \| \tilde{\pi}_{\theta,\phi}(\mathbf{z}|\mathbf{x})]\right]$ in Eq. (123). As in the case of MINE-DV, we obtain the standard MINE-F lower bound $I_{\text{MINE-F}}(T_\phi) = I_{\text{GMINE-F}}(p(\mathbf{z}), T_\phi)$ when using the marginal $p(\mathbf{z})$ as the proposal.

**Optimal Critic Function** The optimal critic function of Generalized MINE-F is $T^*(\mathbf{x}, \mathbf{z}) = 1 + \log \frac{p(\mathbf{x},\mathbf{z})}{p(\mathbf{x})q(\mathbf{z}|\mathbf{x})}$ (Poole et al., 2019). In this case, we obtain $I_{\text{GMINE-F}}(q_\theta, T^*) = I(\mathbf{x}; \mathbf{z})$ in Eq. (124).

**Relationship with IBAL** We would now like to relate the gap in $I_{\text{GMINE-F}}(q_\theta, T_\phi)$ to the gap in $I_{\text{IBAL}}(q_\theta, T_\phi)$. First, note that the normalized distribution corresponding to $\tilde{\pi}_{\theta,\phi}(\mathbf{z}|\mathbf{x}) = q_\theta(\mathbf{z}|\mathbf{x})e^{T_\phi(\mathbf{x},\mathbf{z})-1}$ matches the the energy-based posterior in the IBAL, $\pi_{\theta,\phi}(\mathbf{z}|\mathbf{x}) = \frac{1}{\mathcal{Z}(\mathbf{x})}q_\theta(\mathbf{z}|\mathbf{x})e^{T_\phi(\mathbf{x},\mathbf{z})}$. Using Eq. (118), we have

$$\mathbb{E}_{p(\mathbf{x})}\left[D_{\text{GKL}}[p(\mathbf{z}|\mathbf{x}) \| \tilde{\pi}_{\theta,\phi}(\mathbf{z}|\mathbf{x})]\right] = \mathbb{E}_{p(\mathbf{x})}\left[D_{\text{KL}}[p(\mathbf{z}|\mathbf{x}) \| \pi_{\theta,\phi}(\mathbf{z}|\mathbf{x})]\right] + \mathbb{E}_{p(\mathbf{x})}\left[D_{\text{GKL}}[\pi_{\theta,\phi}(\mathbf{z}|\mathbf{x}) \| \tilde{\pi}_{\theta,\phi}(\mathbf{z}|\mathbf{x})]\right].$$

and substituting in the definition of each term

$$\underbrace{\mathbb{E}_{p(\mathbf{x})}\left[D_{\text{GKL}}\left[p(\mathbf{z}|\mathbf{x})\Big\|q_\theta(\mathbf{z}|\mathbf{x})e^{T_\phi(\mathbf{x},\mathbf{z})-1}\right]\right]}_{\text{gap of GMINE-F}} = \underbrace{\mathbb{E}_{p(\mathbf{x})}\left[D_{\text{KL}}\left[p(\mathbf{z}|\mathbf{x})\Big\|\frac{1}{\mathcal{Z}(\mathbf{x})}q_\theta(\mathbf{z}|\mathbf{x})e^{T_\phi(\mathbf{x},\mathbf{z})}\right]\right]}_{\text{gap of IBAL}} \tag{126}$$
$$+ \underbrace{\mathbb{E}_{p(\mathbf{x})}\left[D_{\text{GKL}}\left[\frac{1}{\mathcal{Z}(\mathbf{x})}q_\theta(\mathbf{z}|\mathbf{x})e^{T_\phi(\mathbf{x},\mathbf{z})}\Big\|q_\theta(\mathbf{z}|\mathbf{x})e^{T_\phi(\mathbf{x},\mathbf{z})-1}\right]\right]}_{\geq 0},$$

where the final term $D_{\text{GKL}}[\pi_{\theta,\phi}(\mathbf{z}|\mathbf{x}) \| \tilde{\pi}_{\theta,\phi}(\mathbf{z}|\mathbf{x})] \geq 0$ is nonnegative because it is a generalized KL divergence. Thus, we have

$$\text{IBAL}(q_\theta, T_\phi) = I_{\text{GMINE-F}}(q_\theta, T_\phi) + \mathbb{E}_{p(\mathbf{x})}\left[D_{\text{GKL}}[\pi_{\theta,\phi}(\mathbf{z}|\mathbf{x}) \| \tilde{\pi}_{\theta,\phi}(\mathbf{z}|\mathbf{x})]\right] \geq I_{\text{GMINE-F}}(q_\theta, T_\phi). \tag{127}$$

Alternatively, Poole et al. (2019) use the inequality $\log u \leq u - 1$ to show that

$$D_{\text{GKL}}[\pi_{\theta,\phi}(\mathbf{z}|\mathbf{x}) \| \tilde{\pi}_{\theta,\phi}(\mathbf{z}|\mathbf{x})] = \mathbb{E}_{p(\mathbf{x})}\left[\mathcal{Z}(\mathbf{x}) - 1 - \log \mathcal{Z}(\mathbf{x})\right] \geq 0.$$

We visualize the relationship between IBAL and Generalized MINE-F in Fig. 6.

**Relationship with Generalized MINE-DV** To characterize the gap between Generalized MINE-DV and Generalized MINE-F, we can again use the equality from Eq. (118), but this time using divergences over joint distributions.

$$D_{\text{GKL}}\left[p(\mathbf{x},\mathbf{z})\big\|\tilde{\pi}_{\theta,\phi}(\mathbf{x},\mathbf{z})\right] = D_{\text{KL}}\left[p(\mathbf{x},\mathbf{z})\big\|\pi_{\theta,\phi}(\mathbf{x},\mathbf{z})\right] + D_{\text{GKL}}\left[\pi_{\theta,\phi}(\mathbf{x},\mathbf{z})\big\|\tilde{\pi}_{\theta,\phi}(\mathbf{x},\mathbf{z})\right]$$

$$\underbrace{D_{\text{GKL}}\left[p(\mathbf{x},\mathbf{z})\Big\|p(\mathbf{x})q_\theta(\mathbf{z}|\mathbf{x})e^{T_\phi(\mathbf{x},\mathbf{z})-1}\right]}_{\text{gap in GMINE-F}} = \underbrace{D_{\text{KL}}\left[p(\mathbf{x},\mathbf{z})\Big\|\frac{1}{\mathcal{Z}}p(\mathbf{x})q_\theta(\mathbf{z}|\mathbf{x})e^{T_\phi(\mathbf{x},\mathbf{z})}\right]}_{\text{gap in GMINE-DV}} \tag{128}$$
$$+ \underbrace{D_{\text{GKL}}\left[\frac{1}{\mathcal{Z}}p(\mathbf{x})q_\theta(\mathbf{z}|\mathbf{x})e^{T_\phi(\mathbf{x},\mathbf{z})}\Big\|p(\mathbf{x})q_\theta(\mathbf{z}|\mathbf{x})e^{T_\phi(\mathbf{x},\mathbf{z})-1}\right]}_{\geq 0}.$$

In this case, we have $D_{\text{GKL}}[\frac{1}{\mathcal{Z}}p(\mathbf{x})q_\theta(\mathbf{z}|\mathbf{x})e^{T_\phi(\mathbf{x},\mathbf{z})}\|p(\mathbf{x})q_\theta(\mathbf{z}|\mathbf{x})e^{T_\phi(\mathbf{x},\mathbf{z})-1}] \geq 0$. Thus, Generalized MINE-DV is tighter than Generalized MINE-F (see Fig. 6), which generalizes the finding in Poole et al. (2019) that standard MINE-DV is tighter than standard MINE-F.

# K  CONJUGATE DUALITY INTERPRETATIONS

In this section, we interpret the energy-based MI lower bounds in MINE-AIS, Generalized MINE-DV, Generalized MINE-F, GIWAE, IWAE, and INFONCE from the perspective of conjugate duality. In particular, we highlight that the critic or negative energy function in the above bounds arises as a dual variable in the convex conjugate representation of the KL divergence. In all cases, the KL divergence of interest corresponds to the gap in the BA lower bound

$$I(\mathbf{x};\mathbf{z}) = \mathbb{E}_{p(\mathbf{x},\mathbf{z})}\left[\log\frac{q_\theta(\mathbf{z}|\mathbf{x})}{p(\mathbf{z})}\right] + \mathbb{E}_{p(\mathbf{x})}\left[D_{\mathrm{KL}}[p(\mathbf{z}|\mathbf{x})\|q_\theta(\mathbf{z}|\mathbf{x})]\right]. \tag{129}$$

For $q_\theta(\mathbf{z}|\mathbf{x}) = p(\mathbf{z})$, the BA lower bound term is $0$ and our derivations correspond to taking dual representation of MI directly, e.g. $\mathbb{E}_{p(\mathbf{x})}\left[D_{\mathrm{KL}}[p(\mathbf{z}|\mathbf{x})\|p(\mathbf{z})]\right]$, as in Belghazi et al. (2018).

Our conjugate duality interpretations are complementary to our probabilistic interpretations, with either approach equally valid for deriving lower bounds and characterizing their gaps.

## K.1  CONVEX CONJUGATE BACKGROUND

The KL divergence to a fixed reference distribution $\pi_0$ is a convex function of the distribution in the first argument $\Omega(\cdot) := D_{\mathrm{KL}}[\cdot\|\pi_0(\mathbf{z})]$. In later subsections, we will also consider KL divergences over joint distributions, unnormalized density functions, and extended state spaces.

For a given $\Omega(\cdot)$, we can define the conjugate function $\Omega^*(\cdot)$ over a dual variable or function $T(\mathbf{z})$, with the following relationships (Boyd & Vandenberghe, 2004)

$$\Omega^*(T(\mathbf{z})) := \sup_{\pi(\mathbf{z})} \langle \pi(\mathbf{z}), T(\mathbf{z}) \rangle - \Omega(\pi(\mathbf{z})), \tag{130}$$

$$\Omega(\pi(\mathbf{z})) = \sup_{T(\mathbf{z})} \langle \pi(\mathbf{z}), T(\mathbf{z}) \rangle - \Omega^*(T(\mathbf{z})), \tag{131}$$

where inner product notation indicates $\langle \pi(\mathbf{z}), T(\mathbf{z}) \rangle := \int \pi(\mathbf{z}) T(\mathbf{z}) d\mathbf{z}$. Solving for the optimizing argument in each of Eq. (130) and Eq. (131) yields the following dual correspondences

$$\pi_T(\mathbf{z}) = \nabla_T \Omega^*(T(\mathbf{z})), \qquad T_\pi(\mathbf{z}) = \nabla_\pi \Omega(\pi(\mathbf{z})). \tag{132}$$

We will proceed to derive closed form expressions for various special cases of $\Omega$ and $\Omega^*$ below.

For an arbitrary $p(\mathbf{z})$ and $T(\mathbf{z})$ which are not in dual correspondence according to Eq. (132), we can use Eq. (131) to derive a lower bound on $\Omega(p(\mathbf{z}))$ known as Fenchel's inequality

$$\Omega(p(\mathbf{z})) = \langle p(\mathbf{z}), T(\mathbf{z}) \rangle - \Omega^*(T(\mathbf{z})) + D_{\Omega(\cdot)}\big[p(\mathbf{z}), \pi_T(\mathbf{z})\big] \tag{133}$$

$$\geq \langle p(\mathbf{z}), T(\mathbf{z}) \rangle - \Omega^*(T(\mathbf{z})). \tag{134}$$

where the gap in the inequality $D_{\Omega(\cdot)}\big[p(\mathbf{z}), \pi_T(\mathbf{z})\big]$ is the Bregman divergence generated by $\Omega$ and $\pi_T(\mathbf{z})$ is the dual variable corresponding to $T(\mathbf{z})$ using Eq. (132).[3]

## K.2  CONJUGATE DUALITY INTERPRETATION OF IBAL

To obtain an alternative derivation of IBAL$(q_\theta, T_\phi)$, we consider the conditional KL divergence from a reference $q_\theta(\mathbf{z}|\mathbf{x})$, which is a convex function of its first argument

$$\Omega(\cdot) = D_{\mathrm{KL}}[\cdot\|q_\theta(\mathbf{z}|\mathbf{x})], \tag{135}$$

To derive the conjugate function $\Omega^*(T)$, note that we must restrict the optimization to the simplex $\pi(\mathbf{z}|\mathbf{x}) \in \Delta$, since the standard KL divergence requires a normalized distribution as input

$$\Omega^*(T(\mathbf{x},\mathbf{z})) := \sup_{\pi(\mathbf{z}|\mathbf{x})} \int \pi(\mathbf{z}|\mathbf{x}) \cdot T(\mathbf{x},\mathbf{z}) d\mathbf{z} - \Omega(\pi(\mathbf{z}|\mathbf{x})) - \lambda\big(\int \pi(\mathbf{z}|\mathbf{x}) d\mathbf{z} - 1\big) \tag{136}$$

$$= \log \mathbb{E}_{q_\theta(\mathbf{z}|\mathbf{x})}\left[e^{T(\mathbf{x},\mathbf{z})}\right] \tag{137}$$

---

[3]This follows directly from the definition of the Bregman divergence by using $T_q = \nabla\Omega(q)$ and the identity in Eq. (130), $D_\Omega[p,q] = \Omega(p) - \Omega(q) - \langle \nabla\Omega(q), p - q \rangle = \Omega(p) - \Omega(q) + \langle T_q, q \rangle - \langle T_q, p \rangle = \Omega(p) + \Omega^*(q) - \langle T_q, p \rangle$.

where we have solved for the optimizing argument $\pi_T(\mathbf{z}|\mathbf{x})$ to obtain the conjugate function in Eq. (137). Eq. (132) suggests the following dual correspondence between primal and dual variables

$$\pi_T(\mathbf{z}|\mathbf{x}) = \frac{1}{\mathcal{Z}(\mathbf{x};T)} q_\theta(\mathbf{z}|\mathbf{x}) e^{T(\mathbf{x},\mathbf{z})} \qquad T_\pi(\mathbf{x},\mathbf{z}) = \log \frac{\pi(\mathbf{z}|\mathbf{x})}{q_\theta(\mathbf{z}|\mathbf{x})} + c(\mathbf{x}). \tag{138}$$

We would like to leverage this duality to estimate the KL divergence $\Omega(p(\mathbf{z}|\mathbf{x})) = D_{\mathrm{KL}}[p(\mathbf{z}|\mathbf{x})\|q_\theta(\mathbf{z}|\mathbf{x})]$ from $q_\theta(\mathbf{z}|\mathbf{x})$ to the true posterior $p(\mathbf{z}|\mathbf{x})$. In particular, plugging Eq. (138) into Eq. (131) suggests the following variational representation

$$D_{\mathrm{KL}}[p(\mathbf{z}|\mathbf{x})\|q_\theta(\mathbf{z}|\mathbf{x})] = \sup_{T(\mathbf{x},\mathbf{z})} \int p(\mathbf{z}|\mathbf{x}) \, T(\mathbf{x},\mathbf{z}) d\mathbf{z} - \log \mathbb{E}_{q_\theta(\mathbf{z}|\mathbf{x})} \left[ e^{T(\mathbf{x},\mathbf{z})} \right]. \tag{139}$$

For a suboptimal $T_\pi(\mathbf{x},\mathbf{z})$, which is in dual correspondence with $\pi_T(\mathbf{z}|\mathbf{x})$ instead of the desired posterior $p(\mathbf{z}|\mathbf{x})$, we can use Eq. (134) to obtain a lower bound on $D_{\mathrm{KL}}[p(\mathbf{z}|\mathbf{x})\|q_\theta(\mathbf{z}|\mathbf{x})]$. To characterize the gap in this inequality, one can confirm that the Bregman divergence generated by the KL divergence in Eq. (135) is also the KL divergence $D_{D_{\mathrm{KL}}[\cdot\|q]}[p,\pi] = D_{\mathrm{KL}}[p\|\pi]$. Thus, we have

$$D_{\mathrm{KL}}[p(\mathbf{z}|\mathbf{x})\|q_\theta(\mathbf{z}|\mathbf{x})] = \int p(\mathbf{z}|\mathbf{x}) \, T(\mathbf{x},\mathbf{z}) d\mathbf{z} - \log \mathbb{E}_{q_\theta(\mathbf{z}|\mathbf{x})} \left[ e^{T(\mathbf{x},\mathbf{z})} \right] + D_{\mathrm{KL}}[p(\mathbf{z}|\mathbf{x})\|\pi_T(\mathbf{z}|\mathbf{x})] \tag{140}$$

Finally, the IBAL$(q_\theta, T_\phi)$ uses this variational representation of the gap in the BA lower bound, $\mathbb{E}_{p(\mathbf{x})}[D_{\mathrm{KL}}[p(\mathbf{z}|\mathbf{x})\|q_\theta(\mathbf{z}|\mathbf{x})]]$, to obtain a tighter bound on MI. In particular, for any learned critic function $T(\mathbf{x},\mathbf{z})$, we can use Eq. (140) to derive the IBAL and its gap

$$I(\mathbf{x};\mathbf{z}) = \underbrace{\mathbb{E}_{p(\mathbf{x},\mathbf{z})}\left[\log \frac{q_\theta(\mathbf{z}|\mathbf{x})}{p(\mathbf{z})}\right]}_{I_{\mathrm{BA_L}}(q_\theta)} + \mathbb{E}_{p(\mathbf{x})}\left[D_{\mathrm{KL}}[p(\mathbf{z}|\mathbf{x})\|q_\theta(\mathbf{z}|\mathbf{x})]\right] \tag{141}$$

$$= \underbrace{\mathbb{E}_{p(\mathbf{x},\mathbf{z})}\left[\log \frac{q_\theta(\mathbf{z}|\mathbf{x})}{p(\mathbf{z})}\right] + \mathbb{E}_{p(\mathbf{x})}\left[\mathbb{E}_{p(\mathbf{z}|\mathbf{x})}[T(\mathbf{x},\mathbf{z})] - \log \mathbb{E}_{q_\theta(\mathbf{z}|\mathbf{x})}\left[e^{T(\mathbf{x},\mathbf{z})}\right]\right]}_{\mathrm{IBAL}(q_\theta, T_\phi)} + \mathbb{E}_{p(\mathbf{x})}\left[D_{\mathrm{KL}}[p(\mathbf{z}|\mathbf{x})\|\pi_T(\mathbf{z}|\mathbf{x})]\right].$$

The optimal critic function $T^*(\mathbf{x},\mathbf{z})$ provides the maximizing argument in Eq. (139) and is in dual correspondence with the true posterior $p(\mathbf{z}|\mathbf{x})$. In particular, we have $\pi_{T^*}(\mathbf{z}|\mathbf{x}) = p(\mathbf{z}|\mathbf{x})$, resulting in $I(\mathbf{x};\mathbf{z}) = \mathrm{IBAL}(q_\theta, T^*)$.

### K.3 CONJUGATE DUALITY INTERPRETATION OF GENERALIZED MINE-DV

To obtain a conjugate duality interpretation of (Generalized) MINE-DV, we consider the dual representation of the KL divergence over *joint* distributions. Choosing $\pi_0(\mathbf{x},\mathbf{z}) = p(\mathbf{x})q_\theta(\mathbf{z}|\mathbf{x})$ as the reference distribution, the KL divergence is a convex function of the first argument

$$\Omega(\cdot) = D_{\mathrm{KL}}[\cdot\|p(\mathbf{x})q_\theta(\mathbf{z}|\mathbf{x})] \tag{142}$$

This KL divergence is equivalent to the gap in the BA bound $\mathbb{E}_{p(\mathbf{x})}[D_{\mathrm{KL}}[p(\mathbf{z}|\mathbf{x})\|q_\theta(\mathbf{z}|\mathbf{x})]]$ after noting the marginal distribution of both $p(\mathbf{x},\mathbf{z})$ and $p(\mathbf{x})q_\theta(\mathbf{z}|\mathbf{x})$ is $p(\mathbf{x})$.

However, the duality associated with $\Omega(\cdot) = D_{\mathrm{KL}}[\cdot\|\pi_0(\mathbf{x},\mathbf{z})]$ holds for general joint distributions, and we will see that the conjugate duality perspective using this divergence leads to looser bound on MI than in App. K.2. To evaluate the expression $\Omega(\pi(\mathbf{x},\mathbf{z})) = \sup_{T(\mathbf{x},\mathbf{z})}\langle\pi(\mathbf{x},\mathbf{z}), T(\mathbf{x},\mathbf{z})\rangle - \Omega^*(T(\mathbf{x},\mathbf{z}))$, we need to derive the conjugate function $\Omega^*(T(\mathbf{x},\mathbf{z}))$. Similarly to Eq. (136), we constrain the joint distribution to be normalized and obtain

$$\Omega^*(T(\mathbf{x},\mathbf{z})) := \sup_{\pi(\mathbf{x},\mathbf{z})} \int \pi(\mathbf{x},\mathbf{z}) \cdot T(\mathbf{x},\mathbf{z}) d\mathbf{x}d\mathbf{z} - \Omega(\pi(\mathbf{x},\mathbf{z})) - \lambda\left(\int \pi(\mathbf{x},\mathbf{z}) d\mathbf{z}d\mathbf{x} - 1\right) \tag{143}$$

$$= \log \mathbb{E}_{p(\mathbf{x})q_\theta(\mathbf{z}|\mathbf{x})}\left[e^{T(\mathbf{x},\mathbf{z})}\right], \tag{144}$$

where we have used $\pi_0(\mathbf{x},\mathbf{z}) = p(\mathbf{x})q_\theta(\mathbf{z}|\mathbf{x})$. Note that the expectation over $p(\mathbf{x})$ now appears inside the log in $\Omega^*(T(\mathbf{x},\mathbf{z}))$, compared with the conjugate for the conditional KL divergence in Eq. (137).

Solving for the optimizing arguments in Eq. (131) and Eq. (143), we have the following relationship between primal and dual variables,

$$\pi_T(\mathbf{x},\mathbf{z}) := \frac{1}{\mathcal{Z}(T)} p(\mathbf{x})q_\theta(\mathbf{z}|\mathbf{x})e^{T(\mathbf{x},\mathbf{z})}, \qquad T_\pi(\mathbf{x},\mathbf{z}) = \log \frac{\pi(\mathbf{x},\mathbf{z})}{p(\mathbf{x})q_\theta(\mathbf{z}|\mathbf{x})} + c, \tag{145}$$

Finally, we use Eq. (131) to write the dual representation of joint KL divergence as

$$D_{\text{KL}}[p(\mathbf{x},\mathbf{z})\|p(\mathbf{x})q_\theta(\mathbf{z}|\mathbf{x})] = \sup_{T(\mathbf{x},\mathbf{z})} \int p(\mathbf{x},\mathbf{z}) \cdot T(\mathbf{x},\mathbf{z})d\mathbf{x}d\mathbf{z} - \log \mathbb{E}_{p(\mathbf{x})q_\theta(\mathbf{z}|\mathbf{x})}[e^{T(\mathbf{x},\mathbf{z})}]. \quad (146)$$

As in the previous section and Eq. (134), a suboptimal $T(\mathbf{x},\mathbf{z})$, which is in dual correspondence with $\pi_T(\mathbf{z}|\mathbf{x})$ instead of the desired posterior $p(\mathbf{z}|\mathbf{x})$, yields a lower bound on $D_{\text{KL}}[p(\mathbf{x},\mathbf{z})\|p(\mathbf{x})q_\theta(\mathbf{z}|\mathbf{x})]$, with the gap equal to a KL divergence

$$D_{\text{KL}}[p(\mathbf{x},\mathbf{z})\|p(\mathbf{x})q_\theta(\mathbf{z}|\mathbf{x})] = \int p(\mathbf{x},\mathbf{z})T(\mathbf{x},\mathbf{z})d\mathbf{x}d\mathbf{z} - \log \mathbb{E}_{p(\mathbf{x})q_\theta(\mathbf{z}|\mathbf{x})}[e^{T(\mathbf{x},\mathbf{z})}] + D_{\text{KL}}[p(\mathbf{x},\mathbf{z})\|\pi_T(\mathbf{x},\mathbf{z})].$$
$$(147)$$

The Generalized MINE-DV bound $I_{\text{GMINE-DV}}(q_\theta, T_\phi)$ uses this variational representation of the gap in the BA lower bound to obtain a tighter bound on MI. For a learned critic $T(\mathbf{x},\mathbf{z})$, we can use Eq. (147) to write

$$I(\mathbf{x};\mathbf{z}) = \underbrace{\mathbb{E}_{p(\mathbf{x},\mathbf{z})}\left[\log \frac{q_\theta(\mathbf{z}|\mathbf{x})}{p(\mathbf{z})}\right]}_{I_{\text{BA}_L}(q_\theta)} + \mathbb{E}_{p(\mathbf{x})}\left[D_{\text{KL}}[p(\mathbf{z}|\mathbf{x})\|q_\theta(\mathbf{z}|\mathbf{x})]\right] \quad (148)$$

$$= \underbrace{\mathbb{E}_{p(\mathbf{x},\mathbf{z})}\left[\log \frac{q_\theta(\mathbf{z}|\mathbf{x})}{p(\mathbf{z})}\right] + \mathbb{E}_{p(\mathbf{x})p(\mathbf{x}|\mathbf{x})}[T(\mathbf{x},\mathbf{z})] - \log \mathbb{E}_{q_\theta(\mathbf{z}|\mathbf{x})}\left[e^{T(\mathbf{x},\mathbf{z})}\right]}_{I_{\text{GMINE-DV}}(q_\theta, T_\phi)} + D_{\text{KL}}[p(\mathbf{x})p(\mathbf{z}|\mathbf{x})\|\pi_T(\mathbf{x},\mathbf{z})]$$

As noted in App. J.2, the Generalized MINE-DV bound is looser than the IBAL.

### K.4 CONJUGATE DUALITY INTERPRETATION OF GENERALIZED MINE-F

Nguyen et al. (2010) consider the conjugate duality associated with the family of $f$-divergences, of which the KL divergence is a special case. We will show that this dual representation corresponds to taking the conjugate function of the Generalized KL divergence, which can take unnormalized densities as input. See App. J.3 for definition of the Generalized KL divergence, which is convex in either argument since it differs from the standard KL divergence by only a linear term..

For $\Omega(\cdot) = D_{\text{GKL}}[\cdot\|\tilde{q}_\theta(\mathbf{z}|\mathbf{x})]$, we write the dual variable using the notation $T'(\mathbf{x},\mathbf{z})$ and consider

$$\Omega^*(T'(\mathbf{x},\mathbf{z})) := \sup_{\tilde{\pi}(\mathbf{z}|\mathbf{x})} \int \tilde{\pi}(\mathbf{z}|\mathbf{x})T'(\mathbf{x},\mathbf{z})d\mathbf{z} - D_{\text{GKL}}[\tilde{\pi}(\mathbf{z}|\mathbf{x})\|\tilde{q}_\theta(\mathbf{z}|\mathbf{x})] \quad (149)$$

Note that we do not explicitly include a Lagrange multiplier to enforce restriction to normalized distributions. Solving for the optimizing argument or writing the dual correspondence in Eq. (132), we obtain

$$\tilde{\pi}_T(\mathbf{z}|\mathbf{x}) = \tilde{q}_\theta(\mathbf{z}|\mathbf{x})e^{T'(\mathbf{x},\mathbf{z})}, \qquad T'_{\tilde{\pi}}(\mathbf{x},\mathbf{z}) = \log \frac{\tilde{\pi}(\mathbf{z}|\mathbf{x})}{\tilde{q}_\theta(\mathbf{z}|\mathbf{x})}. \quad (150)$$

Plugging this $\tilde{\pi}_T(\mathbf{z}|\mathbf{x})$ back into Eq. (149) yields the conjugate function

$$\Omega^*(T'(\mathbf{x},\mathbf{z})) = \int \tilde{q}(\mathbf{z}|\mathbf{x})e^{T'(\mathbf{x},\mathbf{z})}d\mathbf{z} - \int \tilde{q}(\mathbf{z}|\mathbf{x})d\mathbf{z}, \quad (151)$$

which leads to a dual representation of the Generalized KL divergence that matches Nguyen et al. (2010) after plugging into Eq. (131)

$$D_{\text{GKL}}[\tilde{p}(\mathbf{z}|\mathbf{x})\|\tilde{q}(\mathbf{z}|\mathbf{x})] = \sup_{T'(\mathbf{x},\mathbf{z})} \int \tilde{p}(\mathbf{z}|\mathbf{x})T'(\mathbf{x},\mathbf{z})d\mathbf{z} - \int \tilde{q}(\mathbf{z}|\mathbf{x})e^{T'(\mathbf{x},\mathbf{z})}d\mathbf{z} + \int \tilde{q}(\mathbf{z}|\mathbf{x})d\mathbf{z}. \quad (152)$$

We now use the reparameterization $T'(\mathbf{x},\mathbf{z}) = T(\mathbf{x},\mathbf{z}) - 1$. Assuming a normalized $\tilde{q}(\mathbf{z}|\mathbf{x}) = q(\mathbf{z}|\mathbf{x})$ and $\tilde{p}(\mathbf{z}|\mathbf{x}) = p(\mathbf{z}|\mathbf{x})$, and noting that $D_{\text{GKL}}[p(\mathbf{z}|\mathbf{x})\|q_\theta(\mathbf{z}|\mathbf{x})] = D_{\text{KL}}[p(\mathbf{z}|\mathbf{x})\|q_\theta(\mathbf{z}|\mathbf{x})]$ for normalized distributions, we obtain

$$D_{\text{KL}}[p(\mathbf{z}|\mathbf{x})\|q_\theta(\mathbf{z}|\mathbf{x})] = \sup_{T(\mathbf{x},\mathbf{z})} \int p(\mathbf{z}|\mathbf{x})\,T(\mathbf{x},\mathbf{z})d\mathbf{z} - \int q_\theta(\mathbf{z}|\mathbf{x})e^{T(\mathbf{x},\mathbf{z})-1}d\mathbf{z}, \quad (153)$$

which matches dual representation of the KL divergence found in Belghazi et al. (2018); Nowozin et al. (2016). See Ruderman et al. (2012) for further discussion.

For a suboptimal $T_\pi(\mathbf{x}, \mathbf{z})$ which is in dual correspondence with $\pi_T(\mathbf{z}|\mathbf{x})$ instead of the desired posterior $p(\mathbf{z}|\mathbf{x})$, we can use Eq. (134) to obtain a lower bound on $D_{\mathrm{GKL}}[p(\mathbf{z}|\mathbf{x})\|q_\theta(\mathbf{z}|\mathbf{x})] = D_{\mathrm{KL}}[p(\mathbf{z}|\mathbf{x})\|q_\theta(\mathbf{z}|\mathbf{x})]$. To characterize the gap of this lower bound, note that the Bregman divergence generated by the $D_{\mathrm{GKL}}$ divergence is also the $D_{\mathrm{GKL}}$ divergence $D_{D_{\mathrm{GKL}}[\cdot\|\tilde{q}]}[\tilde{p}\|\tilde{\pi}] = D_{\mathrm{GKL}}[\tilde{p}\|\tilde{\pi}]$. Thus, using Eq. (133), we have

$$D_{\mathrm{KL}}[p(\mathbf{z}|\mathbf{x})\|q_\theta(\mathbf{z}|\mathbf{x})] = \int p(\mathbf{z}|\mathbf{x})\,T(\mathbf{x}, \mathbf{z})d\mathbf{z} - \int q_\theta(\mathbf{z}|\mathbf{x})e^{T(\mathbf{x},\mathbf{z})-1}d\mathbf{z} + D_{\mathrm{GKL}}[p(\mathbf{z}|\mathbf{x})\|q_\theta(\mathbf{z}|\mathbf{x})e^{T(\mathbf{x},\mathbf{z})-1}].$$
(154)

Finally, we obtain the Generalized MINE-F bound by evaluating the dual representation of the Generalized KL divergence for normalized $p(\mathbf{z}|\mathbf{x})$ and $q_\theta(\mathbf{z}|\mathbf{x})$. In particular, $I_{\mathrm{GMINE\text{-}F}}(q_\theta, T_\phi)$ uses the critic function $T_\phi$ to tighten the gap in $I_{\mathrm{BA_L}}(q_\theta)$ via the dual optimization in Eq. (154),

$$I(\mathbf{x}; \mathbf{z}) = \underbrace{\mathbb{E}_{p(\mathbf{x},\mathbf{z})}\left[\log \frac{q_\theta(\mathbf{z}|\mathbf{x})}{p(\mathbf{z})}\right]}_{I_{\mathrm{BA_L}}(q_\theta)} + \mathbb{E}_{p(\mathbf{x})}\left[D_{\mathrm{KL}}[p(\mathbf{z}|\mathbf{x})\|q_\theta(\mathbf{z}|\mathbf{x})]\right]$$
(155)

$$= \underbrace{\mathbb{E}_{p(\mathbf{x},\mathbf{z})}\left[\log \frac{q_\theta(\mathbf{z}|\mathbf{x})}{p(\mathbf{z})}\right] + \mathbb{E}_{p(\mathbf{x})}\left[\mathbb{E}_{p(\mathbf{z}|\mathbf{x})}[T(\mathbf{x}, \mathbf{z})] - \log \mathbb{E}_{q_\theta(\mathbf{z}|\mathbf{x})}\left[e^{T(\mathbf{x},\mathbf{z})}\right]\right]}_{I_{\mathrm{GMINE\text{-}F}}(q_\theta, T_\phi)} + \mathbb{E}_{p(\mathbf{x})}\left[D_{\mathrm{GKL}}[p(\mathbf{z}|\mathbf{x})\|\tilde{\pi}_T(\mathbf{z}|\mathbf{x})]\right]$$
(156)

The optimal critic function is the dual variable of the true posterior $p(\mathbf{z}|\mathbf{x})$, which can be found using Eq. (150) as $T(\mathbf{x}, \mathbf{z}) = 1 + \log \frac{\tilde{p}(\mathbf{z}|\mathbf{x})}{\tilde{q}_\theta(\mathbf{z}|\mathbf{x})}$. With this optimal critic, the Generalized MINE-F bound is tight.

## K.5  CONJUGATE DUALITY INTERPRETATION OF GIWAE, IWAE, AND INFONCE

In this section, we use conjugate duality to derive the GIWAE, IWAE, and INFO-NCE bounds on mutual information and characterize their gaps. Our approach extends that of Poole et al. (2019), where the MINE-F dual representation (App. K.4) was used to derive INFO-NCE. We provide an alternative derivation using the dual representation associated with the conditional KL divergence and IBAL in App. K.2. For either dual representation, GIWAE, IWAE, and INFO-NCE arise from limiting the family of the critic functions $T_\phi$ in order to eliminate the intractable log partition function term.

We start from the decomposition of $I(\mathbf{x}; \mathbf{z})$ into $I_{\mathrm{BA_L}}(q_\theta)$ and its gap

$$I(\mathbf{x}; \mathbf{z}) = \underbrace{\mathbb{E}_{p(\mathbf{x},\mathbf{z})}\left[\log \frac{q_\theta(\mathbf{z}|\mathbf{x})}{p(\mathbf{z})}\right]}_{I_{\mathrm{BA_L}}(q_\theta)} + \mathbb{E}_{p(\mathbf{x})}\left[D_{\mathrm{KL}}[p(\mathbf{z}|\mathbf{x})\|q_\theta(\mathbf{z}|\mathbf{x})]\right].$$
(157)

We will focus on the dual representation of the normalized, conditional KL divergence $\Omega(\cdot) = D_{\mathrm{KL}}[\cdot\|q_\theta(\mathbf{z}|\mathbf{x})]$, as in App. K.2.

**Multi-Sample IBAL**   Consider extending the state space of the posterior or target distribution $p(\mathbf{z}|\mathbf{x})$, by using an additional $K-1$ samples from a base variational distribution $q_\theta(\mathbf{z}|\mathbf{x})$ to construct

$$p_{\mathrm{TGT}}(\mathbf{z}^{(1:K)}, s|\mathbf{x}) := \mathcal{U}(s)p(\mathbf{z}^{(s)}|\mathbf{x})\prod_{\substack{k \neq s \\ k=1}}^{K} q_\theta(\mathbf{z}^{(k)}|\mathbf{x}),$$
(158)

where $s$ is an index variable $s \sim \mathcal{U}(s)$ drawn uniformly at random from $1 \leq s \leq K$, which specifies the index of the posterior sample $p(\mathbf{z}|\mathbf{x})$. We similarly expand the state space of the base variational distribution to write

$$q_{\mathrm{PROP}}(\mathbf{z}^{(1:K)}, s|\mathbf{x}) := \mathcal{U}(s)\prod_{k=1}^{K} q_\theta(\mathbf{z}^{(k)}|\mathbf{x}).$$
(159)

It can be easily verified that this construction does not change the value of the KL divergence

$$D_{\text{KL}}[p_{\text{TGT}}(\mathbf{z}^{(1:K)}, s|\mathbf{x})\|q_{\text{PROP}}(\mathbf{z}^{(1:K)}, s|\mathbf{x})] = \mathbb{E}_{p_{\text{TGT}}(\mathbf{z}^{(1:K)}, s|\mathbf{x})}\left[\log\frac{p_{\text{TGT}}(\mathbf{z}^{(1:K)}, s|\mathbf{x})}{q_{\text{PROP}}(\mathbf{z}^{(1:K)}, s|\mathbf{x})}\right]$$

$$= \mathbb{E}_{p_{\text{TGT}}(\mathbf{z}^{(1:K)}, s|\mathbf{x})}\left[\log\frac{\cancel{\mathcal{U}(s)}p(\mathbf{z}^{(s)}|\mathbf{x})\prod_{k\neq s}q_\theta(\mathbf{z}^{(k)}|\mathbf{x})}{\cancel{\mathcal{U}(s)}\prod_{k=1}^{K}q_\theta(\mathbf{z}^{(k)}|\mathbf{x})}\right]$$

$$= \mathbb{E}_{\mathcal{U}(s)p(\mathbf{z}^{(s)}|\mathbf{x})}\left[\log\frac{p(\mathbf{z}^{(s)}|\mathbf{x})}{q_\theta(\mathbf{z}^{(s)}|\mathbf{x})}\right]$$

$$= \mathbb{E}_{p(\mathbf{z}|\mathbf{x})}\left[\log\frac{p(\mathbf{z}|\mathbf{x})}{q_\theta(\mathbf{z}|\mathbf{x})}\right]$$

$$= D_{\text{KL}}[p(\mathbf{z}|\mathbf{x})\|q_\theta(\mathbf{z}|\mathbf{x})].$$

Consider the convex function

$$\Omega(\cdot) := D_{\text{KL}}\big[\,\cdot\,\|q_{\text{PROP}}(\mathbf{z}^{(1:K)}, s|\mathbf{x})\big], \tag{160}$$

where the primal variable is a distribution in the extended-state space of $(\mathbf{z}^{(1:K)}, s)$, and the dual variable is a critic function $\mathcal{T}(\mathbf{x}, \mathbf{z}^{(1:K)}, s)$. We derive a conjugate optimization in similar fashion to App. K.2, but now over the extended state space. For this $\Omega(\cdot)$, the conjugate function $\Omega^*(\mathcal{T})$ takes a log-mean-exp form analogous to Eq. (137)[4]. We can write the variational representation of $\Omega(p_{\text{TGT}})$ as

$$\Omega(p_{\text{TGT}}) = \sup_{\mathcal{T}} \int \sum_{s=1}^{K} p_{\text{TGT}}(\mathbf{z}^{(1:K)}, s|\mathbf{x})\mathcal{T}(\mathbf{x}, \mathbf{z}^{(1:K)}, s)d\mathbf{z}^{(1:K)} - \log\mathbb{E}_{q_{\text{PROP}}(\mathbf{z}^{(1:K)}, s|\mathbf{x})}\left[e^{\mathcal{T}(\mathbf{x}, \mathbf{z}^{(1:K)}, s)}\right]. \tag{161}$$

For a particular choice of $\mathcal{T}(\mathbf{x}, \mathbf{z}^{(1:K)}, s)$, we can use Eq. (161) to obtain a lower bound on the KL divergence $D_{\text{KL}}[p(\mathbf{z}|\mathbf{x})\|q_\theta(\mathbf{z}|\mathbf{x})]$ (as in Eq. (134)). This lower bound translates to the *Multi-Sample* IBAL lower bound on MI, $I_{\text{MS-IBAL}}(q_\theta, \mathcal{T})$ via Eq. (157) .

$$I(\mathbf{x}; \mathbf{z}) \geq \mathbb{E}_{p(\mathbf{x}, \mathbf{z})}\left[\log\frac{q_\theta(\mathbf{z}|\mathbf{x})}{p(\mathbf{z})}\right] + \mathbb{E}_{p(\mathbf{x})}\left[\mathbb{E}_{p_{\text{TGT}}(\mathbf{z}^{(1:K)}, s|\mathbf{x})}\left[\mathcal{T}(\mathbf{x}, \mathbf{z}^{(1:K)}, s)\right]\right] - \underbrace{\log\mathbb{E}_{q_{\text{PROP}}(\mathbf{z}^{(1:K)}, s|\mathbf{x})}\left[e^{\mathcal{T}(\mathbf{x}, \mathbf{z}^{(1:K)}, s)}\right]}_{\mathcal{Z}(\mathbf{x}; \mathcal{T})}$$

$$=: I_{\text{MS-IBAL}}(q_\theta, \mathcal{T}), \tag{162}$$

where $\mathcal{Z}(\mathbf{x}; \mathcal{T})$ is the normalization constant of the dual distribution $\pi_{\mathcal{T}}(\mathbf{z}^{(1:K)}, s|\mathbf{x})$ corresponding to $\mathcal{T}$,

$$\pi_{\mathcal{T}}(\mathbf{z}^{(1:K)}, s|\mathbf{x}) = \frac{1}{\mathcal{Z}(\mathbf{x}; \mathcal{T})}q_{\text{PROP}}(\mathbf{z}^{(1:K)}, s|\mathbf{x})e^{\mathcal{T}(\mathbf{x}, \mathbf{z}^{(1:K)}, s)}. \tag{163}$$

As in Eq. (133), we can write the gap of $I_{\text{MS-IBAL}}(q_\theta, \mathcal{T})$ as a Bregman divergence or KL divergence

$$I(\mathbf{x}; \mathbf{z}) = I_{\text{MS-IBAL}}(q_\theta, \mathcal{T}) + \mathbb{E}_{p(\mathbf{x})}\left[D_{\text{KL}}\big[p_{\text{TGT}}(\mathbf{z}^{(1:K)}, s|\mathbf{x})\|\pi_{\mathcal{T}}(\mathbf{z}^{(1:K)}, s|\mathbf{x})\big]\right]. \tag{164}$$

The optimal $K$-sample energy function in Eq. (161) should result in $\pi_{\mathcal{T}^*}(\mathbf{z}^{(1:K)}, s|\mathbf{x}) = p_{\text{TGT}}(\mathbf{z}^{(1:K)}, s|\mathbf{x})$. Using similar reasoning as in App. C.5 or App. L.1, this occurs for $\mathcal{T}^*(\mathbf{x}, \mathbf{z}^{(1:K)}, s) = \log\frac{p(\mathbf{z}^{(s)}|\mathbf{x})}{q_\theta(\mathbf{z}^{(s)}|\mathbf{x})} + c(\mathbf{x})$, for which we have $I(\mathbf{x}; \mathbf{z}) = I_{\text{MS-IBAL}}(q_\theta, \mathcal{T}^*)$.

**GIWAE is a Multi-Sample IBAL with a Restricted Function Family**  Although extending the state space did not change the value of the KL divergence or alter the optimal critic function, it does allow us to consider a restricted class of multi-sample energy functions that yield tractable, low variance estimators. In particular, GIWAE and INFO-NCE arise from choosing a restricted family of functions $\mathcal{T}_{\text{GIWAE}}(\mathbf{x}, \mathbf{z}^{(1:K)}, s)$, under which the problematic $\log\mathcal{Z}(\mathbf{x}; \mathcal{T})$ term evaluates to 0. This function family is defined as

---

[4]We consider the conjugate with restriction to normalized distributions as in App. K.2 Eq. (136).

$$\mathcal{T}_{\text{GIWAE}}(\mathbf{x}, \mathbf{z}^{(1:K)}, s) := \log \frac{e^{T_\phi(\mathbf{x}, \mathbf{z}^{(s)})}}{\frac{1}{K} \sum_{k=1}^{K} e^{T_\phi(\mathbf{x}, \mathbf{z}^{(k)})}}, \tag{165}$$

where $\mathcal{T}_{\text{GIWAE}}(\mathbf{x}, \mathbf{z}^{(1:K)}, s)$ is specified by an arbitrary single-sample critic function $T_\phi(\mathbf{x}, \mathbf{z})$. We can now see that $\log \mathcal{Z}(\mathbf{x}; \mathcal{T}_{\text{GIWAE}}) = 0$,

$$\log \mathcal{Z}(\mathbf{x}; \mathcal{T}_{\text{GIWAE}}) = \log \mathbb{E}_{q_{\text{PROP}}(\mathbf{z}^{(1:K)}, s | \mathbf{x})} \left[ e^{\mathcal{T}_{\text{GIWAE}}(\mathbf{x}, \mathbf{z}^{(1:K)}, s)} \right] = \log \mathbb{E}_{\mathcal{U}(s) \prod_{k=1}^{K} q_\theta(\mathbf{z}^{(k)} | \mathbf{x})} \left[ \frac{e^{T_\phi(\mathbf{x}, \mathbf{z}^{(s)})}}{\frac{1}{K} \sum_{k=1}^{K} e^{T_\phi(\mathbf{x}, \mathbf{z}^{(k)})}} \right] = 0 \, ,$$

using the fact that $\mathbf{z}^{(1:K)} \sim \prod_{k=1}^{K} q_\theta(\mathbf{z}^{(k)} | \mathbf{x})$ is invariant to re-indexing. With this simplification, $I_{\text{MS-IBAL}}(q_\theta, \mathcal{T}_{\text{GIWAE}})$ recovers the GIWAE lower bound on MI

$$I(\mathbf{x}; \mathbf{z}) \geq I_{\text{MS-IBAL}}(q_\theta, \mathcal{T}_{\text{GIWAE}}) \tag{166}$$

$$= \mathbb{E}_{p(\mathbf{x}, \mathbf{z})} \left[ \log \frac{q_\theta(\mathbf{z} | \mathbf{x})}{p(\mathbf{z})} \right] + \mathbb{E}_{\frac{1}{K} p(\mathbf{x}) p(\mathbf{z}^{(s)} | \mathbf{x}) \prod_{\substack{k=1 \\ k \neq s}}^{K} q_\theta(\mathbf{z}^{(k)} | \mathbf{x})} \left[ \log \frac{e^{T_\phi(\mathbf{x}, \mathbf{z}^{(s)})}}{\frac{1}{K} \sum_{k=1}^{K} e^{T_\phi(\mathbf{x}, \mathbf{z}^{(k)})}} \right] \tag{167}$$

$$= \mathbb{E}_{p(\mathbf{x}, \mathbf{z})} \left[ \log \frac{q_\theta(\mathbf{z} | \mathbf{x})}{p(\mathbf{z})} \right] + \mathbb{E}_{p(\mathbf{x}) p(\mathbf{z}^{(1)} | \mathbf{x}) \prod_{k=2}^{K} q_\theta(\mathbf{z}^{(k)} | \mathbf{x})} \left[ \log \frac{e^{T_\phi(\mathbf{x}, \mathbf{z}^{(1)})}}{\frac{1}{K} \sum_{k=1}^{K} e^{T_\phi(\mathbf{x}, \mathbf{z}^{(k)})}} \right] \tag{168}$$

$$= I_{\text{GIWAE}}(q_\theta, T_\phi, K).$$

Finally, we can use Eq. (164) to recover the probabilistic interpretation of GIWAE and the gap in the lower bound on MI. As we saw above, the dual distribution $\pi_{\mathcal{T}_{\text{GIWAE}}}(\mathbf{z}^{(1:K)}, s | \mathbf{x})$ is normalized with $\mathcal{Z}(\mathbf{x}; \mathcal{T}_{\text{GIWAE}}) = 1$. In particular, we can write

$$\pi_{\mathcal{T}_{\text{GIWAE}}}(\mathbf{z}^{(1:K)}, s | \mathbf{x}) = \frac{1}{\mathcal{Z}(\mathbf{x}; \mathcal{T}_{\text{GIWAE}})} q_{\text{PROP}}(\mathbf{z}^{(1:K)}, s | \mathbf{x}) e^{\mathcal{T}_{\text{GIWAE}}(\mathbf{x}, \mathbf{z}^{(1:K)}, s)} \tag{169}$$

$$= \frac{1}{\mathcal{Z}(\mathbf{x}; \mathcal{T}_{\text{GIWAE}})} \cancel{\mathcal{U}(s)} \prod_{k=1}^{K} q_\theta(\mathbf{z}^{(k)} | \mathbf{x}) \frac{e^{T_\phi(\mathbf{x}, \mathbf{z}^{(s)})}}{\cancel{\frac{1}{K}} \sum_{k=1}^{K} e^{T_\phi(\mathbf{x}, \mathbf{z}^{(k)})}} \tag{170}$$

$$= \prod_{k=1}^{K} q_\theta(\mathbf{z}^{(k)} | \mathbf{x}) \frac{e^{T_\phi(\mathbf{x}, \mathbf{z}^{(s)})}}{\sum_{k=1}^{K} e^{T_\phi(\mathbf{x}, \mathbf{z}^{(k)})}} \tag{171}$$

which recovers $q_{\text{PROP}}^{\text{GIWAE}}(\mathbf{z}^{(1:K)}, s | \mathbf{x})$ from the probabilistic interpretation of GIWAE in Eq. (42) or App. C.1. We can write the gap of $I_{\text{GIWAE}}(q_\theta, T_\phi, K) = I_{\text{MS-IBAL}}(q_\theta, \mathcal{T}_{\text{GIWAE}})$ as the Bregman divergence or KL divergence as in Eq. (133)

$$I(\mathbf{x}; \mathbf{z}) = I_{\text{MS-IBAL}}(q_\theta, \mathcal{T}_{\text{GIWAE}}) + \mathbb{E}_{p(\mathbf{x})} \left[ D_{\text{KL}} \left[ p_{\text{TGT}}(\mathbf{z}^{(1:K)}, s | \mathbf{x}) \middle\| \pi_{\mathcal{T}_{\text{GIWAE}}}(\mathbf{z}^{(1:K)}, s | \mathbf{x}) \right] \right] , \tag{172}$$

which matches the reverse KL divergence $\mathbb{E}_{p(\mathbf{x})} \left[ D_{\text{KL}} [p_{\text{TGT}}^{\text{GIWAE}}(\mathbf{z}^{(1:K)}, s | \mathbf{x}) \| q_{\text{PROP}}^{\text{GIWAE}}(\mathbf{z}^{(1:K)}, s | \mathbf{x})] \right]$ derived from the probabilistic approach in App. C.1. Recall that INFO-NCE is a special case of GIWAE with $q_\theta(\mathbf{z} | \mathbf{x}) = p(\mathbf{z})$ (Sec. 2.4).

**Conjugate Duality Interpretation of IWAE** We can gain alternative perspective on IWAE from this conjugate duality interpretation. In particular, IWAE is a special case of GIWAE, where the optimal single-sample critic function $T^*(\mathbf{x}, \mathbf{z}) = \log \frac{p(\mathbf{x}, \mathbf{z})}{q_\theta(\mathbf{z} | \mathbf{x})} + c(\mathbf{x})$ (see Sec. 2.4) is used in Eq. (165) to construct the optimal multi-sample function $\mathcal{T}_{\text{IWAE}}$, within the GIWAE restricted multi-sample function family. Thus, we have $I_{\text{MS-IBAL}}(q_\theta, \mathcal{T}_{\text{IWAE}}) = I_{\text{IWAE}}(q_\theta, K)$.

Although IWAE uses the optimal critic function, the restriction to the function family in Eq. (165) is necessary to obtain a tractable bound on the KL divergence $D_{\text{KL}}[p(\mathbf{z}|\mathbf{x})\|q_\theta(\mathbf{z}|\mathbf{x})]$ and mutual information. Without the restricted function family, the intractable log partition term in Eq. (161) would require MCMC methods such as AIS to accurately estimate, as we saw for the single-sample IBAL in Sec. 4.

## L    PROPERTIES OF THE IBAL

Our MINE-AIS method in Sec. 4 and App. M optimizes the *Implicit Barber Agakov* lower bound (IBAL) on MI from Eq. (18). We first recall the probabilistic interpretation of the IBAL bound from App. J.1. For a posterior approximation with a learned negative energy function $T_\phi$ and base variational distribution $q_\theta(\mathbf{z}|\mathbf{x})$,

$$\pi_{\theta,\phi}(\mathbf{z}|\mathbf{x}) = \frac{1}{\mathcal{Z}_{\theta,\phi}(\mathbf{x})} q_\theta(\mathbf{z}|\mathbf{x}) e^{T_\phi(\mathbf{x},\mathbf{z})}, \quad \text{where} \quad \mathcal{Z}_{\theta,\phi}(\mathbf{x}) = \mathbb{E}_{q_\theta(\mathbf{z}|\mathbf{x})}\left[e^{T_\phi(\mathbf{x},\mathbf{z})}\right], \quad (173)$$

we consider the BA lower bound on MI,

$$I(\mathbf{x},\mathbf{z}) \geq I_{\text{BA}_L}(\pi_{\theta,\phi}) \qquad\qquad (174)$$

$$= I(\mathbf{x},\mathbf{z}) - \mathbb{E}_{p(\mathbf{x})}\left[D_{\text{KL}}[p(\mathbf{z}|\mathbf{x})\|\pi_{\theta,\phi}(\mathbf{z}|\mathbf{x})]\right] \qquad\qquad (175)$$

$$= \underbrace{\mathbb{E}_{p(\mathbf{x},\mathbf{z})}\left[\log \frac{q_\theta(\mathbf{z}|\mathbf{x})}{p(\mathbf{z})}\right]}_{I_{\text{BA}_L}(q_\theta)} + \underbrace{\mathbb{E}_{p(\mathbf{x},\mathbf{z})}\left[\log \frac{e^{T_\phi(\mathbf{x},\mathbf{z})}}{\mathbb{E}_{q_\theta(\mathbf{z}|\mathbf{x})}\left[e^{T_\phi(\mathbf{x},\mathbf{z})}\right]}\right]}_{\text{contrastive term}} \qquad (176)$$

$$=: \text{IBAL}(q_\theta, T_\phi). \qquad\qquad (177)$$

The gap of this lower bound on mutual information is $\mathbb{E}_{p(\mathbf{x})}\left[D_{\text{KL}}[p(\mathbf{z}|\mathbf{x})\|\pi_{\theta,\phi}(\mathbf{z}|\mathbf{x})]\right]$, as in Sec. 2.2.

### L.1    PROOFS FOR IBAL OPTIMAL CRITIC FUNCTION (PROP. 4.1 AND L.1)

**Proposition 4.1.** *For a given $q_\theta(\mathbf{z}|\mathbf{x})$, the optimal IBAL critic function equals the log importance weights up to a constant $T^*(\mathbf{x},\mathbf{z}) = \log \frac{p(\mathbf{x},\mathbf{z})}{q_\theta(\mathbf{z}|\mathbf{x})} + c(\mathbf{x})$. For this $T^*$, we have $\text{IBAL}(q_\theta, T^*) = I(\mathbf{x};\mathbf{z})$.*

*Proof.* Recall that the gap in $\text{IBAL}(q_\theta, T_{\phi^*})$ is $\mathbb{E}_{p(\mathbf{x})}\left[D_{\text{KL}}[p(\mathbf{z}|\mathbf{x})\|\pi_{\theta,\phi}(\mathbf{z}|\mathbf{x})]\right] = I(\mathbf{x};\mathbf{z}) - \text{IBAL}(q_\theta, T_{\phi^*})$. This implies that the bound will be tight iff $p(\mathbf{z}|\mathbf{x}) = \pi_{\theta,\phi}(\mathbf{z}|\mathbf{x}) \propto p(\mathbf{x},\mathbf{z})$. We can easily show that the true log importance weights (plus a constant) satisfy this property

$$\pi_{\theta,\phi}(\mathbf{z}|\mathbf{x}) = \frac{1}{\mathcal{Z}_{\theta,\phi}(\mathbf{x})} q_\theta(\mathbf{z}|\mathbf{x}) e^{\log \frac{p(\mathbf{x},\mathbf{z})}{q_\theta(\mathbf{z}|\mathbf{x})} + c(\mathbf{x})} = \frac{e^{c(\mathbf{x})}}{\mathcal{Z}_{\theta,\phi}(\mathbf{x})} p(\mathbf{x},\mathbf{z}) = p(\mathbf{z}|\mathbf{x}), \qquad (178)$$

where $e^{c(\mathbf{x})}$ is absorbed into the normalization constant. Conversely, using $g(\mathbf{x},\mathbf{z})$ which depends on $\mathbf{z}$ in $T(\mathbf{x},\mathbf{z}) = \log \frac{p(\mathbf{x},\mathbf{z})}{q_\theta(\mathbf{z}|\mathbf{x})} + g(\mathbf{x},\mathbf{z})$ would change the density over $\mathbf{z}$ to no longer match $p(\mathbf{z}|\mathbf{x})$.

At this value, the IBAL is exactly equal to $I(\mathbf{x};\mathbf{z})$

$$\text{IBAL}(q_\theta, T^*) = \mathbb{E}_{p(\mathbf{x},\mathbf{z})}\left[\log \frac{q_\theta(\mathbf{z}|\mathbf{x})}{p(\mathbf{z})}\right] + \mathbb{E}_{p(\mathbf{x},\mathbf{z})}\left[T_\phi(\mathbf{x},\mathbf{z})\right] - \mathbb{E}_{p(\mathbf{x})}\left[\log \mathcal{Z}_{\theta,\phi}(\mathbf{x})\right]$$

$$= \mathbb{E}_{p(\mathbf{x},\mathbf{z})}\left[\log \frac{q_\theta(\mathbf{z}|\mathbf{x})}{p(\mathbf{z})}\right] + \mathbb{E}_{p(\mathbf{x},\mathbf{z})}\left[\log \frac{p(\mathbf{x},\mathbf{z})}{q_\theta(\mathbf{z}|\mathbf{x})} + \log c(\mathbf{x})\right] - \mathbb{E}_{p(\mathbf{x})}\left[\log \mathbb{E}_{q_\theta(\mathbf{z}|\mathbf{x})} \frac{p(\mathbf{x},\mathbf{z})}{q_\theta(\mathbf{z}|\mathbf{x})} + \log c(\mathbf{x})\right]$$

$$= \mathbb{E}_{p(\mathbf{x},\mathbf{z})}\left[\log \frac{p(\mathbf{x},\mathbf{z})}{p(\mathbf{z})p(\mathbf{x})}\right] = I(\mathbf{x};\mathbf{z}).$$

$\square$

**Proposition L.1.** *Suppose the critic function $T_\phi(\mathbf{x},\mathbf{z})$ is parameterized by $\phi$, and that $\exists \phi_0 \text{ s.t. } T_{\phi_0}(\mathbf{x},\mathbf{z}) = const$. For a given $q_\theta(\mathbf{z}|\mathbf{x})$, let $T_{\phi^*}(\mathbf{x},\mathbf{z})$ denote the critic function that maximizes $\text{IBAL}(q_\theta, T_\phi)$. Then,*

$$I_{\text{BA}_L}(q_\theta) \leq \text{IBAL}(q_\theta, T_{\phi^*}) \leq I(\mathbf{x};\mathbf{z}) = I_{\text{BA}_L}(q_\theta) + \mathbb{E}_{p(\mathbf{x})}\left[D_{\text{KL}}[p(\mathbf{z}|\mathbf{x})\|q_\theta(\mathbf{z}|\mathbf{x})]\right]. \qquad (179)$$

*In particular, the contrastive term in Eq. (18) is upper bounded by $\mathbb{E}_{p(\mathbf{x})}\left[D_{\text{KL}}[p(\mathbf{z}|\mathbf{x})\|q_\theta(\mathbf{z}|\mathbf{x})]\right]$.*

*Proof.* The BA bound is a special case of the IBAL$(q_\theta, T_\phi)$ with constant $T_{\phi_0} = c$ and the contrastive term equal to 0. Since $\phi_0$ is a possible parameterization, we can only improve upon $I_{\text{BA}_L}(q_\theta)$ by learning $T_\phi$. The parameterized family of $T_\phi$ may not be expressive enough to match the true log importance weights, so IBAL$(q_\theta, T_{\phi^*}) \leq$ IBAL$(q_\theta, T^*) = I(\mathbf{x}; \mathbf{z})$ using Prop. 4.1. $\qquad\square$

## L.2 PROOF OF IBAL AS LIMITING BEHAVIOR OF THE GIWAE OBJECTIVE AS $K \to \infty$ (PROP. L.2)

**Proposition L.2** (IBAL as Limiting Behavior of GIWAE). *For given $q_\theta(\mathbf{z}|\mathbf{x})$ and $T_\phi(\mathbf{x}, \mathbf{z})$, we have*

$$\lim_{K \to \infty} I_{\text{GIWAE}_L}(q_\theta, T_\phi, K) = \text{IBAL}(q_\theta, T_\phi). \tag{180}$$

*Proof.* Comparing the form of $I_{\text{GIWAE}_L}(q_\theta, T_\phi, K)$ to the IBAL$(q_\theta, T_\phi)$ for a fixed $q_\theta$ and $T_\phi$,

$$I_{\text{GIWAE}}(q_\theta, T_\phi, K) = \mathbb{E}_{p(\mathbf{x}, \mathbf{z})} \left[ \log \frac{q(\mathbf{z}|\mathbf{x})}{p(\mathbf{z})} \right] + \mathbb{E}_{p(\mathbf{x}, \mathbf{z}^{(1)}) \prod_{k=2}^{K} q_\theta(\mathbf{z}^{(k)}|\mathbf{x})} \left[ \log \frac{e^{T_\phi(\mathbf{x}, \mathbf{z}^{(1)})}}{\frac{1}{K} \sum_{i=1}^{K} e^{T_\phi(\mathbf{x}, \mathbf{z}^{(k)})}} \right]$$

$$\text{IBAL}(q_\theta, T_\phi) = \mathbb{E}_{p(\mathbf{x}, \mathbf{z})} \left[ \log \frac{q_\theta(\mathbf{z}|\mathbf{x})}{p(\mathbf{z})} \right] + \mathbb{E}_{p(\mathbf{x}, \mathbf{z})} \left[ \log \frac{e^{T_\phi(\mathbf{x}, \mathbf{z})}}{\mathbb{E}_{q_\theta(\mathbf{z}|\mathbf{x})} \left[ e^{T_\phi(\mathbf{x}, \mathbf{z})} \right]} \right].$$

we can see that both bounds include the same first term $I_{\text{BA}_L}(q_\theta)$ and the same numerator of the contrastive term. To prove the proposition, we can thus focus on characterizing the limiting behavior of the denominator

$$\forall \mathbf{x} : \quad \lim_{K \to \infty} \underbrace{\mathbb{E}_{p(\mathbf{z}^{(1)}|\mathbf{x}) \prod_{k=2}^{K} q_\theta(\mathbf{z}^{(k)}|\mathbf{x})} \left[ \log \frac{1}{K} \sum_{k=1}^{K} e^{T_\phi(\mathbf{x}, \mathbf{z})} \right]}_{= \log \mathcal{Z}_{\text{GIWAE}}(\mathbf{x}, K)} = \underbrace{\log \mathbb{E}_{q_\theta(\mathbf{z}|\mathbf{x})} \left[ e^{T_\phi(\mathbf{x}, \mathbf{z})} \right]}_{= \log \mathcal{Z}_{\theta, \phi}(\mathbf{x})}, \tag{181}$$

where we introduce the notation $\log \mathcal{Z}_{\text{GIWAE}}(\mathbf{x}, K)$ for convenience and the right hand side is the log partition function for the IBAL energy-based posterior $\pi_{\theta, \phi}(\mathbf{z}|\mathbf{x})$. Intuitively, we expect Eq. (181) to hold since the contribution of the single posterior sample $p(\mathbf{z}|\mathbf{x})$ in the GIWAE expectation will vanish as $K \to \infty$.

More formally, we consider the sequence of values $\log \mathcal{Z}_{\text{GIWAE}}(\mathbf{x}, K)$ as a function of $K$. We derive lower and upper bounds on the value of $\log \mathcal{Z}_{\text{GIWAE}}(\mathbf{x}, K)$ for fixed $K$ and show that each of these sequences of lower and upper bounds converge to $\log \mathcal{Z}_{\theta, \phi}(\mathbf{x})$ in the limit as $K \to \infty$. Using the squeeze theorem for sequences, this is sufficient to demonstrate the claim that $\lim_{K \to \infty} \log \mathcal{Z}_{\text{GIWAE}}(\mathbf{x}, K) = \log \mathcal{Z}_{\theta, \phi}(\mathbf{x})$ in Eq. (181). Since the other terms in $I_{\text{GIWAE}_L}(q_\theta, T_\phi, K)$ and IBAL$(q_\theta, T_\phi)$ are identical, this is sufficient to prove the proposition.

*Lower Bound on $\log \mathcal{Z}_{\text{GIWAE}}(\mathbf{x}, K)$:* We rely on the fact that the exponential function $e^{T_\phi(\mathbf{x}, \mathbf{z})} \geq 0$ to simply ignore the contribution of the $p(\mathbf{z}|\mathbf{x})$ term.

$$\log \mathcal{Z}_{\text{GIWAE}}(\mathbf{x}, K) = \mathbb{E}_{p(\mathbf{z}^{(1)}|\mathbf{x}) \prod_{k=2}^{K} q_\theta(\mathbf{z}^{(k)}|\mathbf{x})} \left[ \log \sum_{k=1}^{K} e^{T_\phi(\mathbf{x}, \mathbf{z}^{(k)})} \right] - \log K \tag{182}$$

$$\overset{(1)}{\geq} \mathbb{E}_{\prod_{k=2}^{K} q_\theta(\mathbf{z}^{(k)}|\mathbf{x})} \left[ \log \sum_{k=2}^{K} e^{T_\phi(\mathbf{x}, \mathbf{z}^{(k)})} \right] - \log K \tag{183}$$

$$= \mathbb{E}_{\prod_{k=2}^{K} q_\theta(\mathbf{z}^{(k)}|\mathbf{x})} \left[ \log \frac{1}{K-1} \sum_{k=2}^{K} e^{T_\phi(\mathbf{x}, \mathbf{z}^{(k)})} \right] + \log \frac{K-1}{K} \tag{184}$$

$$=: \text{LB}(\mathbf{x}; K), \tag{185}$$

where in (1), we obtain a lower bound by ignoring the contribution of the positive sample from $p(\mathbf{z}|\mathbf{x})$. The first term is the $k$-sample IWAE lower bound with the proposal $q_\theta(\mathbf{z})$ and target $\pi_{\theta, \phi}(\mathbf{x}, \mathbf{z})$: $\mathbb{E}_{q_\theta(\mathbf{z}^{1:K})}[\log \frac{1}{K} \sum_k \frac{\pi_{\theta, \phi}(\mathbf{x}, \mathbf{z}^{(k)})}{q_\theta(\mathbf{z}^{(k)}|\mathbf{x})}]$, and thus converges to $\log \mathcal{Z}_{\theta, \phi}$ as $K \to \infty$. As $K \to \infty$, we also

have $\log \frac{K-1}{K} \to 0$, so that the limit of their sum is

$$\lim_{K \to \infty} \mathrm{LB}(\mathbf{x}; K) = \log \mathbb{E}_{q_\theta(\mathbf{z}|\mathbf{x})} \left[ e^{T_\phi(\mathbf{x}, \mathbf{z})} \right] = \log \mathcal{Z}_{\theta, \phi}(\mathbf{x}). \tag{186}$$

*Upper Bound on* $\log \mathcal{Z}_{\mathrm{GIWAE}}(\mathbf{x}, K)$: To upper bound $\log \mathcal{Z}_{\mathrm{GIWAE}}(\mathbf{x}, K)$, we separately consider terms arising from $p(\mathbf{z}|\mathbf{x})$ samples and $q_\theta(\mathbf{z}|\mathbf{x})$ samples. Noting that $p_{\mathrm{TGT}}^{\mathrm{GIWAE}}(\mathbf{z}^{(1:K)}, s|\mathbf{x})$ in Eq. (26) is invariant to the index $s$, we can assume $\mathbf{z}^{(1)} \sim p(\mathbf{z}|\mathbf{x})$ and write

$$\log \mathcal{Z}_{\mathrm{GIWAE}}(\mathbf{x}, K) = \mathbb{E}_{p(\mathbf{z}^{(1)}|\mathbf{x}) \prod_{k=2}^{K} q_\theta(\mathbf{z}^{(k)}|\mathbf{x})} \left[ \log \left( \frac{1}{K} e^{T_\phi(\mathbf{x}, \mathbf{z}^{(1)})} + \frac{1}{K} \sum_{k=2}^{K} e^{T_\phi(\mathbf{x}, \mathbf{z}^{(k)})} \right) \right]$$

$$= \mathbb{E}_{p(\mathbf{z}^{(1)}|\mathbf{x}) \prod_{k=2}^{K} q_\theta(\mathbf{z}^{(k)}|\mathbf{x})} \left[ \log \left( \frac{1}{K} e^{T_\phi(\mathbf{x}, \mathbf{z}^{(1)})} + \frac{K-1}{K} \cdot \frac{1}{K-1} \sum_{k=2}^{K} e^{T_\phi(\mathbf{x}, \mathbf{z}^{(k)})} \right) \right]$$

$$\overset{(1)}{\leq} \log \left( \frac{1}{K} \mathbb{E}_{p(\mathbf{z}^{(1)}|\mathbf{x})} \left[ e^{T_\phi(\mathbf{x}, \mathbf{z}^{(1)})} \right] + \mathbb{E}_{\prod_{k=2}^{K} q_\theta(\mathbf{z}^{(k)}|\mathbf{x})} \left[ \frac{K-1}{K} \cdot \frac{1}{K-1} \sum_{k=2}^{K} e^{T_\phi(\mathbf{x}, \mathbf{z}^{(k)})} \right] \right)$$

$$= \log \left( \frac{1}{K} \mathbb{E}_{p(\mathbf{z}^{(1)}|\mathbf{x})} \left[ e^{T_\phi(\mathbf{x}, \mathbf{z}^{(1)})} \right] + \frac{K-1}{K} \cdot \frac{1}{K-1} \sum_{k=2}^{K} \mathbb{E}_{q(\mathbf{z}^{(k)}|\mathbf{x})} \left[ e^{T_\phi(\mathbf{x}, \mathbf{z}^{(k)})} \right] \right)$$

$$= \log \left( \frac{1}{K} \mathbb{E}_{p(\mathbf{z}^{(1)}|\mathbf{x})} \left[ e^{T_\phi(\mathbf{x}, \mathbf{z}^{(1)})} \right] + \frac{K-1}{K} \mathcal{Z}_{\theta, \phi}(\mathbf{x}) \right) \tag{187}$$

$$=: \mathrm{UB}(\mathbf{x}; K). \tag{188}$$

Since $\log(u)$ is a continuous function, we know that $\lim_{K \to \infty} \log(f(K)) = \log \lim_{K \to \infty} f(K)$. We thus reason about the limiting behavior of the terms inside the logarithm in Eq. (187). As $K \to \infty$, we have $\frac{1}{K} \to 0$, and thus the first term inside the log goes to 0. For the second term, we also have $\frac{K-1}{K} \to 1$. Thus, we have

$$\lim_{K \to \infty} \mathrm{UB}(\mathbf{x}; K) = \log \mathbb{E}_{q_\theta(\mathbf{z}|\mathbf{x})} \left[ e^{T_\phi(\mathbf{x}, \mathbf{z})} \right] = \log \mathcal{Z}_{\theta, \phi}(\mathbf{x}). \tag{189}$$

As reasoned above, the convergence of the sequence of both upper and lower bounds to $\log \mathcal{Z}_{\theta, \phi}(\mathbf{x})$ implies that $\lim_{K \to \infty} \log \mathcal{Z}_{\mathrm{GIWAE}}(\mathbf{x}, K) = \log \mathcal{Z}_{\theta, \phi}(\mathbf{x})$. By the reasoning surrounding Eq. (181), this implies $\lim_{K \to \infty} I_{\mathrm{GIWAE}_L}(q_\theta, T_\phi) = \mathrm{IBAL}(q_\theta, T_\phi)$ as desired. $\qquad \square$

### L.3 CONVERGENCE OF GIWAE SNIS DISTRIBUTION TO IBAL ENERGY-BASED POSTERIOR

In this section, we consider the marginal SNIS distribution of GIWAE, which is induced by sampling $K$ times from $q_\theta(\mathbf{z}|\mathbf{x})$ and returning the sample in index $s$ with probability $q_{\mathrm{PROP}}^{\mathrm{GIWAE}}(s|\mathbf{x}, \mathbf{z}^{(1:K)}) \propto e^{T_\phi(\mathbf{x}, \mathbf{z}^{(s)})}$. As $K \to \infty$, we show that this distribution converges to the single-sample energy-based posterior approximation underlying the IBAL and MINE-AIS. A similar observation is made in Sec. 3.2 of Lawson et al. (2019). This result regarding the probabilistic interpretations of GIWAE and the IBAL is complementary to the result in Prop. L.2 regarding the limiting behavior of the bounds.

**Proposition L.3.** *Define the marginal* SNIS *distribution of* GIWAE, $q_{\mathrm{PROP}}^{\mathrm{GIWAE}}(\mathbf{z}|\mathbf{x}; K)$, *using the following sampling procedure*

    *1. Sample from* $s, \mathbf{z}^{(1:K)} \sim q_{\mathrm{PROP}}^{\mathrm{GIWAE}}(s, \mathbf{z}^{(1:K)}|\mathbf{x})$ *according to Eq. (42)*

    *2. Return* $\mathbf{z} = \mathbf{z}^{(s)}$.

*Then, as* $K \to \infty$, *the* KL *divergence (in either direction) between the marginal* SNIS *distribution of* GIWAE *and the energy-based variational distribution of* IBAL, $\pi_{\theta, \phi}(\mathbf{z}|\mathbf{x}) \propto q_\theta(\mathbf{z}|\mathbf{x}) e^{T_\phi(\mathbf{x}, \mathbf{z})}$, *goes to zero.*

$$\lim_{K \to \infty} D_{\mathrm{KL}}[q_{\mathrm{PROP}}^{\mathrm{GIWAE}}(\mathbf{z}|\mathbf{x}; K) \| \pi_{\theta, \phi}(\mathbf{z}|\mathbf{x})] = 0$$

$$\text{and} \quad \lim_{K \to \infty} D_{\mathrm{KL}}[\pi_{\theta, \phi}(\mathbf{z}|\mathbf{x}) \| q_{\mathrm{PROP}}^{\mathrm{GIWAE}}(\mathbf{z}|\mathbf{x}; K)] = 0.$$

*Proof.* To prove the proposition, we consider a mixture target distribution similar to the case of IWAE. However, in this case, we take a single sample from $\pi_{\theta,\phi}(\mathbf{z}|\mathbf{x})$ instead of $p(\mathbf{z}|\mathbf{x})$

$$p_{\text{TGT}}^{\text{GIWAE},\pi}(s, \mathbf{z}^{(1:K)}, \mathbf{x}) = \frac{1}{K}\pi_{\theta,\phi}(\mathbf{z}^{(s)}|\mathbf{x})\prod_{\substack{k=1 \\ k\neq s}}^{K} q_\theta(\mathbf{z}^{(s)}|\mathbf{x}). \tag{190}$$

Note that the probabilistic interpretations $q_{\text{PROP}}^{\text{GIWAE}}(s, \mathbf{z}^{(1:K)}|\mathbf{x})$ and $p_{\text{TGT}}^{\text{GIWAE},\pi}(s, \mathbf{z}^{(1:K)}, \mathbf{x})$ exactly match the IWAE proposal and target distributions in App. B.1 where, in the target distribution, the posterior $p(\mathbf{z}|\mathbf{x})$ has been replaced by $\pi_{\theta,\phi}(\mathbf{z}|\mathbf{x})$. Thus, we can analyze the KL divergences $D_{\text{KL}}[q_{\text{PROP}}^{\text{GIWAE}}(s, \mathbf{z}^{(1:K)}|\mathbf{x})\|p_{\text{TGT}}^{\text{GIWAE},\pi}(s, \mathbf{z}^{(1:K)}|\mathbf{x})]$ and $D_{\text{KL}}[p_{\text{TGT}}^{\text{GIWAE},\pi}(s, \mathbf{z}^{(1:K)}|\mathbf{x})\|q_{\text{PROP}}^{\text{GIWAE}}(s, \mathbf{z}^{(1:K)}|\mathbf{x})]$ using techniques from previous work on IWAE. Following similar arguments as in Domke & Sheldon (2018) (Thm 2) and Cremer et al. (2017), these extended state space KL divergences upper bound the KL divergence between the marginal SNIS distribution and energy-based target, e.g. $D_{\text{KL}}[q_{\text{PROP}}^{\text{GIWAE}}(\mathbf{z}|\mathbf{x};K)\|\pi_{\theta,\phi}(\mathbf{z}|\mathbf{x})] \leq D_{\text{KL}}[q_{\text{PROP}}^{\text{GIWAE}}(s, \mathbf{z}^{(1:K)}|\mathbf{x})\|p_{\text{TGT}}^{\text{GIWAE},\pi}(s, \mathbf{z}^{(1:K)}|\mathbf{x})]$. As $K \to \infty$, $D_{\text{KL}}[q_{\text{PROP}}^{\text{GIWAE}}(s, \mathbf{z}^{(1:K)}|\mathbf{x})\|p_{\text{TGT}}^{\text{GIWAE},\pi}(s, \mathbf{z}^{(1:K)}|\mathbf{x})] \to 0$ since it is an instance of the IWAE gap. Thus, the KL divergence $D_{\text{KL}}[q_{\text{PROP}}^{\text{GIWAE}}(\mathbf{z}|\mathbf{x};K)\|\pi_{\theta,\phi}(\mathbf{z}|\mathbf{x})]$ also vanishes. Similar reasoning applies for the reverse KL divergence. □

# M  MINE-AIS

## M.1  ENERGY-BASED TRAINING OF THE IBAL

Eq. (19) indicates that in order to maximize the IBAL as a function of $\theta$ and $\phi$, we need to increase the value of the energy function $T_\phi$ or score function $\log q_\theta(\mathbf{z}|\mathbf{x})$ on the *real* samples of the true joint $p(\mathbf{x}, \mathbf{z})$ and decrease the value on *fake* samples from the approximate $p(\mathbf{x})\pi_{\theta,\phi}(\mathbf{z}|\mathbf{x})$. However, as is common in training energy-based models, it is difficult to draw samples from $\pi_{\theta,\phi}$.

In order to sample from $\pi_{\theta,\phi}(\mathbf{z}|\mathbf{x})$, a natural approach is to initialize MCMC chains from a sample of the base distribution $q_\theta(\mathbf{z}|\mathbf{x})$, using HMC transition kernels (Neal, 2011), for example. However, we may require infeasibly long MCMC chains when the base distribution is far from desired energy-based model $\pi_{\theta,\phi}(\mathbf{z}|\mathbf{x})$. Instead, we can choose to initialize the chains from the true posterior sample $\mathbf{z} \sim p(\mathbf{z}|\mathbf{x})$ for a simulated data point $\mathbf{x} \sim p(\mathbf{x})$. Letting $\mathcal{T}_{1:M}$ indicate the composition of $T$ transition steps, the approximate energy function gradient becomes

$$\frac{\partial}{\partial\phi}\text{IBAL}(q_\theta, T_\phi) \approx \mathbb{E}_{p(\mathbf{x},\mathbf{z})}\left[\frac{\partial}{\partial\phi}T_\phi(\mathbf{x}, \mathbf{z})\right] - \mathbb{E}_{p(\mathbf{x},\mathbf{z}_0)\mathcal{T}_{1:M}(\mathbf{z}|\mathbf{z}_0,\mathbf{x})}\left[\frac{\partial}{\partial\phi}T_\phi(\mathbf{x}, \mathbf{z})\right], \tag{191}$$

We can use an identical modification for the gradient with respect to $\theta$.

This initialization greatly reduces the computational cost and variance of the estimated gradient and enables training energy functions in high dimensional latent spaces, as shown in our experiments. This approach is in spirit similar to Contrastive Divergence learning of energy-based models (Hinton, 2002) where one starts an MCMC chain from the true data distribution to obtain a lower variance gradient estimate.

## M.2  MULTI-SAMPLE AIS EVALUATION OF THE IBAL

After training the variational base distribution $q_\theta(\mathbf{z}|\mathbf{x})$ and the critic function $T_\phi(\mathbf{x}, \mathbf{z})$ using the MINE-AIS training procedure above, we still need to evaluate the IBAL lower bound on MI, $\text{IBAL}(q_\theta, T_\phi)$. We can easily upper bound $\text{IBAL}(q_\theta, T_\phi)$ using a Multi-Sample AIS lower bound on $\log \mathcal{Z}_{\theta,\phi}(\mathbf{x})$ with expectations under the forward sampling procedure $q_{\text{PROP}}^{\text{AIS}}(\mathbf{z}_{0:T}|\mathbf{x})$. However, an upper bound on $\text{IBAL}(q_\theta, T_\phi)$ is not guaranteed to preserve a lower bound on MI.

In order to obtain a lower bound on the IBAL, we would need to obtain an upper bound on $\log \mathcal{Z}_{\theta,\phi}(\mathbf{x}) = \log \mathbb{E}_{q_\theta(\mathbf{z}|\mathbf{x})}\left[e^{T_\phi(\mathbf{x},\mathbf{z})}\right]$, the log partition function of $\pi_{\theta,\phi}(\mathbf{z}|\mathbf{x})$. However, considering this to be the target distribution $\pi_T(\mathbf{z}|\mathbf{x})$ in Multi-Sample AIS, we would require exact samples from $\pi_{\theta,\phi}(\mathbf{z}|\mathbf{x})$ to guarantee an upper bound on $\log \mathcal{Z}_{\theta,\phi}(\mathbf{x})$. Since these samples are unavailable, we demonstrate conditions under which we can preserve an upper bound on $\log \mathcal{Z}_{\theta,\phi}(\mathbf{x})$ (and lower

bound on $\text{IBAL}(q_\theta, T_\phi)$) by sampling from $p(\mathbf{z}|\mathbf{x})$ instead of $\pi_{\theta,\phi}(\mathbf{z}|\mathbf{x})$ to initialize our backward annealing chains in Prop. M.1 below.

Using the single sample AIS bounds to estimate $\log \mathcal{Z}_{\theta,\phi}(\mathbf{x})$, we have the following extended state space proposal and target distributions,

$$p_{\text{TGT}}^{\text{AIS},\pi}(\mathbf{z}_{0:T}|\mathbf{x}) := \pi_{\theta,\phi}(\mathbf{z}_T|\mathbf{x}) \prod_{t=1}^{T} \tilde{\mathcal{T}}_t(\mathbf{z}_{t-1}|\mathbf{z}_t), \qquad q_{\text{PROP}}^{\text{AIS},\pi}(\mathbf{z}_{0:T}|\mathbf{x}) := q_\theta(\mathbf{z}_0|\mathbf{x}) \prod_{t=1}^{T} \mathcal{T}_t(\mathbf{z}_t|\mathbf{z}_{t-1}).$$
(192)

We emphasize that in $p_{\text{TGT}}^{\text{AIS},\pi}$, $q_{\text{PROP}}^{\text{AIS},\pi}$, the transition kernels and intermediate densities are based on the critic $T_\phi(\mathbf{x}, \mathbf{z})$ and energy-based posterior $\pi_{\theta,\phi}(\mathbf{z}|\mathbf{x})$ whose log partition function we seek to estimate. Recall from Sec. 2.1 that taking the expected log importance weights under $p_{\text{TGT}}^{\text{AIS},\pi}(\mathbf{z}_{0:T}|\mathbf{x})$ yields an upper bound on $\log \mathcal{Z}_{\theta,\phi}(\mathbf{x})$. However, since it is difficult to draw exact samples from $\pi_{\theta,\phi}(\mathbf{z}|\mathbf{x})$ to initialize backward annealing chains and sample from $p_{\text{TGT}}^{\text{AIS},\pi}(\mathbf{z}_{0:T}|\mathbf{x})$, we instead consider sampling from the posterior $p(\mathbf{z}|\mathbf{x})$. Using the same transition kernels $\tilde{\mathcal{T}}_t$ as above, we first define the conditional distribution $p_{\text{TGT}}^{\text{AIS},\pi}(\mathbf{z}_{0:T-1}|\mathbf{x}, \mathbf{z}_T)$ of the backward chain for a given $\mathbf{z}_T$

$$p_{\text{TGT}}^{\text{AIS},\pi}(\mathbf{z}_{0:T-1}|\mathbf{x}, \mathbf{z}_T) := \prod_{t=1}^{T} \tilde{\mathcal{T}}_t(\mathbf{z}_{t-1}|\mathbf{z}_t).$$
(193)

We can then define the *approximate* extended state space target distribution which samples $\mathbf{z}_T \sim p(\mathbf{z}|\mathbf{x})$

$$p_{\text{TGT}}^{\text{APPROX}}(\mathbf{z}_{0:T}|\mathbf{x}) := p(\mathbf{z}_T|\mathbf{x}) \prod_{t=1}^{T} \tilde{\mathcal{T}}_t(\mathbf{z}_{t-1}|\mathbf{z}_t) = p(\mathbf{z}_T|\mathbf{x}) p_{\text{TGT}}^{\text{AIS},\pi}(\mathbf{z}_{0:T-1}|\mathbf{x}, \mathbf{z}_T).$$
(194)

Using this notation, we can also write $p_{\text{TGT}}^{\text{AIS},\pi}(\mathbf{z}_{0:T}|\mathbf{x}) = \pi_{\theta,\phi}(\mathbf{z}_T|\mathbf{x}) p_{\text{TGT}}^{\text{AIS},\pi}(\mathbf{z}_{0:T-1}|\mathbf{x}, \mathbf{z}_T)$ in Eq. (192).

We now characterize the conditions under which sampling from $p_{\text{TGT}}^{\text{APPROX}}(\mathbf{z}_{0:T})$ preserves an upper bound on $\log \mathcal{Z}_{\theta,\phi}(\mathbf{x})$.

**Proposition M.1.** *Define the* AIS *marginal distribution* $q_{\text{PROP}}^{\text{AIS},\pi}(\mathbf{z}_T|\mathbf{x})$ *over the final state in the extended state space proposal as* $q_{\text{PROP}}^{\text{AIS},\pi}(\mathbf{z}_T|\mathbf{x}) := \int q_{\text{PROP}}^{\text{AIS},\pi}(\mathbf{z}_{0:T}|\mathbf{x}) d\mathbf{z}_{0:T-1}$. *If we have*

$$D_{\text{KL}}[p(\mathbf{z}_T|\mathbf{x})\|q_{\text{PROP}}^{\text{AIS},\pi}(\mathbf{z}_T|\mathbf{x})] \geq D_{\text{KL}}[p(\mathbf{z}_T|\mathbf{x})\|\pi_{\theta,\phi}(\mathbf{z}_T|\mathbf{x})],$$
(195)

*then initializing the backward* AIS *chain using* $\mathbf{z}_T \sim p(\mathbf{z}|\mathbf{x})$ *(i.e., sampling under* $p_{\text{TGT}}^{\text{APPROX}}(\mathbf{z}_{0:T}|\mathbf{x})$*), yields an upper bound on* $\log \mathcal{Z}_{\theta,\phi}(\mathbf{x})$,

$$\mathbb{E}_{p_{\text{TGT}}^{\text{APPROX}}(\mathbf{z}_{0:T}|\mathbf{x})} \left[ \log \frac{p_{\text{TGT}}^{\text{AIS},\pi}(\mathbf{x}, \mathbf{z}_{0:T})}{q_{\text{PROP}}^{\text{AIS},\pi}(\mathbf{z}_{0:T}|\mathbf{x})} \right] \geq \log \mathcal{Z}_{\theta,\phi}(\mathbf{x}).$$
(196)

*Proof.* We begin by writing several definitions, which will allow us to factorize the extended state space proposal $q_{\text{PROP}}^{\text{AIS},\pi}(\mathbf{z}_{0:T}|\mathbf{x})$ in the time-reversed direction. This factorization includes the final AIS marginal $q_{\text{PROP}}^{\text{AIS},\pi}(\mathbf{z}_T|\mathbf{x})$.

Starting from Eq. (192), the forward transitions $q_{\text{PROP}}^{\text{AIS},\pi}(\mathbf{z}_{0:T}|\mathbf{x}) = q_\theta(\mathbf{z}_0|\mathbf{x}) \prod_{t=1}^{T} \mathcal{T}_t(\mathbf{z}_t|\mathbf{z}_{t-1})$ induce the marginal distributions $q_{\text{PROP}}^{\text{AIS},\pi}(\mathbf{z}_t|\mathbf{x})$ at each step. The forward transitions and marginals induce a *posterior* kernel $\tilde{\mathcal{T}}_t^q(\mathbf{z}_{t-1}|\mathbf{z}_t)$, which allows us to rewrite $q_{\text{PROP}}^{\text{AIS},\pi}(\mathbf{z}_{0:T}|\mathbf{x})$ using a reverse factorization

$$q_{\text{PROP}}^{\text{AIS},\pi}(\mathbf{z}_{0:T}|\mathbf{x}) = q_{\text{PROP}}^{\text{AIS},\pi}(\mathbf{z}_T|\mathbf{x}) \prod_{t=1}^{T} \tilde{\mathcal{T}}_t^q(\mathbf{z}_{t-1}|\mathbf{z}_t),$$
(197)

$$\text{where} \quad \tilde{\mathcal{T}}_t^q(\mathbf{z}_{t-1}|\mathbf{z}_t) = \frac{q_{\text{PROP}}^{\text{AIS},\pi}(\mathbf{z}_{t-1}|\mathbf{x}) \mathcal{T}_t(\mathbf{z}_t|\mathbf{z}_{t-1})}{q_{\text{PROP}}^{\text{AIS},\pi}(\mathbf{z}_t|\mathbf{x})}.$$
(198)

The posterior reverse transitions $\tilde{\mathcal{T}}_t^q(\mathbf{z}_{t-1}|\mathbf{z}_t)$ are intractable in practice, and cannot be simplified to match the kernels in the target distribution $p_{\text{TGT}}^{\text{AIS},\pi}(\mathbf{z}_{0:T-1}|\mathbf{x}, \mathbf{z}_T) = \prod_{t=1}^{T} \tilde{\mathcal{T}}_t(\mathbf{z}_{t-1}|\mathbf{z}_t)$ using the

invariance or detailed balance conditions. Doucet et al. (2022) provide a promising approach using score matching to approximate these posterior transitions $\tilde{\mathcal{T}}_t^q(\mathbf{z}_{t-1}|\mathbf{z}_t)$.

Finally, we write the posterior reverse process conditioned on a particular $\mathbf{z}_T$ as

$$q_{\text{PROP}}^{\text{AIS},\pi}(\mathbf{z}_{0:T-1}|\mathbf{x},\mathbf{z}_T) := \prod_{t=1}^{T} \tilde{\mathcal{T}}_t^q(\mathbf{z}_{t-1}|\mathbf{z}_t). \tag{199}$$

With the goal of upper bounding $\log \mathcal{Z}_{\theta,\phi}(\mathbf{x}) = \int q_\theta(\mathbf{z}|\mathbf{x})e^{T_\phi(\mathbf{x},\mathbf{z})}d\mathbf{z}$, we consider the log importance weights with expectations under the target distribution $p_{\text{TGT}}^{\text{APPROX}}(\mathbf{z}_{0:T}|\mathbf{x}) = p(\mathbf{z}_T|\mathbf{x})p_{\text{TGT}}^{\text{AIS},\pi}(\mathbf{z}_{0:T-1}|\mathbf{x},\mathbf{z}_T)$, as in Eq. (196). Using the above notation, we have

$$\mathbb{E}_{p_{\text{TGT}}^{\text{APPROX}}(\mathbf{z}_{0:T}|\mathbf{x})}\left[\log\frac{p_{\text{TGT}}^{\text{AIS},\pi}(\mathbf{x},\mathbf{z}_{0:T})}{q_{\text{PROP}}^{\text{AIS},\pi}(\mathbf{z}_{0:T}|\mathbf{x})}\right] = \log \mathcal{Z}_{\theta,\phi}(\mathbf{x}) + \mathbb{E}_{p_{\text{TGT}}^{\text{APPROX}}(\mathbf{z}_{0:T}|\mathbf{x})}\left[\log\frac{p_{\text{TGT}}^{\text{AIS},\pi}(\mathbf{z}_{0:T}|\mathbf{x})}{q_{\text{PROP}}^{\text{AIS},\pi}(\mathbf{z}_{0:T}|\mathbf{x})}\right] \tag{200}$$

$$= \log \mathcal{Z}_{\theta,\phi}(\mathbf{x}) + \mathbb{E}_{p_{\text{TGT}}^{\text{APPROX}}(\mathbf{z}_{0:T}|\mathbf{x})}\left[\log\frac{\pi_{\theta,\phi}(\mathbf{z}_T|\mathbf{x})p_{\text{TGT}}^{\text{AIS},\pi}(\mathbf{z}_{0:T-1}|\mathbf{x},\mathbf{z}_T)}{q_{\text{PROP}}^{\text{AIS},\pi}(\mathbf{z}_T|\mathbf{x})q_{\text{PROP}}^{\text{AIS},\pi}(\mathbf{z}_{0:T-1}|\mathbf{x},\mathbf{z}_T)}\frac{p(\mathbf{z}_T|\mathbf{x})}{p(\mathbf{z}_T|\mathbf{x})}\right] \tag{201}$$

$$= \log \mathcal{Z}_{\theta,\phi}(\mathbf{x}) + D_{\text{KL}}\big[p(\mathbf{z}_T|\mathbf{x})\|q_{\text{PROP}}^{\text{AIS},\pi}(\mathbf{z}_T|\mathbf{x})\big] - D_{\text{KL}}\big[p(\mathbf{z}_T|\mathbf{x})\|\pi_{\theta,\phi}(\mathbf{z}_T|\mathbf{x})\big] \tag{202}$$

$$+ \mathbb{E}_{p(\mathbf{z}_T|\mathbf{x})}\big[D_{\text{KL}}[p_{\text{TGT}}^{\text{AIS},\pi}(\mathbf{z}_{0:T-1}|\mathbf{x},\mathbf{z}_T)\|q_{\text{PROP}}^{\text{AIS},\pi}(\mathbf{z}_{0:T-1}|\mathbf{x},\mathbf{z}_T)]\big]$$

The intractable KL divergence $D_{\text{KL}}[p_{\text{TGT}}^{\text{AIS},\pi}(\mathbf{z}_{0:T-1}|\mathbf{x},\mathbf{z}_T)\|q_{\text{PROP}}^{\text{AIS},\pi}(\mathbf{z}_{0:T-1}|\mathbf{x},\mathbf{z}_T)]$ compares the reverse kernels in the target distribution $\tilde{\mathcal{T}}_t(\mathbf{z}_{t-1}|\mathbf{z}_t)$ against the posterior $\tilde{\mathcal{T}}_t^q(\mathbf{z}_{t-1}|\mathbf{z}_t)$. Ignoring this nonnegative term, we can lower bound the expectation on the LHS

$$\mathbb{E}_{p_{\text{TGT}}^{\text{APPROX}}}\left[\log\frac{p_{\text{TGT}}^{\text{AIS},\pi}(\mathbf{x},\mathbf{z}_{0:T})}{q_{\text{PROP}}^{\text{AIS},\pi}(\mathbf{z}_{0:T}|\mathbf{x})}\right] \geq \log \mathcal{Z}_{\theta,\phi}(\mathbf{x}) + D_{\text{KL}}\big[p(\mathbf{z}_T|\mathbf{x})\|q_{\text{PROP}}^{\text{AIS},\pi}(\mathbf{z}_T|\mathbf{x})\big] - D_{\text{KL}}\big[p(\mathbf{z}_T|\mathbf{x})\|\pi_{\theta,\phi}(\mathbf{z}_T|\mathbf{x})\big].$$

Finally, under the assumption of the proposition that $D_{\text{KL}}\big[p(\mathbf{z}_T|\mathbf{x})\|q_{\text{PROP}}^{\text{AIS},\pi}(\mathbf{z}_T|\mathbf{x})\big] \geq D_{\text{KL}}\big[p(\mathbf{z}_T|\mathbf{x})\|\pi_{\theta,\phi}(\mathbf{z}_T|\mathbf{x})\big]$, we have the desired result. □

**KL Divergence Condition** In Prop. M.1, we have shown that we can preserve an upper bound on $\log \mathcal{Z}_{\theta,\phi}(\mathbf{x})$ by initializing the reverse chain using a posterior sample, under a condition on the KL divergence,

$$D_{\text{KL}}\big[p(\mathbf{z}_T|\mathbf{x})\|q_{\text{PROP}}^{\text{AIS},\pi}(\mathbf{z}_T|\mathbf{x})\big] \geq D_{\text{KL}}\big[p(\mathbf{z}_T|\mathbf{x})\|\pi_{\theta,\phi}(\mathbf{z}_T|\mathbf{x})\big]. \tag{203}$$

While we cannot guarantee this condition, we intuitively expect Eq. (203) to hold in practice since $\pi_{\theta,\phi}(\mathbf{z}|\mathbf{x})$ has been directly trained to match $p(\mathbf{z}|\mathbf{x})$. By contrast, $q_{\text{PROP}}^{\text{AIS},\pi}(\mathbf{z}_T|\mathbf{x})$ is the final state of an AIS procedure, which approximates $\pi_{\theta,\phi}(\mathbf{z}|\mathbf{x})$ and does not have access to information about $p(\mathbf{z}|\mathbf{x})$. Burda et al. (2015) use a similar approach for lower bounding the log likelihood in EBMs, but give an example of a Restricted Boltzmann Machines (RBM) model (in their Sec. 5) in which Eq. (203) does not hold.

As desired, we find in our experiments in Fig. 7 that our approximate reverse annealing procedure underestimates the IBAL in all of the MINE-AIS, GIWAE and INFONCE experiments, for all numbers of intermediate distributions $T$.

**T = 1 Special Case** For $T = 1$, we have $q_{\text{PROP}}^{\text{AIS},\pi}(\mathbf{z}_1|\mathbf{x}) = q_\theta(\mathbf{z}_1|\mathbf{x})$. In particular, Prop. M.1 will provide an upper bound on $\log \mathcal{Z}_{\theta,\phi}(\mathbf{x})$ if

$$D_{\text{KL}}[p(\mathbf{z}|\mathbf{x})\|q_\theta(\mathbf{z}|\mathbf{x})] \geq D_{\text{KL}}[p(\mathbf{z}|\mathbf{x})\|\pi_{\theta,\phi}(\mathbf{z}|\mathbf{x})]. \tag{204}$$

This condition is guaranteed under the assumptions of Prop. L.1, where IBAL$(q_\theta, T_{\phi^*}) = I_{\text{BA}_L}(\pi_{\theta,\phi^*})$ improves upon $I_{\text{BA}_L}(q_\theta)$ for an energy function $T_{\phi^*}(\mathbf{x},\mathbf{z})$ which has been trained to maximize the IBAL. In Eq. (204) the KL divergence on the left-hand side corresponds to the gap in the BA lower

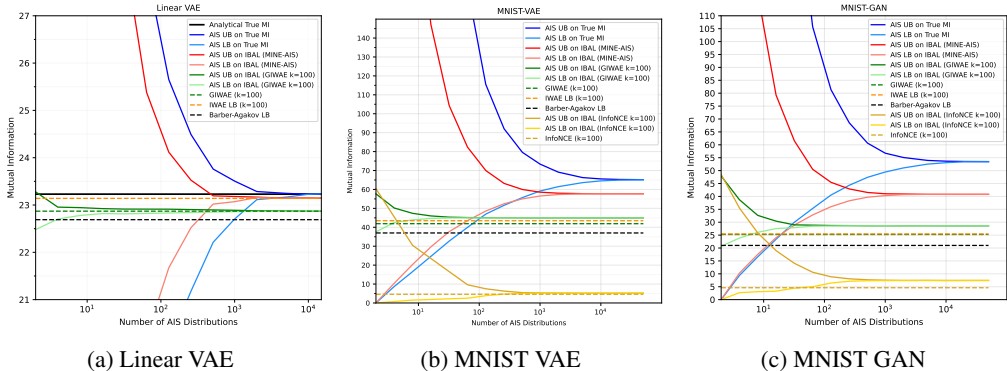

Figure 7: Estimating the mutual information of deep generative models with AIS and MINE-AIS.

bound, which is larger than the gap in the IBAL on the right-hand side. We can further show that the lower bound on $\text{IBAL}(q_\theta, T_\phi)$ resulting from Prop. M.1 with $T = 1$ is the BA lower bound,

$$
\begin{aligned}
\text{IBAL}(q_\theta, T_\phi) &= \mathbb{E}_{p(\mathbf{x},\mathbf{z})}\left[\log \frac{q_\theta(\mathbf{z}|\mathbf{x})}{p(\mathbf{z})}\right] + \mathbb{E}_{p(\mathbf{x},\mathbf{z})}[T_\phi(\mathbf{x},\mathbf{z})] - \mathbb{E}_{p(\mathbf{x})}[\log \mathcal{Z}_{\theta,\phi}(\mathbf{x})] \\
&\geq \mathbb{E}_{p(\mathbf{x},\mathbf{z})}\left[\log \frac{q_\theta(\mathbf{z}|\mathbf{x})}{p(\mathbf{z})}\right] + \mathbb{E}_{p(\mathbf{x},\mathbf{z})}[T_\phi(\mathbf{x},\mathbf{z})] - \mathbb{E}_{p_{\text{TGT}}^{\text{APPROX}}}\left[\log \frac{p_{\text{TGT}}^{\text{AIS},\pi}(\mathbf{x},\mathbf{z}_{0:T})}{q_{\text{PROP}}^{\text{AIS},\pi}(\mathbf{z}_{0:T}|\mathbf{x})}\right] \\
&= \mathbb{E}_{p(\mathbf{x},\mathbf{z})}\left[\log \frac{q_\theta(\mathbf{z}|\mathbf{x})}{p(\mathbf{z})}\right] + \mathbb{E}_{p(\mathbf{x},\mathbf{z})}[T_\phi(\mathbf{x},\mathbf{z})] - \mathbb{E}_{p(\mathbf{x},\mathbf{z})}\left[\log \frac{q_\theta(\mathbf{z}|\mathbf{x})e^{T_\phi(\mathbf{x},\mathbf{z})}}{q_\theta(\mathbf{z}|\mathbf{x})}\right] \\
&= \mathbb{E}_{p(\mathbf{x},\mathbf{z})}\left[\log \frac{q_\theta(\mathbf{z}|\mathbf{x})}{p(\mathbf{z})}\right] + \cancel{\mathbb{E}_{p(\mathbf{x},\mathbf{z})}[T_\phi(\mathbf{x},\mathbf{z})]} - \cancel{\mathbb{E}_{p(\mathbf{x},\mathbf{z})}[T_\phi(\mathbf{x},\mathbf{z})]} \\
&= I_{\text{BA}_L}(q_\theta).
\end{aligned}
$$

We can confirm this in Fig. 7, where for $T = 1$, the approximate lower bounds on $\text{IBAL}(q_\theta, T_\phi)$ begin from the appropriate BA lower bound. For example, in the case of IBAL evaluation for GIWAE ($K = 100$), the light green curve starts from the BA lower bound term reported in the decomposition of the GIWAE ($K = 100$) lower bound in Fig. 3a. Using more intermediate distributions ($T > 1$) for the AIS approximate bound in Prop. M.1, the estimates in Fig. 7 approach the true value of $\text{IBAL}(q_\theta, T_\phi)$ in all cases.

## M.3 MINE-AIS EXPERIMENTS

In Fig. 7, we assess the performance of our MINE-AIS method, which learns an energy function $T_\phi$ and uses Multi-Sample AIS to evaluate lower and upper bounds on the $\text{IBAL}(q_\theta, T_\phi)$, in cases where $p(\mathbf{x}|\mathbf{z})$ is unknown. As a comparison, we consider Multi-Sample AIS lower and upper bounds with known $p(\mathbf{x}|\mathbf{z})$, since these estimators tightly bound the ground truth MI and correspond to the optimal energy function in MINE-AIS (Prop. 4.1). We train MINE-AIS using $\text{IBAL}(p(\mathbf{z}), T_\phi)$ for $p(\mathbf{z}) = \mathcal{N}(0, I)$, and plot our Multi-Sample AIS lower and upper bounds on the IBAL as a function of the number of intermediate AIS distributions $T$ used for evaluation.

Note that our lower bound on IBAL, which uses the approximate reverse annealing procedure from App. M.2, preserves a lower bound on MI for all values of $T$. The upper bound on IBAL may be used to validate the convergence of the Multi-Sample AIS evaluation procedure. For example, in Fig. 7, we observe that our lower and upper bounds converge to the same estimate for high $T$, finding the true value of $\text{IBAL}(p(\mathbf{z}), T_\phi)$.

We find that MINE-AIS underestimates the ground truth MI by $11\%$, and $24\%$ on MNIST-VAE and MNIST-GAN, respectively. We also compare against the BA lower bound with a Gaussian variational family for $q_\theta(\mathbf{z}|\mathbf{x})$. Note that the IBAL corresponds to the BA lower bound for a more flexible, energy-based posterior approximation $\pi_{\theta,\phi}(\mathbf{z}|\mathbf{x})$, and both bounds assume access to only a single known marginal distribution. We find that MINE-AIS outperforms the BA baseline by $2\%$, $52\%$, and $90\%$, on linear VAE, MNIST-VAE, and MNIST-GAN, respectively.

We also use Multi-Sample AIS to evaluate the IBAL corresponding to $(q_\theta, T_\phi)$, which are learned by optimizing the GIWAE (with $K = 100$) and INFONCE (with $K = 100$ and $q_\theta = p(\mathbf{z})$) lower bounds. As argued in Prop. L.2, the IBAL corresponds to the limiting behavior of GIWAE as $K \to \infty$. We observe that the AIS evaluation of the IBAL corresponding to GIWAE or INFONCE critic functions only marginally improves upon evaluation of the original GIWAE or INFONCE lower bounds. This indicates that the improvement of MINE-AIS over GIWAE or INFONCE can be primarily attributed to learning a better critic function using energy-based training. For these evaluation tasks, the approximate reverse annealing procedure from App. M.2 also underestimates the IBAL for all values of $T$.

## N   APPLICATIONS TO MUTUAL INFORMATION ESTIMATION WITHOUT KNOWN MARGINALS

While the focus in of our work is evaluating the mutual information $I(\mathbf{x}; \mathbf{z})$ in settings where at least a single marginal is available, we are often interested in estimating or optimizing the mutual information where no marginal distribution is available. A natural setting where no marginal distribution is available is representation learning where the goal is to maximize the mutual information between the data distribution $q_{\text{data}}(\mathbf{x})$ and the representation induced by a stochastic mapping $q_\psi(\mathbf{z}|\mathbf{x})$ parameterized by $\psi$,

$$I(\mathbf{x}; \mathbf{z}) = \mathbb{E}_{q_{\text{data}}(\mathbf{x})q_\psi(\mathbf{z}|\mathbf{x})}\left[\log \frac{q_\psi(\mathbf{x}, \mathbf{z})}{q_{\text{data}}(\mathbf{x})q_\psi(\mathbf{z})}\right] = \mathbb{E}_{q_{\text{data}}(\mathbf{x})q_\psi(\mathbf{z}|\mathbf{x})}\left[\log \frac{q_\psi(\mathbf{x}|\mathbf{z})}{q_{\text{data}}(\mathbf{x})}\right] \quad (205)$$

where $q_\psi(\mathbf{z}) = \int q_{\text{data}}(\mathbf{x})q_\psi(\mathbf{z}|\mathbf{x})d\mathbf{x}$ represents the "aggregated posterior" (Makhzani et al., 2015) or induced marginal distribution over $\mathbf{z}$.

### N.1   BA LOWER BOUND

Using the notation of Eq. (205), we can write the BA lower bound as

$$I(\mathbf{x}; \mathbf{z}) \geq I(\mathbf{x}; \mathbf{z}) - \underbrace{\mathbb{E}_{q_\psi(\mathbf{z})}\big[D_{\text{KL}}[q_\psi(\mathbf{x}|\mathbf{z})\|p_\theta(\mathbf{x}|\mathbf{z})]\big]}_{\text{gap}}$$

$$= \mathbb{E}_{q_{\text{data}}(\mathbf{x})}\mathbb{E}_{q_\psi(\mathbf{z}|\mathbf{x})}\left[\log \frac{p_\theta(\mathbf{x}|\mathbf{z})}{q_{\text{data}}(\mathbf{x})}\right] \quad (206)$$

$$= \mathbb{E}_{q_{\text{data}}(\mathbf{x})}\underbrace{\mathbb{E}_{q_\psi(\mathbf{z}|\mathbf{x})}\left[\log p_\theta(\mathbf{x}|\mathbf{z})\right]}_{\text{negative reconstruction loss of } \mathbf{x}} + \underbrace{H_{\text{data}}(\mathbf{x})}_{\text{constant}} \quad (207)$$

$$=: I_{\text{BA}_L}\big(p_\theta(\mathbf{x}|\mathbf{z})\big) \quad (208)$$

where $p_\theta(\mathbf{x}|\mathbf{z})$ is a variational distribution, parameterized by $\theta$, that tries to match to the *inverse encoding distribution* $q_\psi(\mathbf{x}|\mathbf{z}) \propto q_{\text{data}}(\mathbf{x})q_\psi(\mathbf{z}|\mathbf{x})$. The first term in Eq. (207) can be interpreted as the reconstruction term, and the second term is the entropy of the data distribution, which is constant.

**Evaluating MI up to a Constant**   Note that the gradient of $I_{\text{BA}_L}\big(p_\theta(\mathbf{x}|\mathbf{z})\big)$ with respect to the parameters $\theta$ does not depend on the marginal $q_{\text{data}}(\mathbf{x})$, and thus we may still optimize the variational distribution when the data distribution is unknown. We can then use the resulting $p_\theta(\mathbf{x}|\mathbf{z})$ to estimate the BA lower bound on mutual information *up to a constant*, $H_{\text{data}}(\mathbf{x})$. This is useful in comparing the MI induced by two different representations $q_{\psi_1}(\mathbf{z}|\mathbf{x})$ and $q_{\psi_2}(\mathbf{z}|\mathbf{x})$ of the same data distribution.

**Optimizing MI with the BA Lower Bound**   The BA lower bound is also amenable to backpropagation through the parameters of the encoding distribution $q_\psi(\mathbf{z}|\mathbf{x})$, even when analytic marginal densities for $q_{\text{data}}(\mathbf{x})$ or $q_\psi(\mathbf{z})$ are not available.

Maximization of the BA lower bound appears in various settings, including in representation learning (Alemi et al., 2016; 2018), reinforcement learning (Mohamed & Rezende, 2015), improving interpretability in GANs (Chen et al., 2016), and variational information bottleneck methods (Tishby et al., 2000; Alemi et al., 2016; 2018).

## N.2 GIWAE LOWER BOUND

In this section, we discuss the applicability of our GIWAE lower bound in mutual information maximization settings. Rewriting the GIWAE lower bound in Eq. (7) for the case of estimating $I(\mathbf{x}; \mathbf{z})$ in Eq. (205), we have

$$
I(\mathbf{x}; \mathbf{z}) \geq \underbrace{\mathbb{E}_{q_{\text{data}}(\mathbf{x}) q_\psi(\mathbf{z}|\mathbf{x})} \big[ \log p_\theta(\mathbf{x}|\mathbf{z}) \big]}_{\text{negative reconstruction loss of } \mathbf{x}} + \underbrace{H_{\text{data}}(\mathbf{x})}_{\text{constant}} + \underbrace{\mathbb{E}_{q_\psi(\mathbf{x},\mathbf{z}) \prod_{k=2}^{K} p_\theta(\mathbf{x}^{(k)}|\mathbf{z})} \left[ \log \frac{e^{T_\phi(\mathbf{x},\mathbf{z})}}{\frac{1}{K}(e^{T_\phi(\mathbf{x},\mathbf{z})} + \sum_{k=2}^{K} e^{T_\phi(\mathbf{x}^{(k)},\mathbf{z})})} \right]}_{\text{contrastive term} \leq \log K}
$$

$$
=: I_{\text{GIWAE}_L}(p_\theta(\mathbf{x}|\mathbf{z}), T_\phi, K)
$$

(209)

Note that we can use a single joint sample from $q_{\text{data}}(\mathbf{x}) q_\psi(\mathbf{z}|\mathbf{x}) = q_\psi(\mathbf{z}) q_\psi(\mathbf{x}|\mathbf{z})$ to obtain a positive sample from the inverse encoding distribution $\mathbf{x} \sim q_\psi(\mathbf{x}|\mathbf{z})$ for a particular $\mathbf{z} \sim q_\psi(\mathbf{z})$, in a similar fashion to our ancestral sampling in Sec. 1.1. For a given $\mathbf{z}$, the negative samples correspond to $K-1$ samples from the stochastic decoder $\mathbf{x}^{(2:K)} \sim p_\theta(\mathbf{x}|\mathbf{z})$.

From the importance sampling perspective, we can view the GIWAE lower bound as corresponding to an upper bound on the "log partition function" $\log q_\psi(\mathbf{z})$ for a particular $\mathbf{z} \sim q_\psi(\mathbf{z})$. The contrastive term in Eq. (209) arises from SNIS sampling of a single $\mathbf{x}^{(s)}$ in the extended state space proposal, with $q_{\text{PROP}}^{\text{GIWAE}}(s|\mathbf{z}, \mathbf{x}^{(1:K)}) = e^{T_\phi(\mathbf{x}^{(s)}, \mathbf{z})} / \sum_{k=1}^{K} e^{T_\phi(\mathbf{x}^{(k)}, \mathbf{z})}$.

Finally, we note that optimization over the energy function $T_\phi(\mathbf{x}, \mathbf{z})$ improves upon $I_{\text{BA}_L}(p_\theta(\mathbf{x}|\mathbf{z}))$ in Eq. (207) by at most $\log K$ nats using the contrastive term.

**Evaluating MI up to a Constant** Similar to the BA bound, the GIWAE lower bound can be used to evaluate the mutual information up to the constant data entropy when $q_{\text{data}}(\mathbf{x})$ is unknown. This may be useful in comparing the MI induced by two different representations $q_{\psi_1}(\mathbf{z}|\mathbf{x})$ and $q_{\psi_2}(\mathbf{z}|\mathbf{x})$ of the same data distribution. Note that the reconstruction and contrastive terms in Eq. (209) depend on samples from $q_{\text{data}}(\mathbf{x})$ and not the density. Thus, our ability to take gradients with respect to the parameters of the variational distribution $p_\theta(\mathbf{x}|\mathbf{z})$ or energy function $T_\phi(\mathbf{x}, \mathbf{z})$ are not affected by the fact that the marginal distribution is unknown.

**Optimizing MI with the GIWAE Lower Bound** The GIWAE lower bound may also be used for mutual information maximization with respect to the parameters of $q_\psi(\mathbf{z}|\mathbf{x})$, since each term in Eq. (209) is amenable to backpropagation. Since GIWAE generalizes both the BA and INFO-NCE lower bounds, it can be used as a drop-in replacement for either of these bounds for optimizing MI (e.g., see van den Oord et al. (2018)).

## N.3 MINE-AIS / IBAL LOWER BOUND

Recall that MINE-AIS estimation would involve an energy based variational approximation to the inverse encoding distribution $q_\psi(\mathbf{x}|\mathbf{z})$,

$$
\pi_{\theta,\phi}(\mathbf{x}|\mathbf{z}) = \frac{1}{\mathcal{Z}(\mathbf{z})} p_\theta(\mathbf{x}|\mathbf{z}) e^{T_\phi(\mathbf{x},\mathbf{z})}. \tag{210}
$$

Note that IBAL lower bound on $I(\mathbf{x}; \mathbf{z})$ involves an intractable log partition function term $\mathbb{E}_{q_\psi(\mathbf{z})}[\log \mathcal{Z}(\mathbf{z})] = \mathbb{E}_{q_\psi(\mathbf{z})}\big[ \log \mathbb{E}_{p_\theta(\mathbf{x}|\mathbf{z})} \big[ e^{T_\phi(\mathbf{x},\mathbf{z})} \big] \big]$.

$$
I(\mathbf{x}; \mathbf{z}) \geq \underbrace{\mathbb{E}_{q_{\text{data}}(\mathbf{x})} \mathbb{E}_{q_\psi(\mathbf{z}|\mathbf{x})} \big[ \log p_\theta(\mathbf{x}|\mathbf{z}) \big]}_{\text{negative reconstruction loss of } \mathbf{x}} + \underbrace{H_{\text{data}}(\mathbf{x})}_{\text{constant}} + \underbrace{\mathbb{E}_{q_\psi(\mathbf{x},\mathbf{z})} \left[ \log \frac{e^{T_\phi(\mathbf{x},\mathbf{z})}}{\mathbb{E}_{p_\theta(\mathbf{x}|\mathbf{z})} \big[ e^{T_\phi(\mathbf{x},\mathbf{z})} \big]} \right]}_{\text{contrastive term} \leq \mathbb{E}_{q_\psi(\mathbf{z})}\big[ D_{\text{KL}}[q_\psi(\mathbf{x}|\mathbf{z}) \| p_\theta(\mathbf{x}|\mathbf{z})] \big]}
$$

(211)

$$
=: \text{IBAL}(p_\theta(\mathbf{x}|\mathbf{z}), T_\phi)
$$

where the term in the denominator is the partition function $\mathcal{Z}(\mathbf{z}) = \mathbb{E}_{p_\theta(\mathbf{x}|\mathbf{z})} \big[ e^{T_\phi(\mathbf{x},\mathbf{z})} \big]$.

When taking gradients as in Eq. (19)-(20), we obtain

$$\frac{\partial}{\partial \theta} \text{IBAL}(p_\theta, T_\phi) = \mathbb{E}_{q_{\text{data}}(\mathbf{x}) q_\psi(\mathbf{z}|\mathbf{x})} \left[ \frac{\partial}{\partial \theta} \log p_\theta(\mathbf{x}|\mathbf{z}) \right] - \mathbb{E}_{q_\psi(\mathbf{z}) \pi_{\theta,\phi}(\mathbf{x}|\mathbf{z})} \left[ \frac{\partial}{\partial \theta} \log p_\theta(\mathbf{x}|\mathbf{z}) \right], \quad (212)$$

$$\frac{\partial}{\partial \phi} \text{IBAL}(p_\theta, T_\phi) = \mathbb{E}_{q_{\text{data}}(\mathbf{x}) q_\psi(\mathbf{z}|\mathbf{x})} \left[ \frac{\partial}{\partial \phi} T_\phi(\mathbf{x}, \mathbf{z}) \right] - \mathbb{E}_{q_\psi(\mathbf{z}) \pi_{\theta,\phi}(\mathbf{x}|\mathbf{z})} \left[ \frac{\partial}{\partial \phi} T_\phi(\mathbf{x}, \mathbf{z}) \right]. \quad (213)$$

To obtain approximate negative samples from $\pi_{\theta,\phi}(\mathbf{x}|\mathbf{z})$ for a given $\mathbf{z}$, we can use MCMC transition kernels since the unnormalized target density $\tilde{\pi}_{\theta,\phi}(\mathbf{x}|\mathbf{z}) = p_\theta(\mathbf{x}|\mathbf{z}) e^{T_\phi(\mathbf{x},\mathbf{z})}$ is tractable.

**Evaluating MI up to a Constant** Since only samples from $q_{\text{data}}(\mathbf{x})$ are required for the MINE-AIS training procedure in Eq. (212)-(213), we can learn the base variational distribution $p_\theta(\mathbf{x}|\mathbf{z})$ and energy function $T_\phi(\mathbf{x}, \mathbf{z})$ in cases when the marginal distribution $q_{\text{data}}(\mathbf{x})$ is unknown.

As in the case of GIWAE and BA lower bounds above, we can also evaluate the IBAL lower bound in Eq. (211) up to a constant. As in the main text, this involves using multi-sample AIS techniques to bound the intractable log partition function term $\log \mathcal{Z}(\mathbf{z}) = \log \mathbb{E}_{p_\theta(\mathbf{x}|\mathbf{z})} \left[ e^{T_\phi(\mathbf{x},\mathbf{z})} \right]$.

**Optimizing MI with the MINE-AIS Lower Bound** If we are interested in optimizing the MINE-AIS lower bound with respect to the parameters of a stochastic encoder $q_\psi(\mathbf{z}|\mathbf{x})$, we would need to backpropagate through the Multi-Sample AIS procedure used to estimate the $\log \mathcal{Z}(\mathbf{z})$ term, but computing the gradients are intractable. We thus conclude that, among our proposed methods, GIWAE is the most directly applicable in settings of MI for representation learning.

## O  EXPERIMENTAL DETAILS

### O.1  EXPERIMENT DETAILS OF SEC. 5.2

In this section, we provide the experiment details used in Sec. 5.2. For more details, see the public GitHub repository, https://github.com/huangsicong/ais_mi_estimation.

#### O.1.1  DATASETS AND MODELS

We used MNIST (LeCun et al., 1998) and CIFAR-10 (Krizhevsky & Hinton, 2009) datasets in our experiments.

**Real-Valued MNIST** For the VAE experiments on the real-valued MNIST dataset (Table 1), the encoder's architecture is $784 - 1024 - 1024 - 1024 - z$, where $z$ is the latent code size shown in the row header of Table 1. The decoder architecture is the reverse of the encoder architecture. The decoder variance is learned scalar.

For the GAN experiments on MNIST (Table 1), we used the same decoder architecture as our VAEs. In order to stabilize the training dynamics, we used the gradient penalty (GP) (Salimans et al., 2016).

The network was trained for 300 epochs with the learning rate of 0.0001 using the Adam optimizer (Kingma & Ba, 2014), and the checkpoint with the best validation loss was used for the evaluation.

**CIFAR-10** For the CIFAR-10 experiments (Table 1), we experimented with a smaller version of DCGAN (Radford et al., 2015) (see the public code). The number at the end of each model name in Table 1 indicates the latent code size.

#### O.1.2  EXPERIMENT DETAILS OF SEC. 5.2

For the AIS temperature schedule, We used sigmoid schedules as used in Wu et al. (2016). The step size of HMC was adaptively tuned to achieve an average acceptance probability of 65% as suggested in Neal (2001). For all MNIST experiments in Table 1, we evaluated on a single batch size of 128 simulated data. For all CIFAR experiments in Table 1 we used a single batch of 32 simulated data. All experiments are run on on Tesla P100 or Quadro RTX 6000 or Tesla T4 GPUs.

### O.1.3 RUNTIME COMPARISON

We benchmarked the runtime on Tesla P100 GPUs. For MNIST, it took about 35 minutes to run IWAE with $K = 1M$, about 8 hours to run AIS with $T = 30K$. For CIFAR, it took about 45 minutes to run IWAE with $K = 1M$, and about 12 hours for the AIS with $T = 100K$.

In Fig. 8, we evaluate the tradeoff between runtime and bound tightness for evaluating the generative MI for VAE and GAN models with 100-dimensional latent codes trained on the MNIST dataset. We compare IWAE and multi-sample AIS evaluation, with the same experimental settings as in App. O.1.2 and an initial distribution $q_\theta(\mathbf{z}|\mathbf{x})$. We plot wall clock time on the $x$-axis, where increasing runtime reflects increasing $K$ for IWAE and increasing $T$ for AIS.

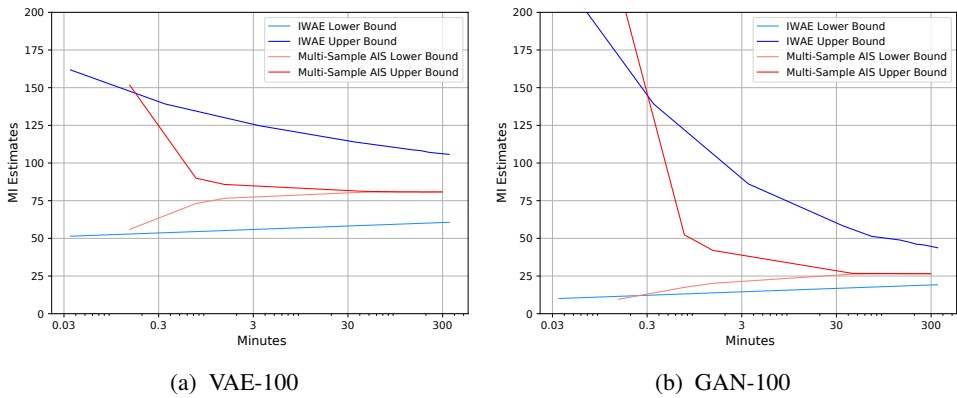

(a) VAE-100          (b) GAN-100

Figure 8: Runtime vs. Bound Tightness for VAE (left) and GAN (right) models on MNIST.

## O.2 EXPERIMENT DETAILS FOR ENERGY-BASED BOUNDS (SEC. 5.2)

**Models and Data** The linear VAE model has a Gaussian prior $\mathbf{z} \sim \mathcal{N}(0, I)$ with the dimension of 10. The dimension of the output $\mathbf{x}$ is 100. The weights are sampled randomly from a Gaussian distribution with the standard deviation of 1, and the standard deviation of the Gaussian observation noise at the output is 1. The MNIST-VAE20, is a VAE model that is trained on the real-valued MNIST dataset. It has a Gaussian prior $\mathbf{z} \sim \mathcal{N}(0, I)$ with the dimension of 20. The decoder has one layer of ReLU non-linearity of size 1000, followed by a linear layer that predicts the mean of a Gaussian observation model with the fixed standard deviation of 0.1. MNIST-GAN20 decoder uses one layer of ReLU non-linearity of size 1000 followed by a sigmoid layer that predicts the mean of a Gaussian observation model with the fixed standard deviation of 0.1. In all the experiments, we use a fixed batch of 100 data points.

**BA, IWAE and GIWAE** For the BA bound, $q_\theta(\mathbf{z}|\mathbf{x})$ is parameterized using a neural network that predicts the mean and log-std of a diagonal Gaussian distribution. The neural network has two layers of ReLU non-linearity of size 2000. The IWAE experiments used the same architecture for $q_\theta(\mathbf{z}|\mathbf{x})$. The GIWAE experiments, in addition to $q_\theta(\mathbf{z}|\mathbf{x})$ with the same architecture, used a critic function $T_\phi(\mathbf{x}, \mathbf{z})$ that is parameterized by a neural network that concatenates $(\mathbf{x}, \mathbf{z})$ and pass them through two layers of ReLU non-linearity of size 2000, followed by a linear layer that outputs a scalar value. The parameters of $q_\theta(\mathbf{z}|\mathbf{x})$ and $T_\phi(\mathbf{x}, \mathbf{z})$ are trained jointly.

**Multi-Sample AIS** For the Multi-Sample AIS bounds, for all models, we used up to $K = 1000$ chains (see Fig. 5), and up to $T = 50K$ intermediate distributions with linear schedule (see Fig. 5 and Fig. 7). We used HMC as the AIS kernel, with $\epsilon = 0.02$ and $L = 20$ leap frog steps.

**MINE-AIS** For the training of the MINE-AIS, we used a critic function $T_\phi(\mathbf{x}, \mathbf{z})$ that is parameterized by a neural network that concatenates $(\mathbf{x}, \mathbf{z})$ and pass them through three layers of ReLU non-linearity of size 2000, followed by a linear layer that outputs a scalar value. We chose the Gaussian prior $\mathcal{N}(0, I)$ as the base distribution $q_\theta(\mathbf{z}|\mathbf{x})$. In order to take a sample from $\pi_{\theta,\phi}(\mathbf{z}|\mathbf{x})$, we used the HMC method that is initialized with a true posterior $p(\mathbf{z}|\mathbf{x})$ sample, with $M = 10$ iterations each with $L = 20$ leapfrog steps. For the step size of HMC, we used $\epsilon = 0.05$ for the linear VAE model

and $\epsilon = 0.02$ for the MNIST-VAE20 and MNIST-GAN20. For the evaluation of MINE-AIS, we used Multi-Sample AIS with the same parameters as the AIS evaluation experiments described above.

### O.3 ANALYTICAL SOLUTION OF THE MUTUAL INFORMATION ON THE LINEAR MNIST-VAE

In order to verify our implementations, we have derived the MI analytically for the linear VAEs and verified that it matches the MI estimated by AIS. For simplicity, we assume a fixed identity covariance matrix $\mathbf{I}$ at the output of the conditional likelihood of the linear VAE decoder, i.e., the decoder of the VAE is simply: $\mathbf{x} = \mathbf{W}\mathbf{z} + \mathbf{b} + \boldsymbol{\epsilon}$, where $\mathbf{x}$ is the observation, $\mathbf{z}$ is the latent code vector $\mathbf{z} \sim \mathcal{N}(\mathbf{0}, \mathbf{I})$, $\mathbf{W}$ is the decoder weight matrix and $\mathbf{b}$ is the bias. The observation noise of the decoder is $\boldsymbol{\epsilon} \sim \mathcal{N}(\mathbf{0}, \mathbf{I})$. It is easily shown that the conditional likelihood is $p(\mathbf{x}|\mathbf{z}) = \mathcal{N}(\mathbf{x}|\mathbf{W}\mathbf{z} + \mathbf{b}, \mathbf{I})$ and thus we can solve for the marginal

$$p(\mathbf{x}) = \mathcal{N}(\mathbf{x}|\boldsymbol{\mu}_{\mathbf{x}} = \mathbf{b}, \boldsymbol{\Sigma}_{\mathbf{x}} = \mathbf{I} + \mathbf{W}\mathbf{W}^{\intercal}). \tag{214}$$

The differential entropy of $\mathbf{x}$ is:

$$H(\mathbf{x}) = \frac{k}{2} + \frac{k}{2}\log(2\pi) + \frac{1}{2}\log(\det\boldsymbol{\Sigma}_{\mathbf{x}}), \tag{215}$$

where k is the dimension of the observation. The conditional entropy is

$$H(\mathbf{x}|\mathbf{z}) = \frac{k}{2} + \frac{k}{2}\log(2\pi) + \frac{1}{2}\log(\det\mathbf{I}) \tag{216}$$

$$= \frac{k}{2} + \frac{k}{2}\log(2\pi). \tag{217}$$

Thus, the mutual information is

$$I(\mathbf{x}; \mathbf{z}) = H(\mathbf{x}) - H(\mathbf{x}|\mathbf{z}) \tag{218}$$

$$= \frac{1}{2}\log(\det\boldsymbol{\Sigma}_{\mathbf{x}}). \tag{219}$$

### O.4 CONFIDENCE INTERVALS FOR MULTI-SAMPLE AIS EXPERIMENTS

Table 4 and Table 5 provides the 95% confidence intervals for the AIS results reported in Table 1. The confidence intervals were computed over confidence interval over 8 batches each with 16 data points for MNIST; and over 8 batches 4 data points for CIFAR.

| Method | Proposal | | VAE10 | VAE100 | GAN10 | GAN100 |
|---|---|---|---|---|---|---|
| **AIS** T=1 | $p(\mathbf{z})$ | UB | (1849.03, 2010.66) | (5564.53, 6096.51) | (745.68, 826.55) | (795.81, 926.94) |
| | | LB | (0.00, 0.00) | (0.00, 0.00) | (0.00, 0.00) | (0.00, 0.00) |
| | $q(\mathbf{z}\|\mathbf{x})$ | UB | (59.34, 66.66) | (345.45, 378.80) | (293.60, 335.83) | (482.40, 544.26) |
| | | LB | (60.09, 66.61) | (32.46, 36.52) | (3.41, 3.94) | (2.40, 2.83) |
| **AIS** T=500 | $p(\mathbf{z})$ | UB | (38.61, 39.57) | (92.32, 98.03) | (21.91, 23.04) | (26.89, 28.22) |
| | | LB | (33.69, 34.41) | (77.77, 82.02) | (21.18, 21.96) | (25.36, 26.36) |
| | $q(\mathbf{z}\|\mathbf{x})$ | UB | (33.94, 34.63) | (79.96, 84.71) | (22.53, 23.58) | (28.82, 30.24) |
| | | LB | (33.80, 34.53) | (78.01, 82.37) | (21.19, 22.00) | (25.09, 26.06) |
| **AIS** T=30K | $p(\mathbf{z})$ | UB | (33.82, 34.60) | (78.62, 83.07) | (21.54, 22.50) | (25.97, 27.07) |
| | | LB | (33.86, 34.56) | (78.51, 83.05) | (21.50, 22.45) | (25.96, 26.98) |
| | $q(\mathbf{z}\|\mathbf{x})$ | UB | (33.85, 34.58) | (78.56, 83.03) | (21.54, 22.47) | (26.01, 27.07) |
| | | LB | (33.86, 34.56) | (78.52, 83.02) | (21.55, 22.48) | (26.02, 27.03) |
| **IWAE** K=1 | $p(\mathbf{z})$ | UB | (3628.86, 4026.29) | (10705.98, 12297.86) | (1556.00, 1704.00) | (1680.50, 1800.28) |
| | | LB | (0.00, 0.00) | (0.00, 0.00) | (0.00, 0.00) | (0.00, 0.00) |
| | $q(\mathbf{z}\|\mathbf{x})$ | UB | (34.82, 35.86) | (92.82, 98.44) | (38.80, 76.14) | (243.34, 278.41) |
| | | LB | (24.54, 25.86) | (41.53, 47.54) | (3.85, 4.61) | (2.94, 3.52) |
| **IWAE** K=1K | $p(\mathbf{z})$ | UB | (1132.54, 1262.97) | (3987.51, 4480.86) | (430.40, 463.19) | (462.95, 526.51) |
| | | LB | (6.91, 6.91) | (6.91, 6.91) | (6.91, 6.91) | (6.91, 6.91) |
| | $q(\mathbf{z}\|\mathbf{x})$ | UB | (33.87, 34.61) | (83.15, 87.46) | (34.11, 71.36) | (182.82, 219.55) |
| | | LB | (31.08, 32.30) | (48.44, 54.45) | (10.75, 11.52) | (9.84, 10.43) |
| **IWAE** K=1M | $p(\mathbf{z})$ | UB | (352.91, 400.87) | (2078.30, 2417.17) | (71.93, 91.09) | (100.27, 127.75) |
| | | LB | (13.82, 13.82) | (13.82, 13.82) | (13.79, 13.82) | (13.82, 13.82) |
| | $q(\mathbf{z}\|\mathbf{x})$ | UB | (33.87, 34.57) | (81.42, 85.36) | (28.51, 33.26) | (52.41, 63.68) |
| | | LB | (33.67, 34.53) | (55.35, 61.36) | (17.31, 18.21) | (16.63, 17.32) |

Table 4: Confidence intervals of AIS and IWAE estimates of MI on MNIST. UB stands for Upper Bound, and LB stands for Lower Bound.

| Model | Proposal | | GAN10 | GAN100 |
|---|---|---|---|---|
| **AIS** (T=1) | $p(\mathbf{z})$ | UB | (3593559.12, 4477712.38) | (4038603.09, 5668217.91) |
| | | LB | (0.00, 0.00) | (0.00, 0.00) |
| | $q(\mathbf{z}\|\mathbf{x})$ | UB | (241342.62, 566015.82) | (1874938.58, 2881576.42) |
| | | LB | (15.70, 18.89) | (18.41, 21.93) |
| **AIS** (T=500) | $p(\mathbf{z})$ | UB | (24683.10, 41496.70) | (25453.02, 101127.79) |
| | | LB | (28.41, 30.63) | (98.37, 110.65) |
| | $q(\mathbf{z}\|\mathbf{x})$ | UB | (116.02, 156.28) | (1603.13, 3969.92) |
| | | LB | (45.45, 50.88) | (134.35, 156.02) |
| **AIS** (T=100K) | $p(\mathbf{z})$ | UB | (72.17, 75.80) | (479.04, 497.10) |
| | | LB | (70.19, 73.55) | (470.05, 490.47) |
| | $q(\mathbf{z}\|\mathbf{x})$ | UB | (71.87, 75.22) | (475.07, 494.61) |
| | | LB | (71.35, 74.75) | (468.64, 489.89) |
| **IWAE** (K=1) | $p(\mathbf{z})$ | UB | (6019901.15, 9511489.86) | (7339411.95, 12492792.05) |
| | | LB | (0.00, 0.00) | (0.00, 0.00) |
| | $q(\mathbf{z}\|\mathbf{x})$ | UB | (75.00, 80.03) | (−1810.70, 12504.41) |
| | | LB | (16.28, 18.62) | (18.39, 21.60) |
| **IWAE** (K=1K) | $p(\mathbf{z})$ | UB | (1673333.03, 2415008.47) | (2181660.84, 3531768.16) |
| | | LB | (6.91, 6.91) | (6.91, 6.91) |
| | $q(\mathbf{z}\|\mathbf{x})$ | UB | (72.31, 75.68) | (−1872.12, 12438.39) |
| | | LB | (22.18, 24.99) | (25.32, 28.64) |
| **IWAE** (K=1M) | $p(\mathbf{z})$ | UB | (582880.62, 838142.63) | (1479829.44, 2327879.56) |
| | | LB | (13.82, 13.82) | (13.82, 13.82) |
| | $q(\mathbf{z}\|\mathbf{x})$ | UB | (71.63, 75.08) | (−1883.22, 12426.34) |
| | | LB | (29.86, 31.60) | (32.21, 35.42) |

Table 5: Confidence intervals of AIS and IWAE estimates of MI on CIFAR. UB stands for Upper Bound, and LB stands for Lower Bound.

