# OpenReview forum: "Improving Mutual Information Estimation with Annealed and Energy-Based Bounds"
_ICLR.cc/2022/Conference — ICLR 2022 Poster_

### Official Review · Reviewer_eNs9 · 2021-10-26

**Correctness:** 4
**Technical Novelty And Significance:** 3
**Empirical Novelty And Significance:** 3
**Recommendation:** 8
**Confidence:** 5

**Main Review:**

First I identify myself as Reviewer cNMQ from the last round of peer review of this paper. I see the author(s) have carefully revised their manuscript to address the concerns from the reviews. In particular, the current draft has made clear the distinction between parametric and non-parametric MI estimation, both are important research topics and they are fitted for different application scenarios. Not only that, the presentation has been improved. Better visualization of the bounds has been provided to help understand the relationship between different estimators, and there are many in-place remarks and discussions on the connection to existing bounds. I believe this is a piece of solid work and enthusiastically recommend acceptance.

Strength
* Importance sampling is a key technique in statistical inference and its application to MI estimation constructs a promising direction for research. Such motivated, this paper has derived a family of multi-sample novel bounds with annealed proposals.
* The connection between the InfoNCE bound and the generalized multi-sample IWAE MI bound is quite fascinating. The author(s) have made an interesting point in the statement that the naive choice of proposal p(z) has led to the well-known log-K cap of InfoNCE bound, which I believe is critical for improving MI estimation. The current presentation is underselling its significance, so I would encourage the author(s) to enrich the discussions and give more concrete examples for easier understanding.
* Clarity of presentation. I am glad to see the significant improvements made compared to the last submission.


Below are some minor suggestions that might help the author(s) to further improve the quality of this manuscript.

Technical points:

A. While the importance sampling perspective for MI estimation is novel, this is not new in the broader context of variational inference. The following works [1-3] are highly relevant: they all follow an annealed path to tighten the bound. The discussions on these works are currently missing from the manuscript, which I think would significantly boost the significance of this work by making strong connections.

[1] T Salimans, et al. Markov Chain Monte Carlo and Variational Inference: Bridging the Gap. ICML 2015
[2] L Maaloe, et al. Auxiliary Deep Generative Models. ICML 2016
[3] V Masrani, et al. The Thermodynamic Variational Objective. NeurIPS 2019

B. Another major missing piece is the discussion of whether this strategy may inadvertently harm learning. This point is related to the seminal work of [4], which analyzed a particular scenario that tight multi-sample VI bounds actually harm inference. Intuitively, the inference arm is the approximate posterior (i.e., proposal distribution), and since tighter bounds are less sensitive to the quality of approximate posterior, the algorithm is less incentivized to improve it. Related to my technical point A, [3] has provided some perspectives on additional tradeoffs involved. Nonetheless, a careful analysis is beyond the scope of this study, so I only encourage the author(s) to give some thoughts and make some remarks in the paper.

[4] T Rainforth. Tighter Variational Bounds are Not Necessarily Better. ICML 2018

C. The derivation of GIWAE is one of the key contributions of this paper, but there is no intuition provided in the main text on how this is derived. The author(s) simply say "we define GIWAE lower bound". At least the probabilistic interpretation should appear in the main paper. Also, the author(s) should mention the disadvantage relative to the InfoNCE estimator. In particular, InfoNCE admits an efficient bi-linear implementation [5], because the z is sampled from the prior. I think the difference between IWAE and GIWAE should be further emphasized, say using a table comparing what is needed for both of them, and changing the order of the first two sentences of Sec 2.3

[5] T Chen, et al. A Simple Framework for Contrastive Learning of Visual Representations. ICML 2020

D. Sorry to see the section "Energy-based Bounds" in the original writing moved to Sec 4, personally, I think it can be presented earlier.  And I do not fully appreciate the importance of Sec 3.3, could the author(s) elaborate?

E. Selecting K, T is a joint optimization problem, but the discussion in Sec 3.4 is singled threaded: it only talks about suggestions when either K ot T is fixed.

Non-technical points:

F. Please add runtime analysis, and highlight the tradeoff between computation/memory and bound tightness.

G. Please give the explicit expression of I_{IWAE} in the paper.

H. Please use the booktab package to format the Tables. Follow the practice of minimalism, remove unnecessary horizontal lines and all vertical lines in the tables. Actually, for Table 1 a figure might be more intuitive.

I. Figure 2 (b) is excellent. I think it is better to present it earlier in the text, right where the bounds are discussed.

**Summary Of The Paper:**

This paper investigates annealed importance sampling (AIS) techniques and multi-sample bounds to improve mutual information estimation. Assuming the knowledge of extract likelihood (joint or marginal), this paper shows the mutual information bound can be tightened via exploiting a series of intermediate distributions interpolating between a (convenient) proposal and the joint density. An energy-based perspective was presented and discussed, revealing interesting insights. The effectiveness of this proposal is demonstrated with generative modeling applications on real-world datasets.

**Summary Of The Review:**

This is an interesting work and my major concerns from the last round of paper reviews have been adequately addressed. As such I recommend acceptance.

---

> ### Author Response · Authors · 2021-11-17
> **Response to Reviewer eNs9**
>
> First, we would like to thank the reviewer for the constructive feedback in both rounds for reviews, which have significantly improved our paper.
>
> > The derivation of GIWAE is one of the key contributions of this paper, but there is no intuition provided in the main text on how this is derived...
> > The current presentation is underselling the significance of GIWAE, so I would encourage the author(s) to enrich the discussions
>
> We agree with the reviewer that the probabilistic interpretation of GIWAE is insightful, and we have included this in the revised version of the manuscript.  We have also included a discussion of GIWAE in the context of representation learning and mutual information maximization in App. D.
>
> >  I do not fully appreciate the importance of Sec 3.3, could the author(s) elaborate?
>
> We provide two different multi-sample AIS methods for upper bounding the log partition function: Independent Multi-Sample AIS and Coupled Reverse Multi-Sample AIS.  These upper bounds have different tradeoffs, as we discuss in detail in App. M1 and Sec. 3.4. In short, the stochasticity in the Independent Multi-Sample AIS EUBO comes from $K-1$ independent 'negative' forward chains, while the stochasticity in the Coupled Reverse AIS EUBO is induced by MCMC transitions in $K-1$ coupled backward chains (see Fig. 2).
>
> Our probabilistic interpretation of Coupled Reverse Multi-Sample AIS also sheds light on the $\log p(x)$ upper bound used in BDMC (Grosse et. al 2015, 2016).  We can also understand Coupled Reverse Multi-Sample AIS as an extension of *Reverse IWAE* (App. I) using AIS proposals, in a similar fashion as to how Independent Multi-Sample AIS extends standard IWAE.
>
> > Fig. 2 (b) is excellent. I think it is better to present it earlier in the text.
>
> Thank you for the positive feedback.  We feel Fig 2(b) (now Fig. 3(b)) is particularly useful for understanding the results in Fig. 3(a).   However, as per your suggestion, we have included Fig. 1 in the Introduction to graphically summarize the relationships between the bounds discussed in this paper.
>
> > Sorry to see the section "Energy-based Bounds" in the original writing moved to Sec 4, personally, I think it can be presented earlier.
>
> We were forced to omit the “Energy-Based Bounds” section due to space limitations.  However, in App. E,  we have now provided probabilistic interpretations for MINE-DV and MINE-F, which allow us to derive novel *Generalized MINE* bounds.   In a similar spirit to GIWAE, these bounds use a variational distribution $q(z|x)$ to tighten the MINE-DV and MINE-F bounds when $p(z)$ is available.  App. E Fig. 5 summarizes the relationships between these bounds.
>
> >InfoNCE admits an efficient bi-linear implementation [5], because the z is sampled from the prior.
>
> Thank you for raising this point.  We have included this comment in the main text.
>
> > The following works [1-3] are highly relevant: they all follow an annealed path to tighten the bound.
>
> Thank you for pointing out these references.   We have mentioned this related work in App. G when discussing single-sample AIS bounds on $\log p(x)$.
>
> > Rainforth et al. analyzed a particular scenario that tight multi-sample VI bounds actually harm inference. Intuitively...since tighter bounds are less sensitive to the quality of approximate posterior, the algorithm is less incentivized to improve it.
>
> We agree, and our experiments support this observation.  Using the GIWAE perspective (see Eq. 40), we can decompose the IWAE lower bound on MI into the sum of the BA lower bound and a K-sample contrastive term.  Note that the BA term indicates the quality of the base variational distribution $q(z|x)$, since its gap from the (fixed) MI is $E_{p(x)}[D_{KL}[p(z|x)||q(z|x)]]$.
>
> Table 2 empirically shows this decomposition of the IWAE lower bound on MI.   Although the overall IWAE lower bound is higher for $K=1000$ versus $K=100$, we learn a worse variational distribution for $K=1000$ in all cases, as evidenced by the lower BA term.  A worse $q(z|x)$ makes it easier for the energy function $T(x,z)$ to distinguish samples from $q(z|x)$ and $p(z|x)$, and obtain close to the full $\log K$ improvement from contrastive estimation.
>
> Since the IWAE lower bound on MI uses an upper bound on $\log p(x)$, this observation can be viewed as the upper bound analogue of the Rainforth et al.’s observation for IWAE lower bounds on $\log p(x)$.
>
> With more complex multi-sample AIS proposals, we also see in Table 1 that the dependence of our bounds on the initial distribution ($p(z)$ or $q(z|x)$) diminishes with increasing T.
>
> > Selecting K, T is a joint optimization problem, but the discussion in Sec 3.4 is singled threaded
>
> Since our bounds tighten with respect to increasing either K or T, we do not treat these choices as a joint optimization problem.
>
> > Please give the explicit expression of I_{IWAE} in the paper.
>
> We have included an explicit form of the IWAE lower bound in the revised paper.

---

### Official Review · Reviewer_78Uf · 2021-11-02

**Correctness:** 4
**Technical Novelty And Significance:** 4
**Empirical Novelty And Significance:** 3
**Recommendation:** 6
**Confidence:** 4

**Main Review:**

Strength:
(1) This work unifies the view of mutual information from the perspective of importance sampling. This gives us another angle of thinking.
(2) Good summary of the prior literature
(3) Clear theoretical analysis.
(4) The experiments provide evidence on the effectiveness of the proposals.

Weakness:
(1) This paper is more of a theoretical flavor and there is no experiment to show this method is practically useful. For example, is it useful to improve the prediction accuracy (using the learned latent representation z)?
(2) How about the efficiency of the proposed strategies. (e.g. information about the running time). In the experiments, the author(s) have used tens of thousands of annealing steps, which may not be practically feasible. Also, it is helpful to visualize the trade-offs in computing time and bound sharpness.
(3) I am curious about the stability of the training process.
(4) The method has limited applicability in real-world settings, Generative modeling for feature engineering is not used that often, mostly because it is so hard to assume the knowledge of a likelihood model for a machine learning problem.

Minor Comments:
(1) Make clear distinction between empirical estimators and theoretical estimators. For example, I_{BA} is theoretical and I_{IWAE} is empirical. Please add hat over the empirical estimators.
(2) In Figure 1, Z_{T-1} is bigger than other circles. Is there any different meanings?
(3) Table 1 is messy, tidy it up. There too much data and at the first glance, no idea about where I should focus on. Also, there is inconsistency in notations: in the first column, T=30K, K=1K. Should change the capital letter K to lower case letter k.



**Summary Of The Paper:**

In this work, the authors proposed some novel estimators for mutual information, namely the GIWAE, Muti-Sample AIS and MINE-AIS bounds. The key idea is to apply importance sampling to mutual information in the model-based setting. In both theoretical and empirical analysis, they prove the bounds are much tighter than the existing ones in the literature.

**Summary Of The Review:**

The reviewer is convinced by the theoretical analysis of this paper and the novel perspectives it proposed.
It makes some valuable contribution to our community. However, I am concerned about the practicability and efficiency of this method.

---

> ### Author Response · Authors · 2021-11-17
> **Response to Reviewer 78Uf**
>
> > This paper is more of a theoretical flavor and there is no experiment to show this method is practically useful.
>
> We point out that MI estimation is an important and fundamental problem with applications across information theory and machine learning.  Our experiments show that our multi-sample AIS bounds are the only practical bounds which can accurately estimate large values of MI without exponential sample complexity.
>
> > For example, is it useful to improve the prediction accuracy (using the learned latent representation z)?
>
> Based on the reviewers’ feedback, we have also included a comprehensive discussion of our methods (in particular, GIWAE) in the context of representation learning and mutual information maximization in App. D.  See the general response for more details.
>
> > it is so hard to assume the knowledge of a likelihood model for a machine learning problem
>
> We agree, and our GIWAE and MINE-AIS methods are indeed motivated by this observation.   These methods can be used to lower bound MI without a likelihood model and with only a single known marginal distribution.
>
> > I am curious about the stability of the training process.
>
> We did not encounter stability issues in training for GIWAE, which inherits the same low-variance properties as IWAE or InfoNCE.  We also found that MINE-AIS was stable, since the energy-based training procedure avoids the direct sampling from p(z) that causes instability in MINE (McAllester & Stratos 2018).   We have provided confidence intervals for our multi-sample AIS evaluation results in App. M Tables 4 & 5.
>
> > In the experiments, the author(s) have used tens of thousands of annealing steps, which may not be practically feasible.  Also, it is helpful to visualize the trade-offs in computing time and bound sharpness.
>
> Thanks for this suggestion!  We have included Fig. 8 in App. L3, which shows the tradeoff between runtime and bound tightness for both AIS and IWAE evaluation on VAE-100 and GAN-100 models trained on MNIST. See App. L3 for further runtime details.
>
> > Table 1 is messy, tidy it up. In Figure 1, Z_{T-1} is bigger than other circles. Is there any different meanings?
>
> Thank you for pointing these out.  We have updated Table 1 to highlight tight MI bounds in bold, and ensured that z_{T-1} has the same sizing as other samples in Fig. 1 (now Fig. 2).
>
> > Make clear distinction between empirical estimators and theoretical estimators. For example, I_{BA} is theoretical and I_{IWAE} is empirical.
>
> In our notation, I_BA and I_IWAE refer to expectations of their corresponding estimators, and are not themselves estimators (nor random variables).  Accordingly, we have not used the hat notation, which is usually reserved for estimators.

---

### Official Review · Reviewer_YniD · 2021-11-03

**Correctness:** 4
**Technical Novelty And Significance:** 3
**Empirical Novelty And Significance:** 4
**Recommendation:** 8
**Confidence:** 4

**Main Review:**

Strengths:
* The authors present a comprehensive framework for mutual information bounds, called GIWAE, that unifies many existing approaches.
* Combining AIS with IWAE is elegant, especially when one can define a sensible approximation for $p(z | x)$ in the form of either a prior $p(z)$ or approximate posterior $q(z | x)$ as with a VAE. The MINE-AIS extension is also intriguing due to its potential to extend the proposed techniques far beyond the VAE and GAN scenarios considered in this work.
* The experimental results are quite impressive and convincingly demonstrate the utility of the proposed method.
* The authors provide a comprehensive presentation of their method and an exhaustive review and comparison with other methods.

Weaknesses:
* The MINE-AIS technique is not fully explored. It would be interesting to see MINE-AIS applied in a scenario where the IWAE-type AIS bounds are not applicable (i.e. when $p( x | z )$ is unknown) but this is not investigated.
* The core of the proposed method is previously suggested in a blog post by Sobolev. I believe this is not a significant weakness because the impressive effectiveness of this approach was not previously demonstrated.

Other Comments:
* I am unsure of why a single $z$ is used in the numerator but multiple $z$ are used in the denominator of (7) and other IWAE-type equations. A brief explanation would aid my understanding.

**Summary Of The Paper:**

This work combines the technique of Annealed Importance Sampling (AIS) with Importance Weighted Autoencoder (IWAE) bounds on mutual information to obtain new and significantly tighter bounds. They examine the situations where the joint distribution $p(x, z)$ is known via $p(z)$ and $p(x | z)$, and the situation where only $p(z)$ is known. In the former case, they obtain exact bounds (or asymptotically exact given large enough sample sizes), and in the latter case they find approximate bounds by learning an energy function to approximate $p(x |z)$. Their experiments show the utility of their proposed method. In particular, they obtain impressively tight lower and upper bounds for complex latent variable generative models (VAE and GAN) on complex data such as CIFAR-10.

**Summary Of The Review:**

Overall, this is an impressive work both in terms of its comprehensive mathematical presentation and its high-quality experimental results. Personally, I find the experimental results to be the most interesting aspect, because it opens the possibility of accurate MI estimation in complex scenarios studied in contemporary generative modeling. I recommend this work be accepted.

---

> ### Author Response · Authors · 2021-11-17
> **Response to Reviewer YniD**
>
> >  It would be interesting to see MINE-AIS applied in a scenario where the IWAE-type AIS bounds are not applicable (i.e. when p(x|z) is unknown) but this is not investigated.
>
>
> We specifically designed our MINE-AIS experiments for settings where we *do* have the ground truth MI (obtained by multi-sample AIS), so that we could evaluate the bias of MINE-AIS estimates and the effectiveness of our energy-based training procedure.
>
> Scenarios where only a single marginal distribution is available appear in simulation-based inference (Cranmer et. al 2020), where information about input parameters $\theta$ is known and a black-box simulator can generate $x$ for a given $\theta$ , but the likelihood $p(x|\theta)$ is intractable.   We have included this comment in the revised paper.
>
> > The core of proposed method is previously suggested in a blog post by Sobolev. I believe this is not a significant weakness because impressive effectiveness of this approach was not previously demonstrated
>
> The blog post of Sobolev only mentions the single-sample BDMC bound on $\log p(x)$ (“AIS” in Fig. 2) , which we generalize to multi-sample upper and lower bounds in two distinct ways (“Independent” and “Coupled Reverse” Multi-Sample AIS in Fig. 2).  As the reviewer notes, the blog post of Sobolev does not provide any experimental validation. Our contribution also includes the engineering effort to scale multi-sample AIS for tightly bounding large values of MI.
>
> > I am unsure of why a single z is used in the numerator but multiple z are used in the denominator of (7) and other IWAE-type equations. A brief explanation would aid my understanding.
>
> This denominator arises from the self-normalized importance sampling interpretation of IWAE or GIWAE. We have moved the probabilistic interpretation of GIWAE from the appendix to the main text to clarify this point.

---

### Author Response · Authors · 2021-11-17
**Summary of Revisions**

We thank the reviewers for their positive responses and constructive feedback.

We have uploaded a revised manuscript based on the reviewers’ feedback, and have highlighted changes from the original submission in red.  We summarize the notable changes below, and refer to minor changes in the individual responses.
- We have included Figure 1, which summarizes the MI bounds discussed in this paper and the relationships between them.
- As requested by Reviewer eNs9, we have moved the probabilistic interpretation of GIWAE from the appendix to the main paper.
- As requested by Reviewer 78Uf, we have included a comprehensive discussion of our bounds in the context of representation learning in App D.   In the main text, we emphasize the point that GIWAE is amenable to backpropagation and can be used to maximize mutual information even if no marginal distribution is available.  We thus conclude that GIWAE may be used as a drop-in replacement for BA or InfoNCE.  We plan to investigate this direction in future work.
- As requested by Reviewers 78Uf and eNs9, we have included Fig. 8 in App. L3 to visualize the tradeoff between runtime and bound tightness for both AIS and IWAE.
- As requested by Reviewer eNs9, in App. E we have included a full discussion of our methods in relation to “Energy-Based Bounds” such as MINE.   In particular, we provide probabilistic interpretations for MINE-DV and MINE-F, which allow us to derive novel *Generalized MINE* bounds. In a similar spirit to GIWAE, these bounds use a variational distribution $q(z|x)$ to tighten the MINE-DV and MINE-F bounds when $p(z)$ is available.  App. E Fig. 5 summarizes the relationships between these bounds.

---

### Decision · Program_Chairs · 2022-01-20

**Decision:**

Accept (Poster)

**Comment:**

This paper investigates a tighter bound for mutual information and proposes some novel estimators of MI from the importance sampling perspective. The proposed approach provides a unifying framework for mutual information bounds that can deduce many existing approaches. The theoretical and experimental analyses well justify the proposed approach and shows the bounds are much tighter than the existing ones.
Overall, this paper is well written. The relevant literatures are exhaustively reviewed and well compared with the proposed method. The experimental results show remarkable superiority of the proposed method. The proposed framework would shed light on the literatures and open up a new direction of the relevant researches.
For those reasons, I would like to recommend this paper to be accepted by ICLR2022 conference.